# The hagfish genome and the evolution of vertebrates

Ferdinand Marlétaz[1,2 ✉], Nataliya Timoshevskaya[3], Vladimir A. Timoshevskiy[3], Elise Parey[1], Oleg Simakov[2,4], Daria Gavriouchkina[2,10], Masakazu Suzuki[5], Kaoru Kubokawa[6], Sydney Brenner[7,11], Jeremiah J. Smith[3 ✉] & Daniel S. Rokhsar[2,8,9 ✉]

As the only surviving lineages of jawless fishes, hagfishes and lampreys provide a crucial window into early vertebrate evolution[1–3]. Here we investigate the complex history, timing and functional role of genome-wide duplications[4–7] and programmed DNA elimination[8,9] in vertebrates in the light of a chromosome-scale genome sequence for the brown hagfish *Eptatretus atami*. Combining evidence from syntenic and phylogenetic analyses, we establish a comprehensive picture of vertebrate genome evolution, including an auto-tetraploidization ($1R_V$) that predates the early Cambrian cyclostome–gnathostome split, followed by a mid–late Cambrian allo-tetraploidization ($2R_{JV}$) in gnathostomes and a prolonged Cambrian–Ordovician hexaploidization ($2R_{CY}$) in cyclostomes. Subsequently, hagfishes underwent extensive genomic changes, with chromosomal fusions accompanied by the loss of genes that are essential for organ systems (for example, genes involved in the development of eyes and in the proliferation of osteoclasts); these changes account, in part, for the simplification of the hagfish body plan[1,2]. Finally, we characterize programmed DNA elimination in hagfish, identifying protein-coding genes and repetitive elements that are deleted from somatic cell lineages during early development. The elimination of these germline-specific genes provides a mechanism for resolving genetic conflict between soma and germline by repressing germline and pluripotency functions, paralleling findings in lampreys[10,11]. Reconstruction of the early genomic history of vertebrates provides a framework for further investigations of the evolution of cyclostomes and jawed vertebrates.

Hagfishes are deep-sea scavengers with a prodigious capacity for producing slime[12] (Fig. 1a). As one of only two surviving lineages of jawless fishes, hagfishes provide a unique comparative perspective on early vertebrate evolution. Both hagfishes and lampreys stand apart from jawed vertebrates (gnathostomes) in the absence of jaws, bone and dentine[3], and they have been grouped together as cyclostomes[13], the sister group to gnathostomes. However, hagfishes lack several key characteristics that are shared by lampreys and gnathostomes, including definitive vertebrae[14], lensed eyes with oculomotor control and electroreceptive sensory organs[1,2]. The relative simplicity of the hagfish body plan suggests an alternative hypothesis whereby hagfishes diverged before a craniate clade that groups lampreys with jawed vertebrates[3]. Early molecular phylogenies (albeit with limited sequence datasets and taxonomic sampling) have consistently supported cyclostome monophyly[15–17], which implies that hagfishes are secondarily simplified. But the molecular bases of this derived body plan in the light of the tumultuous genomic history of vertebrates are poorly understood.

Early vertebrate evolution was punctuated by multiple polyploidizations, although the nature and timing of these ancient events, and their effects on vertebrate biology, remain elusive[4–6]. An early duplication preceding the gnathostome–cyclostome split ($1R_V$) is generally accepted but has not been clearly resolved by molecular phylogenetics[18]. A gnathostome-specific allo-tetraploidization ($2R_{JV}$) was definitively established on the basis of chromosomal rearrangements observed in gnathostomes but not lampreys[11,19,20], leading to the rejection of the hypothesis that two rounds of genome duplication (1R and 2R) occurred before the cyclostome–gnathostome split[21–23]. Conversely, lampreys experienced additional independent duplication(s) not found in gnathostomes[11,19,20]. A hexaploidy inferred in lampreys was further hypothesized to be ancestral to cyclostomes[20], on the basis of the observation that lampreys and hagfishes both possess six Hox clusters, although their respective orthology remains unclear[24]. Definitive resolution of the duplication history of cyclostome genomes must also account for the disparate karyotypes of hagfishes ($2n ≈ 34$ somatic chromosomes) and lampreys ($2n ≈ 168$) (refs. 7,11,25,26).

Notably, hagfishes[25,27,28] and lampreys[10,11,29] perform programmed elimination of germline-specific chromosomes from the genomes of

[1]Centre for Life's Origins and Evolution, Department of Genetics, Evolution and Environment, University College London, London, UK. [2]Molecular Genetics Unit, Okinawa Institute of Science and Technology Graduate University, Okinawa, Japan. [3]Department of Biology, University of Kentucky, Lexington, KY, USA. [4]Department for Neurosciences and Developmental Biology, University of Vienna, Vienna, Austria. [5]Department of Science, Graduate School of Integrated Science and Technology, Shizuoka University, Shizuoka, Japan. [6]Ocean Research Institute, The University of Tokyo, Tokyo, Japan. [7]Comparative and Medical Genomics Laboratory, Institute of Molecular and Cell Biology, A*STAR, Biopolis, Singapore, Singapore. [8]Department of Molecular and Cell Biology, University of California, Berkeley, Berkeley, CA, USA. [9]Chan Zuckerberg Biohub, San Francisco, CA, USA. [10]Present address: UK Dementia Research Institute, University College London, London, UK. [11]Deceased: Sydney Brenner. ✉e-mail: f.marletaz@ucl.ac.uk; jjsmit3@uky.edu; dsrokhsar@gmail.com

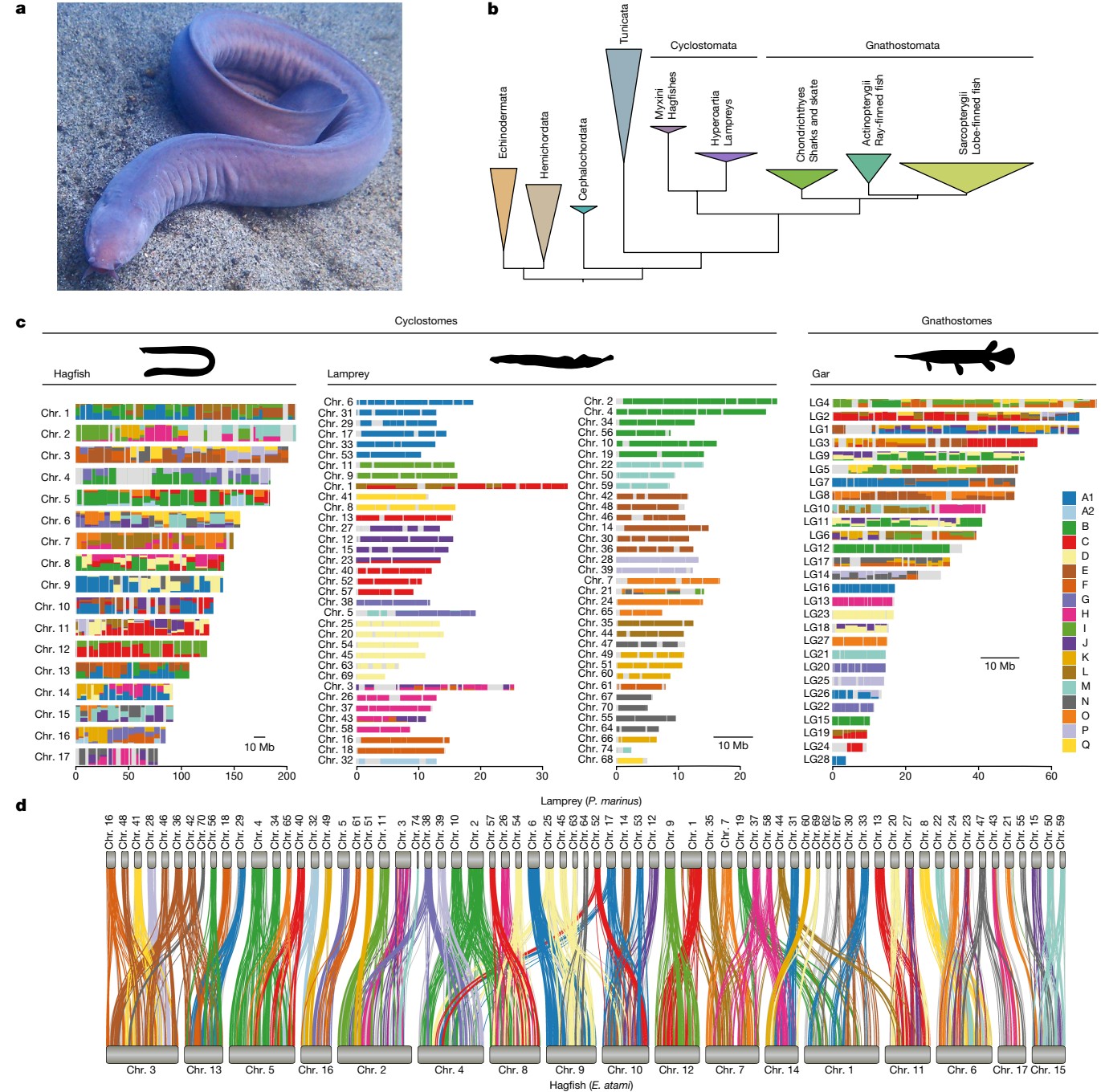

**Fig. 1 | Phylogenetic relationships and syntenic architecture of cyclostomes and gnathostomes. a**, The brown hagfish, *Eptatretus atami* (photo credit, M. Suzuki). **b**, Summary of deuterostome phylogeny based on 176 selected genes (61,939 positions) using a site-heterogeneous model (CAT+GTR). This topology is robust to compositional heterogeneity and similar to what was obtained with 1,467 genes using a site-homogeneous partitioned model (see Methods, Supplementary Note 1 and Extended Data Fig. 2). **c**, Karyograms showing the ancestry of hagfish, lamprey and gar chromosomes in terms of chordate linkage groups (CLGs A1, A2 and B–Q) described previously[19,35] (see also ref. 20 and Supplementary Note 2). Coloured bins contain 20 genes and only genes from CLGs with significant enrichment (Fisher's exact test) are counted (Methods). Hagfish, lamprey and gar silhouettes downloaded from PhyloPic (credit to Gareth Monger for lamprey). **d**, Conserved syntenies show that hagfish chromosomes are typically fusions of multiple lamprey chromosomes. Lines connect orthologous genes and are coloured according to the ancestral chordate linkage groups (colour legend in **c**).

somatic cells[8,9]. In lampreys, germline-specific chromosomes encode numerous genes with putative functions in the maintenance and development of the germline[10,11]. Although hagfishes were the first vertebrate species shown to experience developmentally programmed DNA elimination[25], only germline-enriched satellite repeats have been characterized[30–33]. Because no germline-specific protein-coding genes have been reported in hagfish so far, their possible germline functions,

evolutionary origin and relationship to germline-specific genes in lampreys have not been addressed.

Here we report a chromosome-scale genome assembly for the brown hagfish *E. atami*. Using a synteny-based phylogenetic approach, we definitively resolve and date the timing of duplication and divergence events that shaped the genomes of extant vertebrate lineages, and assess rediploidization after duplication. We further dissect the effect

of these events on the emergence of genes that are involved in vertebrate and hagfish characteristics, and find that hagfish genomes are derived by extensive gene loss, consistent with their morphological simplification. We also find that hagfish genes that are programmatically eliminated during early embryonic development contribute to several aspects of germ cell biology, and reveal the evolution of vertebrate germline-specific chromosomes.

## Evolution of cyclostome genomes

We sequenced the germline genome of the brown hagfish *E. atami* (formerly *Paramyxine atami*) using a combination of long and short reads from testes, and organized the assembly into chromosomes using proximity ligation data from somatic tissue (Supplementary Table 1). Our *E. atami* assembly spans 2.52 Gb and includes 17 large chromosomal scaffolds, consistent with the expected somatic karyotype ($2n = 34$) (Extended Data Fig. 1a and Supplementary Table 2). The length of the assembly is intermediate between the fluorescence-based estimates of genome size for somatic (2.01 Gb) and germline (3.37 Gb) cells[25,28], consistent with $k$-mer estimates (2.02 Gb and 3.28 Gb, respectively, Extended Data Fig. 1b and Methods). The *E. atami* germline genome also includes seven highly repetitive chromosomes that are completely eliminated during development and whose sequences are present in our assembly as sub-chromosomal fragments, similar to what is seen in the highly repetitive germline-specific chromosomes of lampreys[7,11] and songbirds[34]. We annotated 28,469 genes, of which 22,663 show similarity with the protein-coding complement of other species.

We first used our hagfish gene set to test the monophyly of cyclostomes, extending pioneering early studies[16,17] by introducing (i) broader taxonomic sampling of cyclostomes (including new data for the Atlantic hagfish *Myxine glutinosa*) and (ii) improved modelling of site heterogeneity and compositional bias (Methods, Supplementary Note 1 and Supplementary Table 3). A new set of 1,467 orthologues informed by complete hagfish and lamprey genomes alleviates possible paralogy issues, and includes eightfold more markers than earlier studies did (Methods). These analyses confirm the monophyly of cyclostomes with both partitioned analysis (Extended Data Fig. 2a) and site-heterogeneous model analysis (Fig. 1b and Extended Data Fig. 2b). Robustness to compositional heterogeneity is further supported by six-category amino acid recoding validated by posterior predictive tests (Extended Data Fig. 2c,d and Supplementary Table 4).

Despite their disparate karyotypes, the chromosomes of hagfish and lamprey are simply related (Fig. 1c,d) although after around 457 million years of independent evolution, gene order is highly scrambled (Extended Data Fig. 3a,b) and repetitive landscapes are distinct (Extended Data Fig. 1c,d). In general, each hagfish chromosome is typically orthologous to a fusion of between two and six lamprey chromosomes and, conversely, each lamprey chromosome is typically associated with a single hagfish chromosome, with few exceptions (Extended Data Fig. 3b). To differentiate between possible chromosomal fusions in the hagfish lineage and/or fissions (or duplication) in the lamprey lineage, we used the previously reconstructed ancestral chordate linkage groups (CLGs A1, A2 and B-Q)[20,35] (Supplementary Note 2). Although lamprey chromosomes are typically derived from single CLGs (consistent with previous analyses of lamprey genomes[19,20]), hagfish chromosomes are evidently derived from these ancestral elements by irreversible fusions[35], analogous to but distinct from the fusions observed on the gnathostome stem lineage[19] (Fig. 1c). The direct and largely one-to-one segmental correspondence between hagfish and lamprey is consistent with the previous assumption[20,24] that the cyclostomes share the same duplication history, although more detailed phylogenetic analysis is required to rule out scenarios of convergent duplications.

## Genome duplications in early vertebrates

We used two complementary phylogenetic approaches to fully resolve the sequence of early vertebrate polyploidization events: (i) model-based polyploidization inference from a large number of individual gene trees and (ii) concatenation of genes with similar evolutionary histories on the basis of chromosome-scale synteny. We first tested alternative scenarios for the sequence of whole-genome duplications (WGDs) based on 8,931 individual gene trees by probabilistic reconciliation of gene and species trees using WHALE[36] (Methods). This analysis provided significant support for the occurrence of a single genome duplication in the vertebrate stem lineage ($1R_V$), followed by independent polyploidizations on the gnathostome ($2R_{JV}$) and cyclostome ($2R_{CY}$) stem lineages (all Bayes factors $BF_{Null\_vs\_WGD} < 10^{-3}$) (Fig. 2a and Extended Data Fig. 4). By contrast, we found no support for a second round of polyploidization on the vertebrate stem lineage ($2R_V$) or for polyploidization events specific to the lamprey or hagfish lineages (Fig. 2a and Extended Data Fig. 4), consistent with synteny-based analysis[19,20] and Fig. 1d.

We also developed a synteny-based approach that takes advantage of the shared evolutionary history of persistently linked genes to enhance the limited phylogenetic signal of individual gene trees and avoid the confounding effects of differential gene loss[21,37,38]. In this approach, we determined the duplication history of each CLG by concatenating genes from its derivatives in hagfish, lamprey and several jawed vertebrates (for example, chromosomes or chromosomal segments of the same colour in Fig. 1d; Methods). Within each species, the paralogous chromosome segments are called 'paralogons'[39]. Because the CLGs are preserved in diverse invertebrates[35] corresponding sets of chromosomally linked genes can be concatenated to provide outgroups for phylogenetic and molecular dating. We reconstructed paralogon-based molecular phylogenies for 17 of the 18 CLGs or proto-vertebrate chromosomes (PVCs) (Supplementary Table 5); the 18th group (CLG A2 in the notation of ref. 35, and PVC18 in ref. 20) contains a relatively small number of genes consistently linked across vertebrate taxa and has anomalous properties in both lampreys and gnathostomes[20].

Paralogon-based molecular phylogenies support a single early vertebrate auto-tetraploidization ($1R_V$) before the cyclostome–gnathostome split, followed by a later gnathostome-specific allo-tetraploidization ($2R_{JV}$) and a cyclostome-specific polyploidy ($2R_{CY}$) (for example, CLG J in Fig. 2b; Extended Data Fig. 5 and Supplementary Fig. 1). The $1R_V$ duplication node precedes cyclostome–gnathostome speciation in 12 of the 14 CLG paralogon phylogenies with bootstrap support BP > 60 (Supplementary Table 5 and Supplementary Fig. 1). A single shared tetraploidization on the vertebrate stem ($1R_V$) is therefore consistent with both probabilistic inference of genome duplications from single gene trees and paralogon-based phylogenies. Molecular dating indicates that these duplication and speciation events occurred in close succession, and we estimate that the divergence of $1R_V$ paralogons was completed by around 527 million years ago (Ma) and that the cyclostome–gnathostome split occurred around 520 Ma.

Our estimated date for the divergence of $1R_V$ paralogons corresponds to the cessation of homoeologous recombination (rediploidization) rather than the WGD itself, as noted in a previous study[40] We tested for lineage-specific rediploidization across CLGs (relative to the gnathostome–cyclostome divergence) by comparing the likelihoods of gene trees under the ancestral and lineage-specific rediploidization models, as previously proposed[41–45] (Fig. 3a and Extended Data Fig. 6). We found that ancestral rediploidization after $1R_V$ was supported by a larger number of significant gene trees for all CLGs (Fig. 3b), indicating that meiotic rediploidization was essentially complete by the time of the cyclostome–gnathostome split. This contrasts with other more recent vertebrate auto-polyploidizations, in which a number of homoeologous chromosomes have maintained tetrasomic inheritance through subsequent speciation events[41–44].

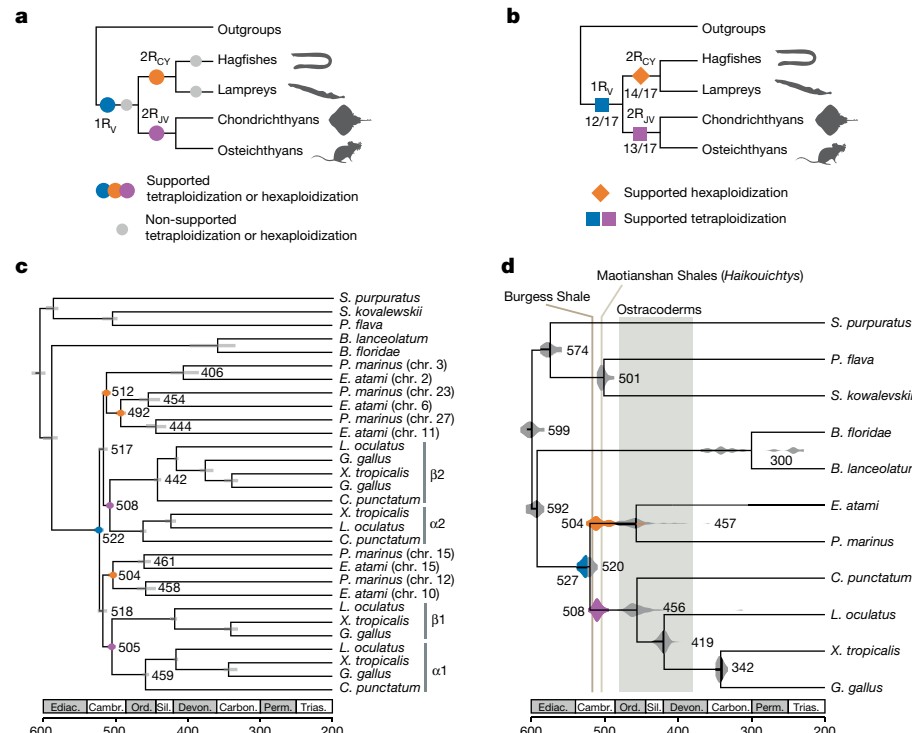

**Fig. 2 | History of genome duplications in vertebrates. a**, Probabilistic inference of polyploidization events in early vertebrate evolution on the basis of gene tree–species tree reconciliation (WHALE[36]; Extended Data Fig. 4, Supplementary Table 8 and Methods) supports an initial tetraploidization shared by all vertebrates (1R_V), a jawed-vertebrate-specific tetraploidization (2R_JV) and a cyclostome-specific polyploidization (2R_CY). Supported polyploidization events (Bayes factors $BF_{Null\_vs\_WGD} < 10^{-3}$) are shown in colour (1R_V, 2R_JV and 2R_CY) and non-supported ones in grey (2R_V, hagfish-specific, lamprey-specific). The WHALE method cannot distinguish between tetraploidization and hexaploidization events. **b**, Paralogon-based polyploidization inference using molecular phylogenies reconstructed for each of the 17 informative CLGs (Supplementary Fig. 1). Successive polyploidization events during vertebrate evolution are shown as coloured polygons and the proportion of CLG trees

displaying these duplication nodes is indicated below. **c**, Sample paralogon tree for CLGJ. As for gene trees, in paralogon trees some nodes correspond to speciation events (grey) and others to duplication events (coloured); both types of events can be dated using a molecular clock. Species and datasets used are listed in Supplementary Table 8, and dating was performed with PhyloBayes (Methods) using fossil calibrations reported in Supplementary Table 7. **d**, Molecular dating of the polyploidizations and speciation events in early vertebrate evolution. Divergence times are indicated for speciation (grey) and duplication nodes (coloured as in **a**) are indicated. In **c**,**d**, each node is labelled with the mean divergence time across CLGs. Ediac., Ediacaran; Cambr., Cambrian; Ord., Ordovician; Sil., Silurian; Devon., Devonian; Carbon., Carboniferous; Perm., Permian; Trias., Triassic.

Unfortunately, molecular phylogenetics can only estimate a later bound on the timing of the 1R_V auto-tetraploidization event itself, because it is likely to be obscured by a period of homoeologous recombination of unknown duration: the 1R_V duplication event could have predated the divergence time of 1R_V ohnologues (around 527 Ma) by millions of years[40].

## Distinct duplications in cyclostomes

Paralogon-based molecular phylogenies also strongly support and refine the 2R_JV allo-tetraploidization scenario[19,20] (Supplementary Table 6). Molecular dating of paralogon trees places the split of the pre-2R_JV alpha and beta progenitors in the middle Cambrian, around 508 Ma (Fig. 2d, Extended Data Fig. 5, Supplementary Table 7 and Methods). The allo-tetraploidization event itself (the hybridization of alpha and beta progenitors and subsequent associated genome doubling), however, occurred some time after the alpha–beta divergence and cannot itself be precisely placed using molecular phylogenies. In recent vertebrate allo-tetraploids such as *Xenopus*[46] and goldfish[47], hybridization occurred within 10–15 million years of the divergence of progenitors. If we take these as analogies for gnathostome allo-tetraploidy, then 2R_JV probably occurred in the late Cambrian, long before the origin of crown-group gnathostomes around 456 Ma near the middle–late Ordovician boundary.

Among cyclostomes, paralogon trees confirm the general orthology of hagfish and lamprey chromosome segments (Fig. 2c), as suggested above (Supplementary Table 5 and Supplementary Fig. 1). We typically observed one or two duplication nodes for each CLG, indicating shared cyclostome genome-wide duplications that took place before the hagfish–lamprey split around 457 Ma. Although the nature of the cyclostome-specific duplications is difficult to decipher owing to extensive losses, the net effect appears to be hexaploidization[20] without the obvious patterns of differential gene retention that are typical of allo-polyploidy and are observed in the gnathostome lineage[19,20] (Extended Data Fig. 7).

The bimodal distribution of divergence times observed between homoeologous (that is, paralogous) cyclostome chromosomes (peaks at around 511 and around 493) Ma; see Extended Data Fig. 5a) is consistent with two-step hexaploidy through the hybridization of diploids and related tetraploids (Extended Data Fig. 5b), as seen in sturgeons[48]. Although the near one-to-one relationship between orthologous hagfish and lamprey chromosome segments (Fig. 1d) suggests that rediploidization after 2R_CY was largely completed by the origination of crown-group cyclostomes, we formally tested for lineage-specific rediploidization (Fig. 3c). From the estimated paralogon divergence times and concatenated paralogon tree topologies, we identified a single case of lineage-specific rediploidization that affected CLGB paralogons in hagfish and lampreys after 2R_CY. Specifically, the paralogon

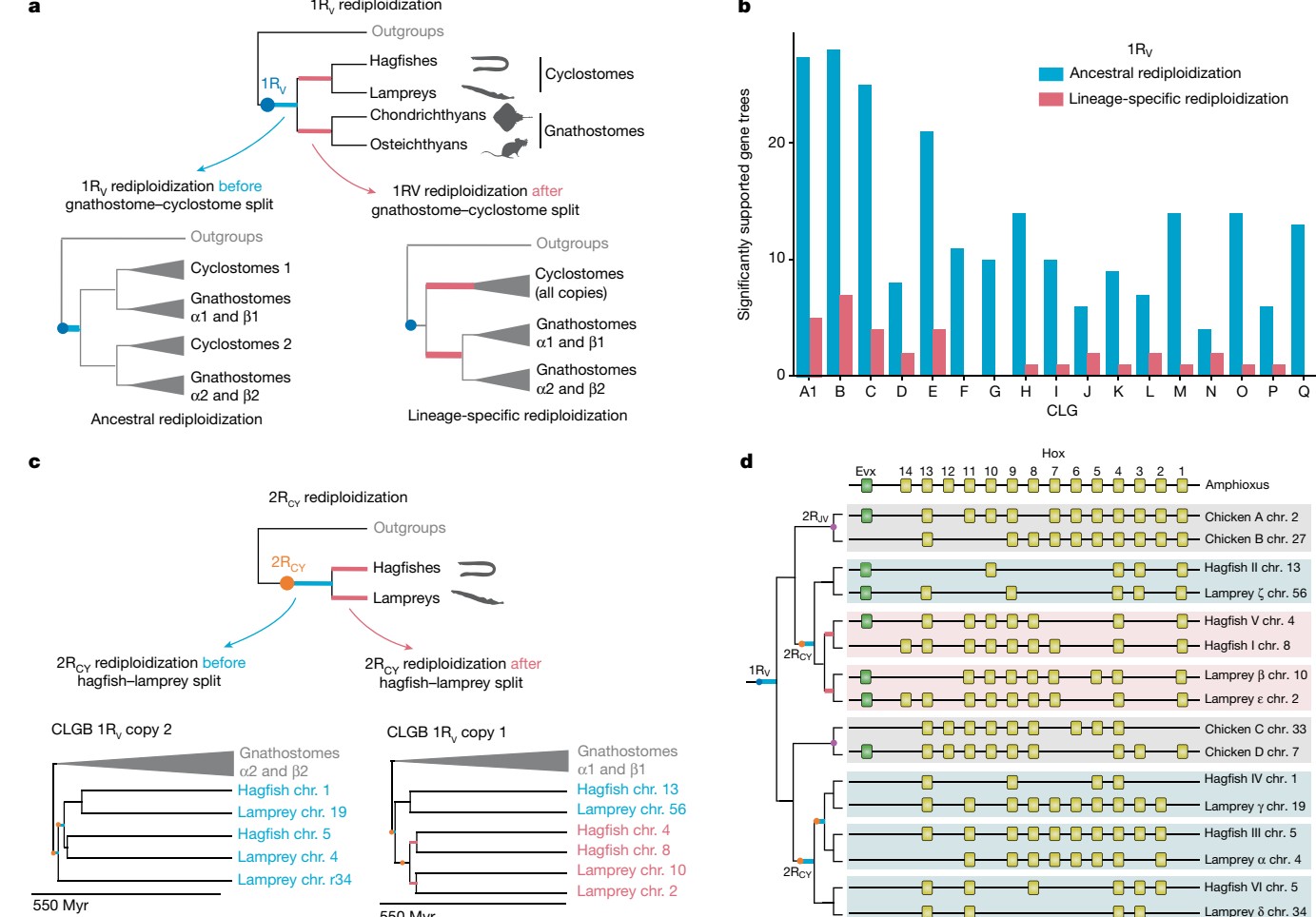

**Fig. 3 | Limited lineage-specific rediploidization after vertebrate genome duplications. a**, Gene-tree topologies expected under the ancestral rediploidization (left) and lineage-specific rediploidization (right) models after the 1R_V (Methods and Extended Data Fig. 6). In the ancestral rediploidization scenario, paralogous gene sequences diverge before the cyclostome–gnathostome split and thus group by duplicated gene copy. In the lineage-specific rediploidization scenario, paralogue sequences diverge independently in the stem gnathostome and cyclostome lineages, and thus genes are grouped by lineage. **b**, Number of significantly supported gene trees in favour of ancestral

and lineage-specific rediploidization scenarios after 1R_V, for each of 17 informative ancestral linkage groups (CLGs). **c**, Tree topologies expected under the ancestral and lineage-specific rediploidization models after 2R_CY. The CLGB paralogon tree shows an ancestral rediploidization topology for 1R_V copy 2, but lineage-specific rediploidization for 1R_V copy 1, where two hagfish (chr. 4 and chr. 5) and two lamprey (chr. 10 and chr. 2) paralogons independently rediploidized. Myr, million years. **d**, Evolutionary history of vertebrate Hox gene clusters resolved by the CLGB paralogon phylogeny (see bottom of **c**).

pairs hagfish chr. 4–chr. 8 and lamprey chr. 10–chr. 12 descending from the 1R_V copy 1 of CLGB each rediploidized independently in hagfish and lampreys, as shown by the CLGB paralogon phylogeny (Fig. 3c) and estimated paralogon divergence times (cyclostome split around 457 Ma; hagfish chr. 4–chr. 8 paralogon divergence around 431 Ma; lamprey chr. 10–chr. 2 paralogon divergence around 442 Ma). This result is confirmed by gene-tree topology tests, albeit with a small number of testable gene trees (Methods; 30 tested trees of which 7 support lineage-specific rediploidization and none support ancestral).

## Evolution of vertebrate Hox clusters

Notably, CLGB contains the Hox cluster, a key locus in early analyses of vertebrate WGD (ref. 40). A more targeted phylogenetic analysis of concatenated Hox and bystander genes also recovered the same lineage-specific rediploidization tree topology as the full CLG paralogon analysis (Extended Data Fig. 8). Through our CLGB paralogon and Hox-plus-bystander trees, we fully resolve the evolutionary history of the four gnathostome and six cyclostome Hox clusters after the 1R_V, 2R_CY and 2R_JV events (Fig. 3d). Although previous studies did not

identify one-to-one relationships between Hox clusters in lampreys and hagfishes[24], we report unambiguous orthologies between lamprey Hox ζ, α, γ and δ and hagfish Hox II, III, IV and VI, with groupings as follows: ζ–II, α–III, γ–IV and δ–VI. By contrast, after lineage-specific rediploidization of CLGB, no true (that is, one-to-one) orthology relationships exist between lamprey Hox β and ε and hagfish Hox V and I, which should be considered as 'tetralogues'[49].

## Origin of neural crest

Our paralogon-based classification makes it possible to robustly assign paralogues to specific duplications, and this sheds some light on the relative origin, with respect to WGDs, of vertebrate characteristics such as neural crest, placodes and hormone systems[50]. As highlighted previously[51], establishing whether both paralogous branches retain (or partition) the ancestral role can help to pinpoint whether a character is likely to have emerged before or after 1R_V.

To assess the origin of neural crest, we considered a set of 22 gene families involved in the specification and migration of neural crest[52,53] (Fig. 4a and Supplementary Table 8) and we asked whether

corresponding 1R$_V$ paralogues perform neural crest cell (NCC)-related functions in gnathostomes and in lampreys on the basis of the literature[54–56] and available RNA sequencing (RNA-seq) data[56]. We find that for many of these gene families, including *Tfap2*, *SoxE*, *EdnR*, *Twist1* and *Gata3*, paralogues on both 1R$_V$ branches are involved in neural-crest-related functions (Supplementary Table 9). This pattern indicates that NCC-related functions were inherited from pre-1R$_V$ genes in the vertebrate ancestor, and thus suggests that the neural crest originated before 1R$_V$. Post-WGD subfunctionalization had a limited role in its emergence, contrary to other gnathostome novelties such as limbs[57]. Consistently, in lampreys, an alternative 1R$_V$ paralogue also seems to be involved in NCCs, involving, for example, *Gata3*, *Six1*, *Msx1* and possibly *FoxD* (Fig. 4a).

By contrast, the establishment of the trunk and cranial NCCs seems to differ among cyclostomes, osteichthyans and even amniotes, with distinct genes being involved[56]. Some but not all of the genes involved in this process seem to show a more recent occurrence of subfunctionalization. For instance, *Lhx5*, *Id3*, *Gid2* and *Dmbx*, which have a role in gnathostome cranial NCCs, do not have 1R$_V$ or 2R$_{JV}$ paralogues with a similar function, whereas *Tfap2* and *SoxE*, which are involved in the ancestral specification of both cranial and trunk NCCs, have paralogues on both of the 1R$_V$ branches that are involved in this function. Lampreys show marked differences: neither *RhoB* nor *Ets* are involved or expressed in lamprey migratory NCCs, and *Lhx5*, *Dmbx* and *Ets1* are expressed in later NCC derivatives (Extended Data Fig. 9c and Supplementary Table 10). Despite the extensive gene loss experienced in the hagfish lineage (see below), we recovered homologues for most of the NCC-related genes that we investigated. Further functional studies will be necessary to determine whether subsequent 2R$_{CY}$ paralogues in hagfish were incorporated in NCC-related functions specific to this lineage[52].

## A distinct fate for paralogues

Paralogues retained in gnathostomes after two rounds of genome duplication were previously shown to be functionally associated with the regulation of development and nervous system activity[18]. To determine whether similar genes were retained preferentially in multiple copies in the cyclostome lineage after 1R$_V$ and/or after cyclostome-specific 2R$_{CY}$, we tested paralogue sets that show distinct retention patterns for functional enrichment (Fig. 4b). We recovered gene ontology terms that were previously found to be enriched in gnathostome paralogues (for example, axon guidance and embryonic organ development), but, notably, we found that they were preferentially associated with paralogues retained after the pan-vertebrate 1R$_V$ regardless of their post-gnathostome duplication 2R$_{JV}$ fate (Fig. 4b), suggesting that 1R$_V$ had a key role in the early elaboration of the vertebrate nervous system. In cyclostomes, however, these terms are preferentially associated with paralogues that were systematically retained after all polyploidizations (1R$_V$ and 2R$_{CY}$); this suggests a distinct path of paralogue evolution at the functional level, possibly coupled with an increased retention after 2R$_{CY}$ compared with 2R$_{JV}$.

The fate of paralogues after WGD is often related to their acquisition of more specific expression domains that can explain subfunctionalization and functional innovation[58–60]. To examine patterns of divergence in gene expression in gnathostomes and cyclostomes, we compared paralogues across a consistent set of six organs in amphioxus, gar, lampreys and hagfish. Considering 3,009 gene families, we found a higher level of gene-expression specificity in gar than in lampreys and hagfish, with the hagfish showing the least specificity (Extended Data Fig. 10b). We then counted the number of expression patterns that were gained or lost in the same gene family between amphioxus and the different vertebrate species, which also indicated a lower level of subfunctionalization in cyclostomes than in gnathostomes (Fig. 4c). Finally, we asked whether particular organs show a significant enrichment of

paralogous genes using gene-expression clustering (weighted gene co-expression network analysis (WGCNA) (Methods and Extended Data Fig. 9a,b). Of note, we found that only neural tissue exhibits enrichment in both gnathostomes (for example, gar) and hagfish, whereas many recently duplicated genes are expressed in an organ-specific manner (Extended Data Fig. 9b). Together, these results imply that cyclostomes—and, to a greater extent, hagfish—show more limited subfunctionalization or specialization of expression patterns than do gnathostomes.

## Gene loss and hagfish novelties

Hagfish underwent the most extensive gene loss among vertebrates, with 1,386 missing gene families, of which 892 were present in the deuterostome ancestor (Fig. 4d and Extended Data Fig. 9d). Hagfishes stand out as having lost all members of several entire gene families, rather than exhibiting just an increased loss of paralogues (Extended Data Fig. 10c).

Several gene families lost in hagfish are functionally enriched for roles associated with missing characters in hagfish (Fig. 4e). For instance, γ-crystallins, which make up the lenses of vertebrate eyes, are absent in hagfish (but independently expanded in lampreys and gnathostomes), as are the *EYS* (eyes shut homologue) and *RBP3* (retinol-binding protein 3) genes that are involved in photoreceptor maintenance and development[61] (Supplementary Table 11). Several genes that are involved in bone development and its hormonal control in other vertebrates[62] are absent in hagfish: two members of the RANK–osteoprotegerin pathway that control osteoclast proliferation in gnathostomes[63], as well as the genes encoding the parathyroid hormones (*PTH* and *PTLH*), which have a role in the regulation of calcium metabolism (their receptor is still present)[64]. These genes are present in lampreys and their loss in the hagfish lineage could be associated with the limited condensation of the hagfish vertebral cartilage.

Hagfish have also gained new traits, most notably their prodigious ability to secrete a highly viscous slime that acts as a defence against predators. We found two clusters of genes that are specifically and highly expressed in the slime gland (Extended Data Fig. 9e) and are related to intermediate filaments (α-keratin)[65]. One of these clusters contains a gene that is expressed mainly in the skin but not in the slime gland, consistent with the recent suggestion that the keratin threads of hagfish slime could have originated as elements of the skin[66] (Fig. 4f). We found that the most highly expressed glycoproteins in the slime gland included von Willebrand A and D domains, rather than mucin-type domains as previously hypothesized[67].

## Programmed DNA and gene elimination

The somatic and germline cells of hagfish exhibit distinct karyotypes, owing apparently to the loss of germline-specific chromosomes through embryonically programmed DNA elimination. On the basis of *k*-mer counts (Extended Data Fig. 1b), we estimate that around 1.3 Gb is lost from the approximately 3.3-Gb germline genome of *E. atami*, consistent with cytofluorometry[25,28]. Analysis of the genome assembly identified a large number of germline-specific genes and confirmed that germline-specific regions contain large numbers of complex repetitive elements[30–33], including one newly identified repeat that accounts for 4% of the genome (Fig. 5, Extended Data Figs. 11 and 12 and Supplementary Note 3).

So far, no germline-specific genes have been identified in any hagfish species. We identified germline-specific genes in *E. atami* by comparing the read depth of germline and somatic reads across low- to medium-copy regions of the assembled genome (Methods and Fig. 5d). We discovered 81 Mb of germline-specific sequence that encode 1,654 genes, 226 of which have identifiable human homologues (to 121 non-redundant human genes) (Supplementary Table 12). We confirmed

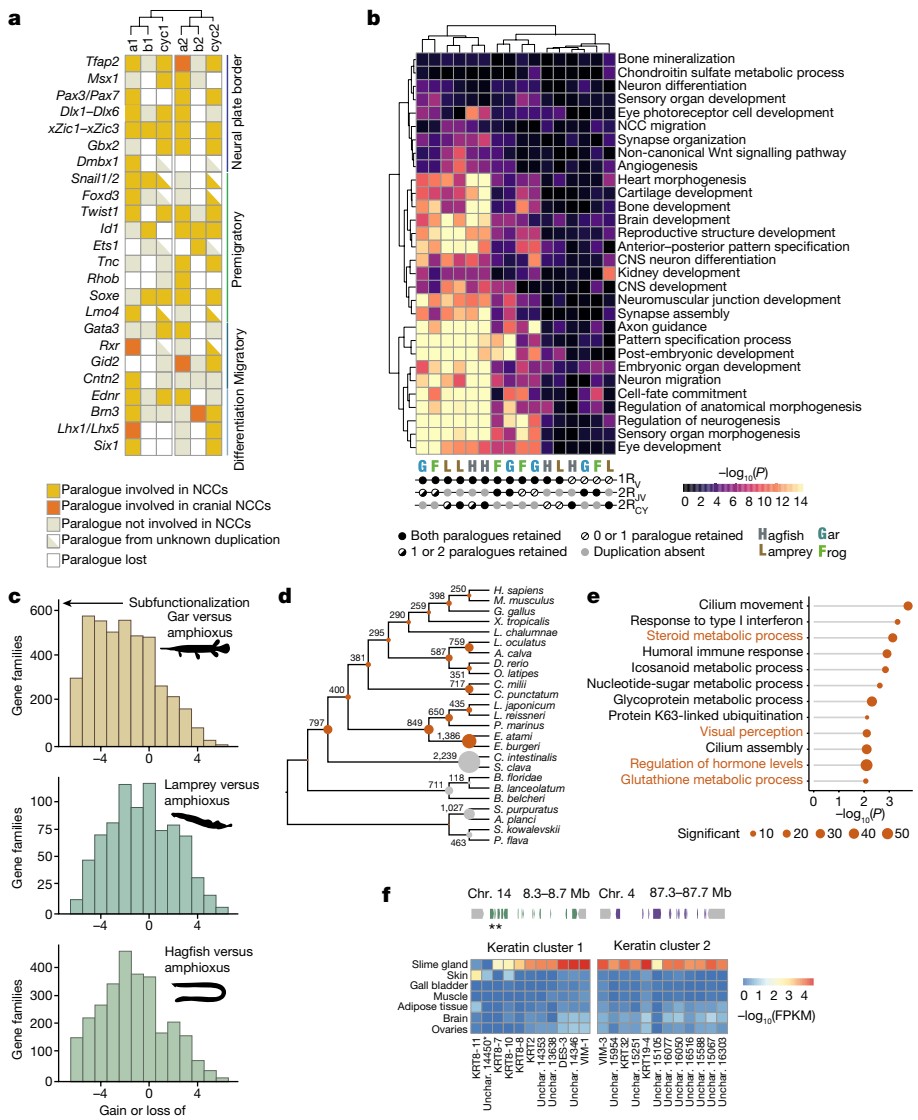

**Fig. 4 | Functional effects of vertebrate WGD and gene loss in vertebrates.**
**a**, Key neural-crest-related gene families with members classified according to their functional role (colour) and paralogy status relative to $1R_v$ and $2R_{JV}$. The involvement of paralogues derived from both copies of the $1R_v$ in NCC-related function, in both gnathostomes and lampreys, supports the hypothesis that NCCs predate $1R_v$. **b**, Enrichment of functional annotation terms (gene ontology) in sets of genes showing a specific pattern of retention after vertebrate WGDs. Each column corresponds to a set of paralogous genes with a specific pattern of post-duplication retention in a given species. We distinguished cases in which both paralogues can be assigned to a specific duplication and are retained, cases in which at least one of the paralogues is retained and cases in which at least one of the two copies is lost. CNS, central nervous system. **c**, Distribution of the difference of positive organ-specific expression domains between selected vertebrate species and the amphioxus outgroup for ohnologue gene families[59]. A shift to the left in the distribution (as seen for the gar) indicates an extensive subfunctionalization through the restriction of gene-expression domains in vertebrates. **d**, Gene-family loss in deuterostomes, highlighting the severe loss in the hagfish lineage relative to that seen in other vertebrates and deuterostomes (grey). Species abbreviations are provided in Supplementary Table 8. **e**, Functional enrichment (gene ontology) for gene families lost in the hagfish lineages, highlighting a simplification of visual and hormonal systems (labels in orange). **f**, Structure of the two clusters of α-keratin genes on chromosomes 14 and 4, and their expression in the slime gland and the skin shown as a heat map (gene expression expressed as fragments per kilobase per million reads (FPKM)). Unchar is the prefix used for naming genes that did not receive a gene name by homology search. Genes are shown in the same order in the heat map as they are located in the two clusters. Stars indicate the two genes that are expressed preferentially in the skin (Extended Data Fig. 10).

that 44 of 46 tested germline-specific intervals can be PCR-amplified from testes but not blood DNA (95.7% validation rate) (Supplementary Table 13). Germline-specific genes in hagfish are enriched in several biological functions on the basis of gene ontology analyses, including functions related to cell cycle, cell motility and chromatin or DNA repair (Fig. 5e and Supplementary Tables 14 and 15). Similar functions were also enriched among germline-specific genes in sea lamprey, and support the hypothesis that somatically eliminated genes generally perform functions that benefit the development and maintenance of the germline[11].

The broad functional similarity between germline-specific genes of hagfish and lamprey suggests that DNA elimination could be a shared ancestral feature of the cyclostome lineage[8]. To attempt to identify the vestiges of theoretical ancestral germline-specific chromosomes in the cyclostome lineage, we examined patterns of orthology and paralogy for eliminated genes. Despite the general functional similarity of eliminated genes in hagfish and sea lampreys, few orthologous genes were found to be eliminated in both genomes. In total, 7 of 121 non-redundant hagfish gene families were also eliminated in sea lampreys (*CDH1*, *CDH2* and *CDH4*; *GJC1*; *MSH4*; *NCAM1*; *SEMA4B* and *SEMA4C*; *WNT5A*, *WNT5B*,

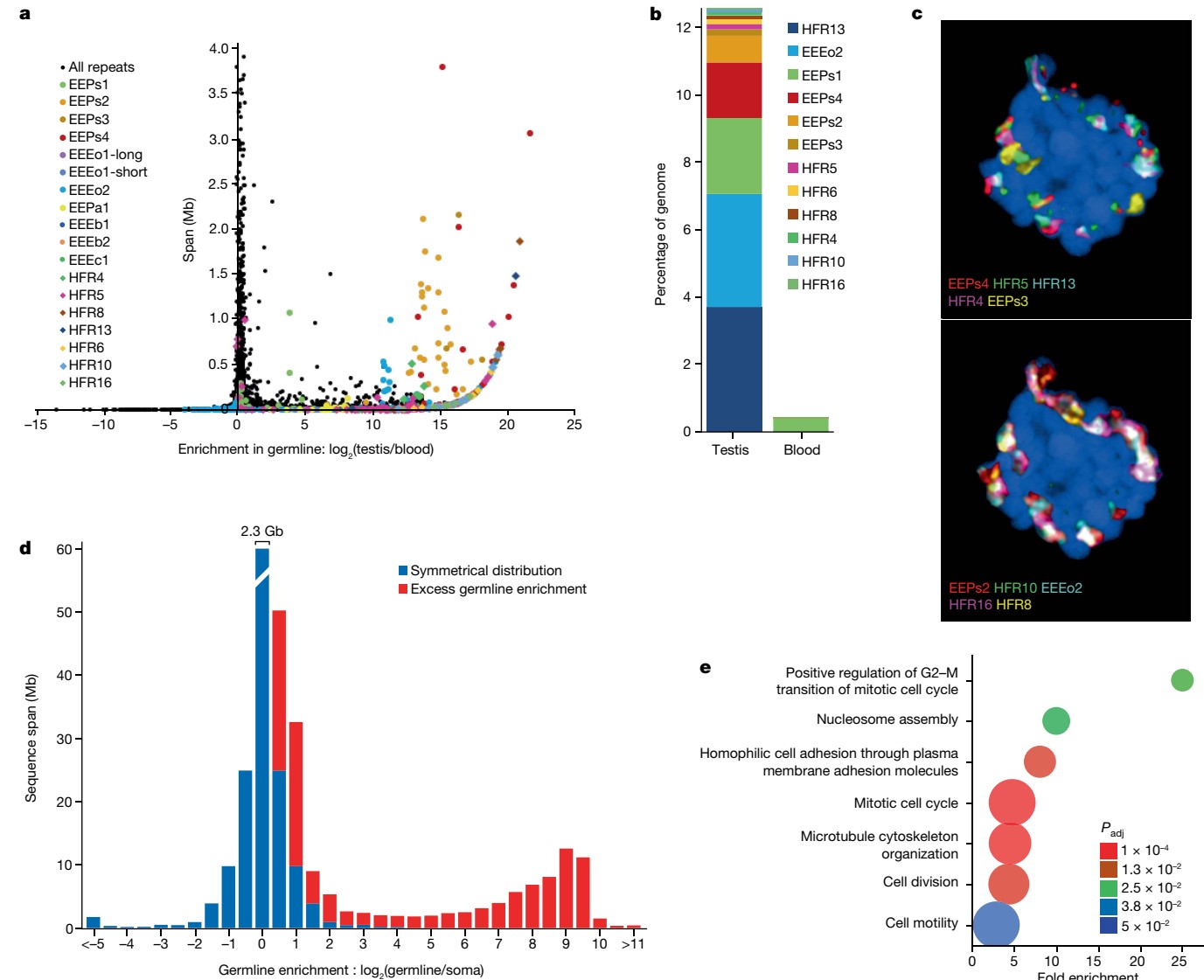

**Fig. 5 | Germline-specific and enriched sequences and genes in hagfish.**
**a**, Plot showing the degree of germline enrichment and estimated span of all predicted repetitive elements in the *E. atami* genome, focusing on elements with a cumulative span of less than 4 Mb (per family member). Previously identified elements[30,33] are highlighted by coloured circles and newly identified high-copy elements are highlighted by coloured diamonds. Additional higher copy repeats are visible in Extended Data Fig. 12m,n. The colouring scheme is the same in **b** and in Extended Data Fig. 12m,n. **b**, Estimated cumulative span of the eight most highly abundant repeats shown as the percentage of the genome covered. **c**, Fluorescence in situ hybridization (FISH) of high-copy germline-specific repeats to a testes metaphase plate showing their distinct spatial clustering within chromosomes (blue counterstaining is NucBlue: Hoechst 33342; individual pairs of probes are shown in Extended Data Fig. 12m,n). **d**, Comparison of the sequence depth of DNA extracted from germline (testes) versus somatic (blood) tissues identifies a large number of genomic intervals with evidence for strong enrichment in the germline. The bin representing no enrichment contains a total of 2.3 Gb of the assembly. **e**, Genes encoded within germline-specific regions are enriched for several ontology terms related to regulation of cell cycle and cell motility (Panther Biological Processes: most specific subclass shown; Supplementary Table 14).

*WNT7A* and *WNT7B*; and *YTHDC2*; Extended Data Fig. 11). An analysis of gene trees indicates that three of these (orthologues of *MSH4*, *WNT7A* and *YTHDC2*; Extended Data Fig. 11) share a last common ancestor that can be traced to a single lineage after the basal vertebrate divergence and duplication events. This small set of genes might reflect the vestiges of shared germline-specific sequences that were eliminated early in the cyclostome lineage, or, alternatively, these genes might have been independently recruited to the germline-specific fraction during the early evolution of both lineages.

Germline-specific chromosomes in songbirds and lampreys are continuously capturing duplicates of somatic genes, establishing new germline-specific genes that often evolve rapidly owing to their unique selective and mutational genomic environment[7,34]. In *E. atami*,

we observe several germline-specific genes that have undergone extra rounds of duplication after duplicating or translocating to germline chromosomes. The genes with the highest germline-specific copy numbers are homologues of *FBXL4*, a modulator of E3 ubiquitin ligase that regulates the proteasomal turnover of the histone demethylase KDM4A (ref. 68) (25 copies); and *TRRAP*, a component of several histone acetyltransferase complexes (18 copies) (Extended Data Fig. 11 and Supplementary Table 12). The *FBXL4* orthogroup also contains 45 paralogues in the draft genome of the closely related hagfish *Eptatretus burgeri* (ref. 69), indicating that the origin of germline-specific *FBXL4* and the expansion of this gene family predates the split between the two hagfish species, with additional lineage-specific expansions and losses underlying differences in paralogue numbers over the past few

million years. These gene families seem to have undergone substantial expansion even in the recent past, emphasizing their high rate of turnover.

The accumulation of epigenetic silencing marks and regulated degradation has been implicated in the cellular mechanisms that underlie the elimination of lamprey germline-specific chromosomes[8,70]. This suggests that some components of hagfish DNA elimination mechanisms might be encoded by the germline-specific chromosomes themselves, or contribute to other aspects of hagfish germ cell development. Other genes involved in the same pathways as *FBXL4* and *TRRAP* have also duplicated in the context of the *E. atami* germline-specific chromosomes, albeit to a lesser extent. These include *KLHL10*, a component of the E3 ubiquitin ligase complex involved in spermatogenesis (4 copies); *SIN3*, a transcriptional repressor whose human homologue is highly expressed in the testis (4 copies); and *DNMT1*, the primary enzyme responsible for maintaining silencing DNA methylation marks after DNA replication (2 copies). Notably, each of these five families of germline-specific genes also possesses at least one somatically retained paralogue, indicating that germline-specific expansion of gene families related to ubiquitination and regulation of chromatin state has evolved in the context of largely intact ancestral somatic pathways.

## Conclusion

Early vertebrate evolution was accompanied by a series of ancient polyploidization events that have been difficult to unambiguously resolve using conventional sequence-based molecular phylogenetics. Challenges include the antiquity of these events and the relatively short intervals between them[18,21], as well as lineage-specific evolution and gene loss after duplication[21,37,38]. We used the hagfish genome and an approach focused on chromosome-scale phylogenetics to fully resolve this history of ancient vertebrate polyploidies (Fig. 2 and Extended Data Fig. 5). The earliest duplication, $1R_V$, occurred on the vertebrate stem lineage in the early Cambrian (around 527 Ma), around 10 million years before the appearance of *Haikouichthys* and *Myllokunmingia* (ref. 3) the earliest vertebrate fossils. Whether the similarity in timing is coincidental or causal remains to be seen.

After the shared duplication, cyclostomes and gnathostomes experienced independent polyploidizations during the late Cambrian–early Ordovician, coinciding with a gap in the vertebrate fossil record. We can, however, begin to relate early genomic events to the emergence and elaboration of vertebrate innovations by correlating the contemporary functions of gene duplicates with their appearance at specific duplication events. For example, we find that when one gene functions in neural crest, its $1R_V$ paralogues also do, as expected if the neural crest regulatory circuits already existed before $1R_V$. More speculatively, we note that Evx homeobox genes, which have a role in the development and patterning of paired fins and limbs, duplicated at $1R_V$, with both lineages being retained in gnathostomes (Evx1–HoxA and Evx2–HoxD), but that cyclostomes are missing HoxC and HoxD-associated Evx paralogues owing to lineage-specific loss. This observation suggests that $1R_V$ duplicates might have acquired roles in fin bud development and patterning very early in the evolution of the gnathostome lineage, consistent with the observation of paired fin fold morphologies in early diverging galeaspids[71].

Finally, analysis of *E. atami* germline-specific chromosomes in comparison with other vertebrates supports the hypothesis that these chromosomes encode functions that are advantageous for the development of germ cells and the production of gametes, and indicates that rapid turnover of germline-specific gene content might be a common feature across highly divergent lineages. As with other features of their biology, differences in the gene content of lamprey and hagfish germline-specific chromosomes might reflect their long history of independent evolution and the marked differences in their reproductive, ecological and developmental biology that have accumulated over

the approximately 460 million years since the last common cyclostome ancestor.

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

## Methods

### Genome sequencing and assembly

DNA was extracted from a testis from a male *E. atami* individual and extracted using proteinase K digestion and phenol:chloroform extraction[72]. Animals were sampled in Suruga Bay, off Yaizu (300–330-m depth) and maintained in seawater aquariums at 11–13 °C. In agreement with procedures authorized by the Guidelines for Proper Conduct of Animal Experiments by the Science Council of Japan (2006), animals were anaesthetized using Tricaine (MS222, Sigma) before euthanasia and dissection. Paired-end and mate-pairs Illumina libraries were generated using Illumina Truseq and Nextera Mate-pair kits and sequenced on HiSeq2000 and HiSeq2500 instruments (Supplementary Table 1). The Illumina dataset was assembled using Meraculous (v.2.2.2.5) with a *k*-mer of 71 and 'diploid mode' set to '1' to attempt the merging haplotypes[73], and subsequently scaffolded using mate-pairs information (Supplementary Table 2). PacBio long-reads data at around 35× coverage were generated on a PacBio RSII instrument (Supplementary Table 2) and incorporated using PBJelly (v.15.8.24)[74]. PBJelly aligns the PacBio reads to the assembly using the Blasr aligner and collects reads surrounding and spanning gaps. Sequences assembled from these spanning reads are used to fill gaps and extend scaffolds. We used the parameters '-minMatch 8 -sdpTupleSize 8 -minPctIdentity 75 -bestn 1 -nCandidates 10 -maxScore -500' for Blasr alignment.

The gap-filled assembly was further scaffolded using proximity ligation information. We used both Chicago libraries relying on syntenic reconstructed chromatin and Hi-C libraries capturing the native chromatin contacts, and scaffolding was performed using the HiRise package[75]. Hagfish liver was cross-linked in 1% paraformaldehyde, and chromatin was subsequently extracted, immobilized on SPRI beads, washed and digested with DpnII (ref. 76). After end-labelling, proximity ligation was performed using T4 DNA ligase and cross-linking was reversed using proteinase K. The DNA fragments were removed from the beads and then purified again on SPRI beads. The sequencing library was constructed using the NEB Ultra Library Preparation Kit (New England Biolabs).

The genome-wide heterozygosity was estimated to be 0.9%. The final BUSCO score (Metazoa) is C:90.0% (S:89.8%, D:0.2%), F:4.0%, M:6.0%, n:954. The size of the hagfish genome was estimated by counting 21-mers with Meryl (v.1.1)[77]. Using a fitting four-peak model as implemented in Genomescope2, the estimated size is 2.02 Gb and 3.28 Gb using sequencing data from blood and testis DNA, respectively[78] (Extended Data Fig. 1b).

### Transcriptome and genome annotation

We generated RNA-seq data for 13 organs with an average depth of 25 million reads. We aligned the reads to the genome using STAR (v.2.5.2b) with an average 78.7% uniquely mapping reads[79]. These alignments were used to assemble transcriptomes for each organ using StringTie (v.1.3.3b) and subsequently merged together using Taco[80]. In parallel, a de novo assembly of the bulk RNA-seq data was performed using Trinity (v.2.11.0) both in reference-free and genome-guided mode[81].

We also sequenced full-length cDNA from brain RNA on eight cells of Pacbio RSII. Following the Iso-Seq protocol, circular consensuses of subreads were calculated and validated as full length on the basis of the presence of SMART adaptors at both extremities. Full-length transcripts were clustered and polished using all circular consensus reads with Quiver (v.2.0.0), yielding 23,343 high-quality transcripts.

Assembled transcripts from de novo and genome-guided Trinity and high-quality Iso-Seq transcripts were aligned to the genome using GMAP (v.2018-03-25). Mikado (v.1.2.1) was used to generate a high-quality reference transcriptome leveraging (i) the aligned Trinity de novo and genome-based transcriptomes; (ii) the Iso-Seq transcripts; (iii) the StringTie transcriptomes merged with Taco and a set of curated splice junctions generated from RNA-seq alignments using Portcullis

(v.1.0.2). Putative fusion transcripts were detected by Blast comparison against Swiss-Prot and ORFs were annotated using TransDecoder[82]. Transcripts derived from the reference transcriptome were selected to train the Augustus de novo gene prediction tool[83]. Intron positions and exon positions were converted into hints for Augustus gene prediction.

Finally, we constructed a database of repetitive elements using RepeatModeler (v.1.0.11) and used it for masking repetitive sequences with RepeatMasker (v.4.0.7). Gene models with half or more of their exons showing 50% overlap with repeats were discarded, yielding 46,822 filtered gene models. Alternative transcripts and UTRs were subsequently incorporated using the PASA pipeline[82]. These gene models contain a total number of 4,915 distinct PFAM domains.

### Phylogenomics and molecular dating

To obtain sequences from a previously unsampled hagfish group, we extracted RNA from *M. glutinosa* liver preserved in RNAlater using the RNAeasy kit (Qiagen). The RNA-seq library was constructed using the NEBNext Ultra II Directional RNA Library Prep Kit for Illumina (NEB) and sequenced on a Novaseq6000 instrument (SRR). The transcriptome was assembled using Trinity (v.2.11.0)[81], enabling read trimming, and was translated using TransDecoder (v.5.5.0)[82].

We inferred a set of 1,467 single-copy orthologues suitable for phylogenetic reconstruction by applying the OMA tool (v.2.4.1)[84] to a subset of deuterostome proteomes including lamprey and the newly generated hagfish gene models (Supplementary Table 8). Selected transcriptomes were assembled using Trinity (v.2.11.0) and translated using TransDecoder (v.5.5.0)[82]. We built hidden Markov model (HMM) profiles using Hmmer (v.3.1b2) for each orthologue family and extracted orthologues for phylogenetic reconstruction using a previously described approach[85]. Subsequent sequences were aligned using Msaprobs[86], mistranslated stretches were filtered out using HmmCleaner[87] and diverging regions intractable for phylogenetic analysis were removed using BMGE (-g 0.9)[88]. Phylogenetic trees were reconstructed for each alignment using IQ-TREE (v.2.1.1) with a LGX+R model[89]. For computational intensive analyses, such as site-heterogenous reconstruction with CAT+GTR, we selected the 20% orthologues with the lowest saturation. Molecular dating analysis was conducted using PhyloBayes (v.4.1e)[90] using the CAT+GTR+G4 model and the CIR relaxed clock (with soft-bound) assuming fossil calibrations[2,91,92] (Supplementary Table 7).

### Synteny reconstruction

Pairs of orthologous genes were obtained by mutual best hit after reciprocal proteome comparison using MMSeqs2 (r12-113e3), and were used to create a system of joint coordinates to plot orthologue position in two species. Fisher's exact test was used to determine mutual enrichment of orthologues between chromosomes, and only significant enrichments were incorporated in binned orthologous content representations (Fig. 1c). Plots connecting orthologues in multiple species (Fig. 1d) were generated using Rideogram (v.0.2.2).

### Gene-family analyses and phylogenetic analysis of paralogons

We reconstructed gene families using Broccoli (ref. 93) for a set of genomes from deuterostome species (Supplementary Table 6). For gene families that included at least 6 genes and 3 species but fewer than 450 sequences in total, we applied GeneRax to infer the losses and duplications that affected a given gene family[94]. To do that, we generated individual alignments using MAFFT (v.7.305)[95], filtered them using BMGE and reconstructed a tree using IQ-TREE and an LG+R model[89]. These curated alignments and trees were used as input for GeneRax (v.1.2.2) assuming a D+L (duplication plus loss model). Reconciled trees in the RecPhyloXML format were parsed to estimate the duplications and lineage-specific losses at each node of the species tree[96] as seen in Extended Data Fig. 10c. Reconciled trees were split if they showed a duplication at the 'deuterostomia' node indicative of a deep paralogy relationship.

For each gene family, we first assigned the CLG by considering the location of amphioxus and sea urchin genes and the corresponding CLG-to-chromosome assignment, and then evaluated the occurrence of the paralogues derived from the 1R and $2R_{jv}$ in gnathostomes on the basis of the vertebrate classification that was previously established[19] and has been revised in this study (Supplementary Table 6). Selected species for gene families including derivatives of the 1R paralogons and at least three out of four possible paralogons for gnathostomes (α1, α2, β1, β1) were collected (Supplementary Table 5). These genes were concatenated for each CLG on the basis of their paralogon identity in gnathostomes, and the chromosomal identity of the CLG derivatives in cyclostomes. Two datasets were generated: a 'strict' one, in which at least three distinct gnathostome paralogons were required for each retained gene family; and a 'relaxed' one, in which only two or more gnathostome paralogons were required (Supplementary Table 5). A similar approach was used to classify individual genes depending on the duplication events from which they derive. We collected gene ontology terms and functional classification information by applying eggNOG (ref. 97) on the proteome of our interest species and term enrichment analysis conducted using the TopGO package (v.2.50.0).

For analyses of gain and loss, we used gene-family reconstruction that incorporated the gene models of the related hagfish *E. burgeri*[69] to assess recent gene-family expansions or contractions in the hagfish lineage. Gene functions were assigned using the PANTHER classification[98].

## Tests of WGD hypotheses on the vertebrate phylogeny

We used WHALE (v.2.1.0)[36] to rigorously test WGD hypotheses on a reduced vertebrate species tree (Fig. 2a and Extended Data Fig. 4). We leveraged a total of 8,931 gene families in this analysis, selected to contain at least one gene copy in each clade from the root, in compliance with the assumption of WHALE that genes were acquired in a common ancestor of all included species. We further filtered large families to reduce the computational burden. For each of the 8,931 retained families, we built a multiple sequence alignment based on the amino acid sequences with MAFFT (v.7.508)[95] and reconstructed 1,000 bootstrap trees with IQ-TREE (v.2.2.0.3)[89] under the LG+G model. We summarized clade conditional distribution (CCD) from bootstrapped trees using the ALEobserve tool from the ALE software[99]. We ran WHALE on the dated species trees and CCD data to test five different WGD hypotheses on the vertebrate species tree: $1R_V$, $2R_{JV}$, $2R_{CY}$, a hagfish-specific duplication and a lamprey-specific duplication (Extended Data Fig. 4a). We used the variable rate DLWGD WHALE model, which models independent duplication and loss rates across branches. We assumed a normal distribution $N(\log(0.15), 2)$ on the mean log-scaled duplication and loss rate, an exponential distribution (mean = 0.1) prior on its variance, a Beta (3, 1) hyper prior on the η parameter (distribution of the number of genes at the root) and uniform priors on the retention parameters (q parameter) for all WGDs. We obtained significant Bayes factors ($BF_{Null\_vs\_WGD} < 10^{-3}$) in support of large-scale duplication (post-duplication retention parameter $q \neq 0$) for the $1R_V$, $2R_{JV}$ and $2R_{CY}$ events (Extended Data Fig. 4b). These results were reproduced using the simpler constant rate DLWGD model. We similarly tested an alternative scenario with two duplications on the vertebrate stem ($1R_V$ and $2R_V$; Extended Data Fig. 4c,d). In this configuration, and using uniform priors on the retention parameters, we observed that WHALE could not distinguish retention parameters for $1R_V$ and $2R_V$: this is revealed by the bimodality of the estimated posterior distribution for each of these two parameters. We found that using distinct priors on retention parameters allows the estimation of distinct retention parameters for $1R_V$ and $2R_V$ and shows support for a single $1R_V$ event (Extended Data Fig. 4e). This investigation of alternative priors was conducted on a pilot run of 1,000 randomly selected gene families, to alleviate computational time (1,000 families were previously suggested to be sufficient for parameter estimation[36]).

## Ancestral and lineage-specific meiotic rediploidization

We selected a set of 1,247 gene families, including genes of 6 vertebrate species (bamboo shark *Chiloscyllium plagiosum*, spotted gar *Lepisosteus oculatus*, chicken *Gallus gallus*, western clawed frog *Xenopus tropicalis*, brown hagfish *E. atami* and sea lamprey *Petromyzon marinus*) and the closest outgroup (depending on taxonomic availability), to test for ancestral and lineage-specific rediploidization after the 1R genome duplication. These 1,247 families were selected so as to result in distinct tree topologies under the ancestral and lineage-specific rediploidization models, on the basis of the following criteria: (i) at least one gnathostome species has retained both 1R_1 (that is, alpha1 and/or beta1) and 1R_2 (that is, alpha2 and/or beta2) gene copies; (ii) at least one hagfish gene and one lamprey gene; (iii) at least one non-vertebrate outgroup gene; and (iv) a non-prohibitive number of hagfish and lamprey genes so that a maximum of 10 possible ancestral rediploidization topologies can be derived for the family (Extended Data Fig. 6). For each gene family, we designed constrained tree topologies as expected under the lineage-specific and ancestral rediploidization models (Fig. 3a). More specifically, for the constrained ancestral rediploidization topologies, we built constrained topologies as follows: we first placed 1R_1 and 1R_2 gnathostome gene copies in two different clades following 1R and then derived possible combinations of hagfish and lamprey genes to be placed on the 1R_1 and 1R_2 clades, using well-supported hagfish and lamprey chromosomal orthologies to limit the number of combinations (Extended Data Fig. 6 and Supplementary Table 5). Next, for each of these 1,247 families, we built gene trees using RAxML (v.8.2.12)[100], with 10 distinct starting trees and the PROTGAMMAJTT model, for: the unconstrained maximum likelihood (ML) tree; the constrained ancestral rediploidization topologies; and the constrained lineage-specific rediploidization topology. We then used the AU-test implemented in CONSEL (ref. 101) to test for significant differences in log-likelihoods reported by RAxML (ref. 100). A tree topology was rejected when significantly less likely than the ML tree at $\alpha = 0.05$.

We used the same approach to test for lineage-specific rediploidization in lampreys and hagfish on CLGB-1R after the 2R cyclostome hexaploidization. We ran likelihood tests on 30 informative gene families, constraining the lineage-specific rediploidization gene-tree topology as presented in Fig. 3c and constraining the ancestral rediploidization topologies according to the two possible ways of grouping hagfish and lamprey genes together (that is, either grouping genes from hagfish chr. 4 with lamprey chr. 10 and hagfish chr. 8 with lamprey chr. 2, or hagfish chr. 8 with lamprey chr. 10 and hagfish chr. 4 with lamprey chr. 8).

The code to reproduce the analysis, as well as the associated resulting gene trees, have been deposited in GitHub (https://github.com/fmarletaz/hagfish/tree/main/rediploidization).

## Phylogenetic tree based on concatenation of Hox clusters

We investigated the phylogenetic relationships between Hox clusters and bystander genes in seven genomes: amphioxus, sea lamprey, hagfish, human, mouse, chicken and spotted gar. We identified Hox and bystander genes in three steps: (i) starting from human gene names, we searched for orthologues in the other species using our set of reconciled gene trees (GeneRax trees); (ii) we used NCBI blastp (ref. 102) to confirm identified hox genes and further search for Hox genes missed by the gene-trees approach; and (iii) we used miniprot (v.0.5-r179)[103] with the sets of human and *E. burgeri* Hox proteins[24] to search for Hox genes missing from genome annotations of other species. We next aligned each gene family using their amino acid sequence with MAFFT (v.7.508)[95] and concatenated alignment from each cluster. Finally, we used the concatenation matrix to build a phylogenetic tree with RAxML-NG (v. 1.1)[100] using the LG+G4+F model, 10 different starting parsimony trees and 100 bootstrap replicates.

## Comparative transcriptomics

RNA-seq reads for hagfish (this study), lamprey *Lampetra japonica* (PRJNA354821, PRJNA349779 and PRJNA312435), gar *Lepisosteus oculatus* (PRJNA255881) and the cephalochordate amphioxus (PRJNA416977) were aligned with STAR (v.2.5.2b)[79], and counts for annotated genes were obtained using featureCount from the subreads package (v.1.6.3)[104]. Counts were converted to FPKM in the R package for subsequent analyses: WGCNA (v.1.7.0) was used to cluster gene expression in the full organ set and, after filtering out genes with limited variance and coverage, the 'softpower' parameter was estimated to be 20, and clustering was run with a 'signed' network type[105]. The gene-expression specificity index (or $\tau$) was calculated as described previously[106] on sets of organs (brain or neural tube; gills; heart; intestine; kidney; liver or hepatic tissues; ovary or female gonad; skin or epidermis; and muscle). For comparative analyses, gene families with paralogues derived from the vertebrate WGD were selected on the basis of their duplication history, and the gene-expression specificity index was compared across species for the same gene families (Extended Data Fig. 10b). We also compared gain and losses of expression domains for a given gene family by binarizing gene expression across a reduced set of six organs (brain, gills, intestine, liver, muscle and ovary) and counting expression patterns gains of lost between genes belonging to a given gene including paralogues and outgroup. The number of gain and loss events is then plotted as a distribution centred around zero (Fig. 4c).

The expression of paralogous genes in lamprey neural crest was assessed by quantifying gene expression using Salmon (v.1.10.0)[107] from RNA-seq data generated in a previous study[56] on dissected cranial and trunk dissected tissues using the latest lamprey genome and annotation[7]. Paralogy status and expression is specified in Supplementary Table 10.

## Detection of germline-enriched and germline-specific regions

DNA was extracted from testes and blood by phenol-chloroform extraction[72]. To enrich for germ cells, testes tissue was ground gently with a plastic pestle in a 1.5-ml microfuge tube and residual connective tissues were discarded before proteinase K digestion. Outsourced library prep and Illumina sequencing (HiSeq2500 V4, 150-bp paired-end reads) were performed by Hudson Alpha Genome Services Laboratory.

Sequence data were aligned to the *E. atami* genome assembly using BWA-mem (v.0.7.5a-r416)[108] with option -a and filtered by samtools view[108] with option -F2308. Only primary alignments with mapping quality 5 and higher were retained for further analysis. The resulting files were processed using DifCover (v.3.0.1)[7,11,34] to calculate the degree of germline enrichment across all discontiguous 500-base intervals of low-copy sequence using modal coverages for testes and blood of 32× and 54× respectively, low-coverage masking of regions with a read depth of less than 1/3× in both samples and high-coverage masking of sequences with a read depth greater than 3× modal coverage in both samples. To identify germline-specific genes that are present at a higher copy number, we ran DifCover using low-coverage masking with a read depth of less than 10× in both samples and high-coverage masking of sequences with a read depth greater than 30× modal coverage.

## PCR validation of germline-enriched loci

Primers were designed using a coverage-masked version of the *E. atami* genome using Primer3 (version 4.1.0)[109]. Amplication of PCR validation reactions was performed using GoTaq DNA polymerase (Promega, 1.2 units per 50 µl reaction), Colorless GoTaq Reaction Buffer, 1 µg genomic DNA template and 100 ng oligonucleotide primer. PCR cycling conditions included a 3-min initial denaturation step at 95 °C, 34 cycles of a three-step thermal cycling consisting of a 30-s denaturation at 95 °C, a 30-s primer annealing step at 55–65 °C (Supplementary Table 13) and a 30-s extension step at 72 °C. A final extension at 72 °C was performed on all reactions to ensure that full-length amplicons were produced.

Amplification was assessed by agarose gel electrophoresis. Eight primer pairs with an ambiguous signal in the first round of PCR were redesigned and retested (Supplementary Table 13). We note that some PCR markers might not be fully diagnostic with respect to germline specificity, as somatic gene duplicates are continuously captured by the germline-specific chromosomes in both lamprey[7,11,34] and songbird lineages[34].

## Computational prediction of germline-enriched and highly abundant somatic repeats

Abundant *k*-mers ($k = 31$) were identified from testes and blood DNA-sequencing data using Jellyfish (v.2.2.4)[110]. Minimal copy-number thresholds for defining abundant *k*-mers were set at 3× the modal copy number: 72 for testes and 120 for blood. Abundant *k*-mers were extracted and assembled into a set of de-novo-assembled repetitive sequences using Velvet (v.1.2.10)[111] with a hash length of 29. These sequences were aligned (blastn with -word_size 17) to repetitive elements generated from the *E. atami* genome assembly by RepeatModeler[112] and sequences that aligned with less than 90% identity or less than 80% of their length were added to the set of reference-derived repeats to form a union set.

Enrichment analysis was performed by separately aligning paired-end reads from testes and blood to the union set of repeats. Primary alignments, identified by samtools view[113] with option -F2308, were also filtered to retain only alignments that either cover more than 80% of a repeat or have more than 80% of read bases aligned. Enrichment scores were calculated with DifCover pipeline (v.3)[11]. Stage 2 of the pipeline was run with parameters v=10000, l=0, a=b=10, A=B=10^8. Stage 3 of the pipeline was modified by using a subroutine from DNAcopy[114] without 'smoothing' the data before analysis. From a set of 180,032 intervals generated by DifCover, we chose 138 highly abundant and germline-specific sequences with enrichment scores of more than 10 and an estimated span size of more than 100 kb. The estimated genomic span of these repeats was computed as [(testes coverage/modal testes coverage) × (number of bases with read depth coverage > 10)], in which modal testes coverage = 32.

Clustering of 138 highly abundant and germline-specific sequences was performed using CD-HIT-EST (v.4.6, with parameters: -c0.8, -G0, -aS 0.3, -aL 0.3, -sc 1, -g 1, -b 4)[115], resulting in the formation of 38 clusters that were further merged to 24 by manual curation and cross-alignment of sequences from the initial clusters. For characterization of repetitive structures and identification of motifs, representative sequences from each cluster were mapped to the assembly (blastn, -word_size 15) and to a collection of published hagfish repeats. We found that 4 of 24 clusters have sequences that are homologous to the published repeats of *Paramyxine sheni* EEPs2, EEPs3 and EEPs4 and *Eptatretus okinoseanus* EEEo2 (refs. 30,33). Primers for these and representatives of 7 other clusters were designed with the Primer3 (v.0.4.0) tool (Supplementary Table 16).

To facilitate FISH visualization, we also searched for possible candidates for centromeric repeats. Such candidates are expected to be (1) highly abundant in both somatic and germline sequence and (2) enriched in a 'centromeric' region of every chromosome. From the union set, we chose repeats with blood coverage > $10^5$ or span > 1 Mb and aligned them to the assembly (blastn -word_size 15, p>75, coverage > 80%). Repeats with more than 200 hits in a 1-Mb window were grouped to three families labelled Soma1–Soma3. Soma2 seemed to be homologous to the *P. sheni* repeat EEPs1 (ref. 33) and Soma1 and Soma3 to the *E. burgeri* contigs LC047612.1 and LC047003.1. FISH analysis confirmed the in silico prediction that EEPs1 is highly abundant in both testes and blood DNA of *E. atami*.

To estimate more accurately the genomic span of chosen germline-enriched and somatic repeats, we realigned reads from blood and testes to the sequences of these repeats or to the sequence extended as a tandem repetition of repeat motifs spanning at least 150 bp

(Supplementary Table 12), and applied all described previously steps for filtering and coverage and span estimation.

## In situ hybridization

**Slide preparation.** Snap-frozen samples of blood and testes were used for slide preparation of somatic and germline cells for validation of the presence and specificity of repeats in different cell types. A small amount of blood (about 20 mg) was gently thawed on ice, mixed with 2 ml buffered hypotonic solution (0.4% KCl, 0.01 M HEPES, pH 6.8) and incubated for 30 min at room temperature. The cells were prefixed by gently mixing the suspension with several drops of fixative solution (methanol:acetic acid 3:1). After centrifugation (5,000g for 10 min), the supernatant was removed and cells were resuspended and fixed with methanol:acetic acid 3:1. Three further fixative solution changes were performed to ensure that cells were fully equilibrated to fixative solution. Fixed cells were stored at −20 °C. One fixative change was made before spreading the cell suspension onto slides. A drop of about 20 ml was applied to a steamed slide, which was immediately placed on a heating block in a humidity chamber at 60 °C for 1–2 min. After air drying, slides were examined with a microscope using a low condenser position to aid in viewing unstained nuclei and metaphases. Slides were aged for 1–3 days on a warming stage at 37 °C before hybridization. For germline cells, a piece of testis (30–40 ng) was minced with a razor blade, placed in a homogenizer and disaggregated in hypotonic solution. Testis cell suspensions were filtered through a 40–50-mm cell strainer to remove excess tissue. Subsequent steps of fixation and slide preparation for testis tissue were as described for blood.

**Probe labelling.** Probes for FISH were generated using a modified conventional PCR: the reaction mix with a final volume of 25 μl contained 0.1 mM each of unlabelled dATP, dCTP and dGTP and 0.03 mM of dTTP; 0.5 μl one fluorophore conjugated dUTP (cyanine 3-dUTP (Enzo), cyanine 5-dUTP (Enzo) or fluorescein-12-dUTP (Thermo Fisher Scientific)); 1× Taq-buffer; and 0.625 U GoTaq DNA Polymerase (Promega). Each PCR amplification was performed using 0.5 μg of genomic DNA template from testes, 34 PCR cycles and a 30-s extension step to obtain appropriately sized probes for FISH. After cycling the reaction, 25 μl PCR mix was combined with 5 μl sheared salmon sperm DNA (1 mg ml$^{-1}$; Thermo Fisher Scientific), 3 μl 3 M sodium acetate, pH 5.2 and 80 μl 100% cold ethanol, and kept overnight at −20 °C for probe precipitation. After spinning and supernatant removal, the pellet was dissolved in 25–30 μl of 50% formamide and stored at −20 °C before use.

**FISH.** FISH on chromosome preparations was performed according to a standard protocol for chromosome spreads[116] with modifications[117]. Before hybridization, slides were incubated in 2× SSC for 30 min at 37 °C, passed through an ethanol series (70%, 80%, 100%), dried and denatured in formamide (70% in 2× SSC) for 2 min, prewarmed to 70 °C. After the formamide denaturation, slides were placed immediately in cold (−20 °C) 70% ethanol, further dehydrated in 80% and 100% ethanol, and kept on a slide warmer at 37 °C until the hybridization mix with probe was applied.

Differently labelled hybridization probes were mixed (1 μl of each per slide) with hybridization master mix (60% formamide, 10% dextran sulfate and 1.2× SSC) to a final volume of 10 μl. The hybridization mix was denatured at 95 °C for 7 min, cooled in ice, prewarmed to 37 °C, applied to the slide, coverslipped and sealed with rubber cement. After overnight incubation in a humidity chamber at 37 °C, slides were washed in 0.4× SSC and 0.3% NP-40 for 3 min at 70 °C and in 2× SSC, 0.1% NP-40 for 5 min at room temperature. One drop of ProLong Glass Antifade Mountant with NucBlue Stain was placed in the centre of an area to be examined and covered with a coverslip.

**Microscopy and image analysis.** Slides were analysed with an Olympus-BX63 microscope using filter sets for DAPI, FITC, Cy3 and Cy5. Images were captured using CellSens software (Olympus) and processed with Adobe Photoshop CC 2019 and ImageJ 1.53k (NIH).

## Reporting summary

Further information on research design is available in the Nature Portfolio Reporting Summary linked to this article.

## Data availability

Raw and processed sequences have been deposited in the NCBI Sequence Read Archive (SRA) (PRJNA953751) and Gene Expression Omnibus (GEO) (GSE230176). The RNA-seq data for *M. glutinosa* are available on the SRA (SRR25213276). The resequenced somatic tissues are also available on the SRA (blood, SRR24133795; testes, SRR24130678). RNA-seq datasets used for comparative analyses are publicly available for Japanese lamprey (PRJNA354821, PRJNA349779 and PRJNA312435), gar (PRJNA255881), amphioxus (PRJNA416977) and sea lamprey (PRJNA497902). The read data are available at PRJNA953751. The genome and its annotation are also deposited in zenodo: https://zenodo.org/records/10227719. Source data are provided with this paper.

## Code availability

The code used is available at https://github.com/fmarletaz/hagfish.

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

**Acknowledgements** We thank B. Venkatesh and S. Kuraku for early discussions; N. Segi, T. Suzuki, B. Muramatsu and H. Dohra for technical support; H. Hasegawa and K. Hasegawa for hagfish supply; and M. Levine, M. Martik, H. van Mullem, C. Amemiya and L. Piovani for comments on the manuscript. Work at the OIST Molecular Genetics Unit (D.S.R., O.S., D.G. and F.M.) was supported by OIST internal funds. F.M. is supported by the Royal Society Fellowship URF\R1\191161 and the BBSRC grant BB/V01109X/1. D.S.R. is a Chan Zuckerberg Biohub Investigator and is supported by the Marthella Foskett Brown Family Chair of Biological Sciences at the University of California, Berkeley. J.J.S. is supported by grants from the National Institutes of Health (NIH) (R35GM130349) and the National Science Foundation (NSF) (MCB1818012). M.S. was in part supported by the Field Science Center and Research Institute of Green Science and Technology, Shizuoka University. E.P. is supported by a Newton International Fellowship from the Royal Society (NIF\R1\222125). We thank the OIST Sequencing Section for DNA and RNA sequencing and acknowledge the support of OIST supercomputing and the University of Kentucky High-Performance Computing complex.

**Author contributions** D.S.R., O.S., F.M., J.J.S. and S.B. conceived the study, which was led by F.M., J.J.S. and D.S.R. F.M. and O.S. sequenced, assembled and annotated the genome. N.T., V.A.T. and J.J.S. performed the DNA elimination analysis. F.M., O.S. and D.S.R. performed synteny analyses. F.M. completed phylogenetic and paralogon analysis. E.P. contributed duplication phylogenetic modelling, lineage-specific rediploidization and Hox cluster analysis. D.G. analysed transcriptomic data. M.S. provided biological samples and figure material. K.K. provided biological material and took part in transcriptomic analyses. D.S.R., F.M., E.P. and J.J.S. wrote the paper with input from N.T. and V.A.T. All authors read and approved the manuscript.

**Competing interests** The authors declare no competing interests.

**Additional information**
**Correspondence and requests for materials** should be addressed to Ferdinand Marlétaz, Jeramiah J. Smith or Daniel S. Rokhsar.

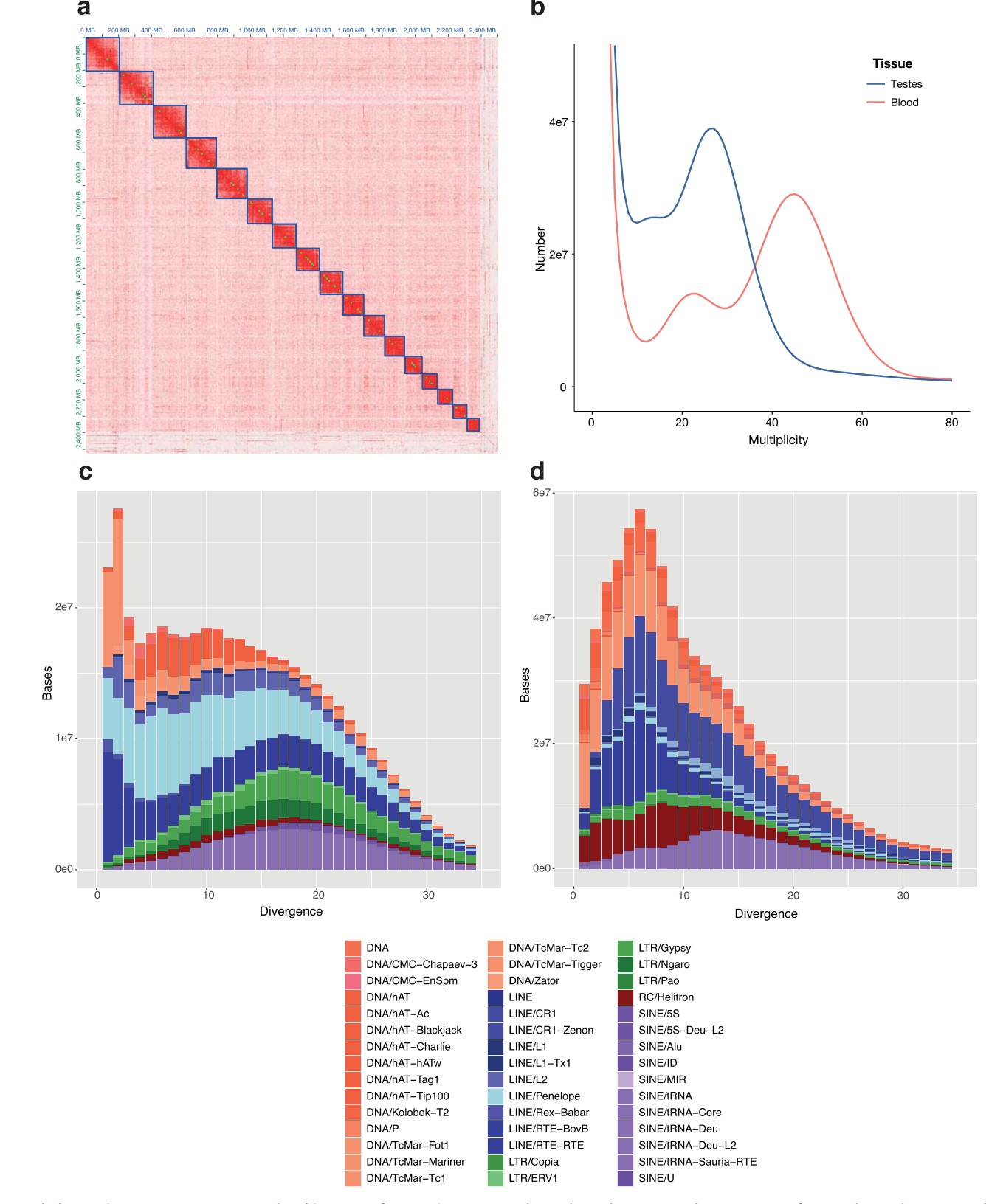

**Extended Data Fig. 1 | Genome content and architecture of *E. atami*.**
**a**, Hi-C contact map visualizing the density of interactions between binned genomic regions in the proximity ligation data. The high contact regions are consistent with the 17 somatic chromosomes. **b**, Density of 21-mer of increasing multiplicity in the somatic (blood) and germline (testes) shotgun sequence data indicated an estimated genome sizes of 2.02 and 3.28 Gb, respectively. **c**,**d**, Repeat landscape summarizing the fraction of regions diverging from consensus repeats at varying levels of divergence (Kimura 2-parameter distance) in lampreys (**c**) and hagfish (**d**). Lamprey and hagfish show a markedly different profile with respect to the number and diversity of repetitive element classes.

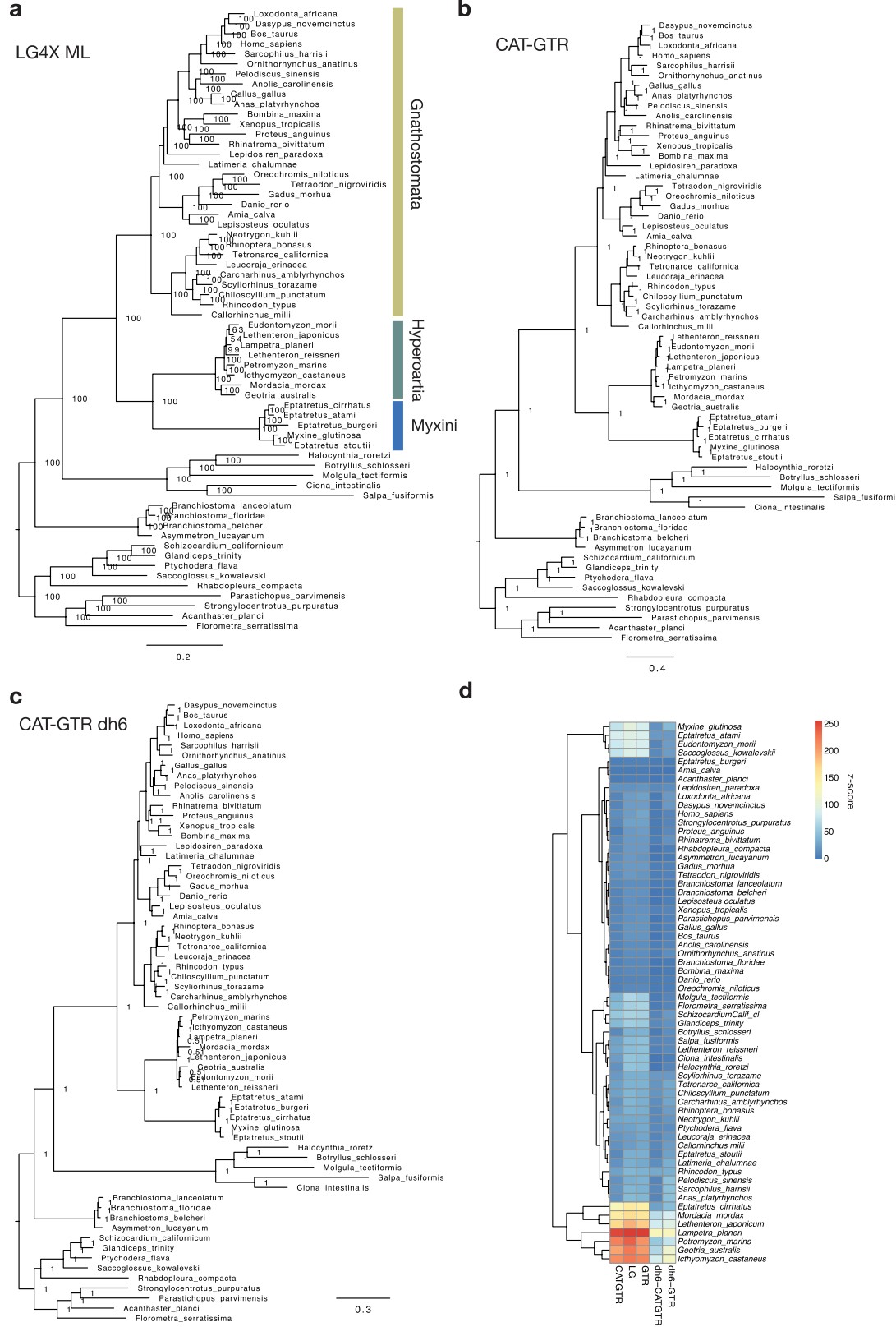

**Extended Data Fig. 2 | Phylogenetic reconstruction of deuterostome relationships with a focus on cyclostome position. a,** Tree reconstructed with IQ-TREE assuming LG4X model using a dataset of 1,467 single-copy orthologues and a partitioned model. **b,** Tree reconstructed using PhyloBayes and a CAT+GTR+G4 model using a subset of 176 orthologues showing the lowest saturation (see methods). **c,** Tree reconstructed using the same set of orthologues after Dayhoff 6 categories amino acid recoding to account for possible compositional heterogeneity due to high GC% in cyclostome genomes. **d,** z-score of posterior predictive analyses to assess composition heterogeneity. Positive z-scores indicate that average amino acid diversity is underestimated (negative z-scores indicate an overestimation) which highlights the composition bias existing in some lamprey and hagfish species and shows that recoding (Dayhoff 6) alleviates these biases.

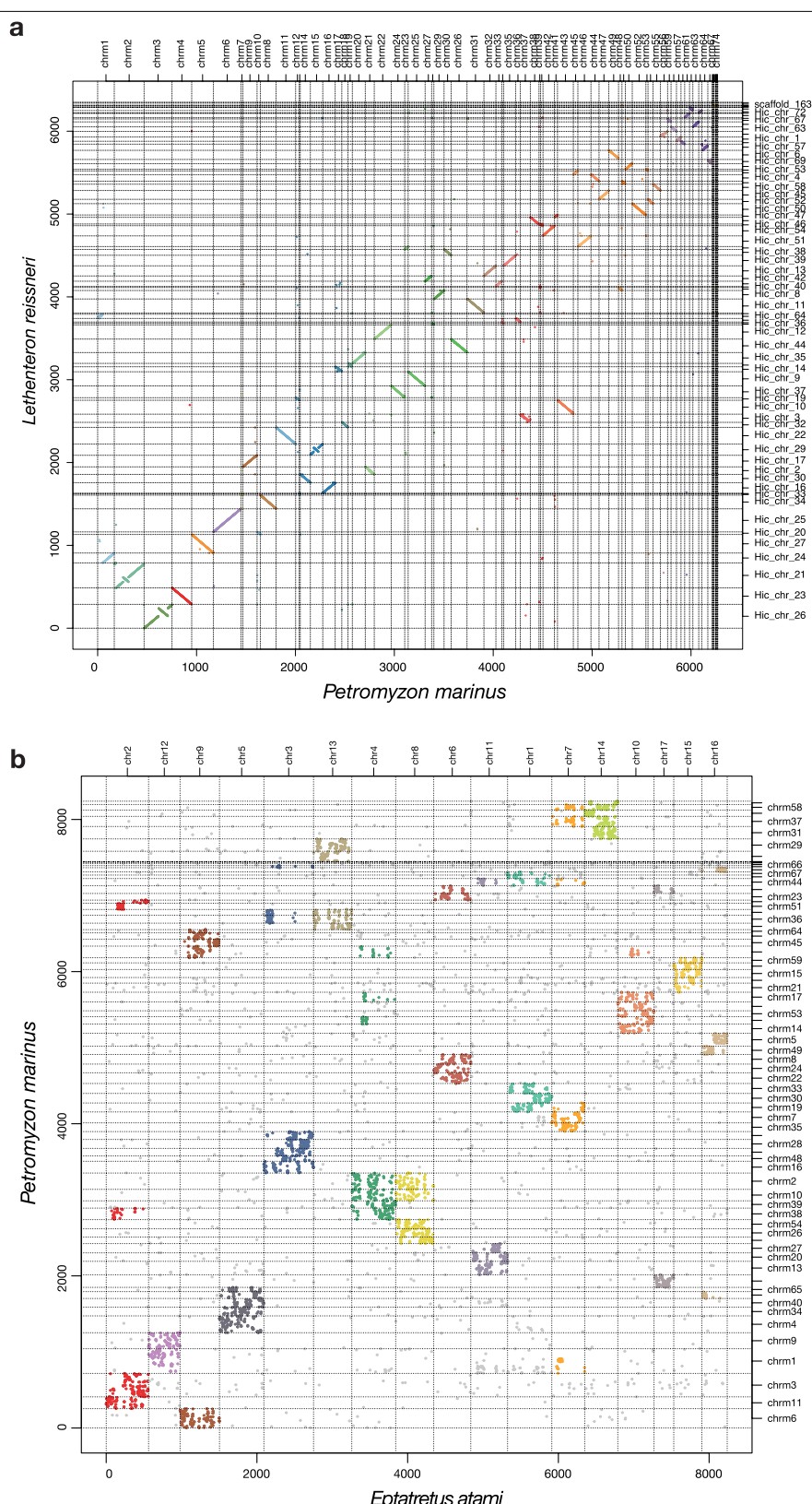

**Extended Data Fig. 3 | Comparison of the chromosomal architectures of cyclostome genomes. a**, Comparison between two lampreys (*Lethenteron reissneri* and *P. marinus*) highlighting the conservation of both chromosomal identity and extensive collinear segments. **b**, Comparison between the hagfish *E. atami* and the lamprey *P. marinus*. In both panels, dots show the relative location of orthologous genes between two species, coloured if the chromosome: chromosome enrichment is significant by Fisher's exact test (Methods); others shown in grey. The colours in **a** and **b** are based on *P. marinus* and *E. atami* chromosomes, respectively. In **b**, *P. marinus* chromosomes are sorted to aid in visualizing many-to-one mappings shown in Fig. 1d.

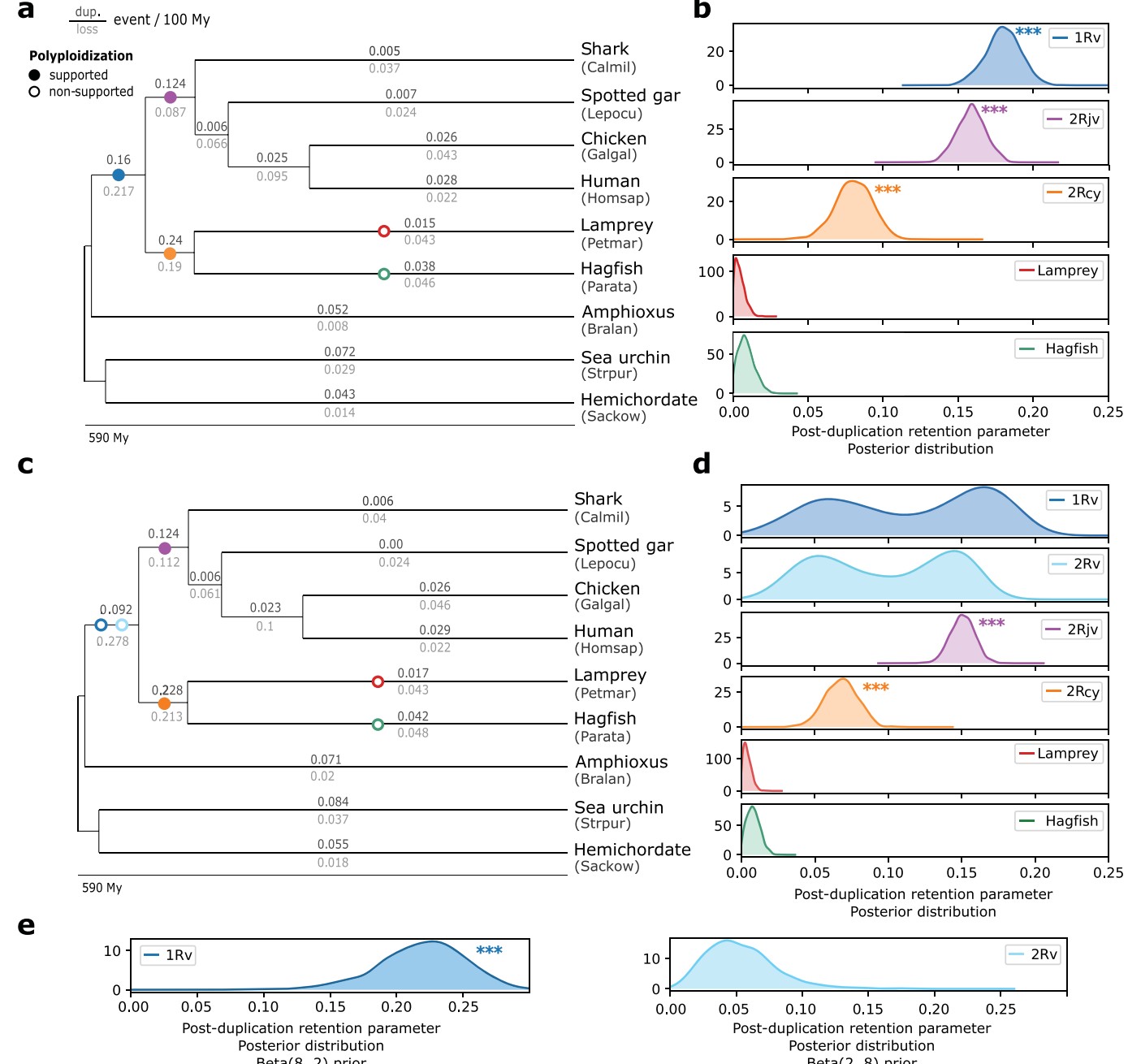

**Extended Data Fig. 4 | Tests of genome duplication hypotheses on the vertebrate tree. a**, Species phylogeny and polyploidization hypotheses tested with WHALE[36] using 8,931 gene families (Methods, see Supplementary Table 8 for details of the genomes used in the analysis). Polyploidization hypotheses are indicated by circles on the corresponding branches, with supported polyploidizations indicated with solid circles. Inferred background gene duplication and loss rates are presented on the branches. **b**, Posterior distribution obtained for the WHALE post-duplication retention parameter q, for each hypothesis presented in **a**. Stars indicate distributions significantly different from 0 (Bayes factors $BF_{Null\_vs\_WGD} < 10^{-3}$), which correspond to the supported polyploidization events. **c**, Alternative set of polyploidization

hypotheses tested, as in **a**, but with two successive duplications proposed in the ancestral vertebrate lineage ($1R_v$ and $2R_v$). **d**, Posterior distribution obtained for the WHALE post-duplication retention parameter, for each hypothesis presented in **c**. Here, the posterior distribution for retention parameters of the $1R_v$ and $2R_v$ events are bimodal, suggesting that the method cannot effectively separate parameters estimated for $1R_v$ and $2R_v$ when starting from identical priors. **e**, Use of distinct priors on $1R_v$ (Beta(8, 2)) and $2R_v$ (Beta(2, 8)) separates the estimated posterior distribution into distinct unimodal posterior distributions and provides support for a single shared $1R_v$ event in the vertebrate stem lineage. This analysis was performed on a random subset of 1,000 gene families, to reduce computational time (Methods).

**a**

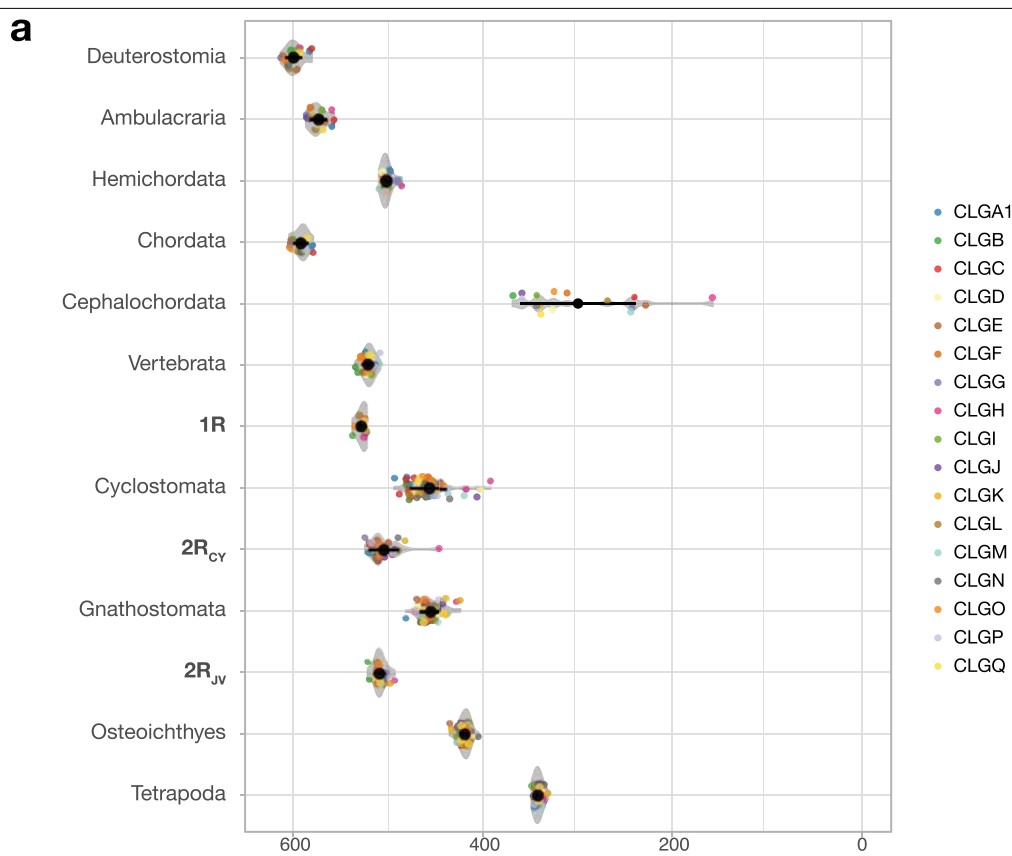

**b**

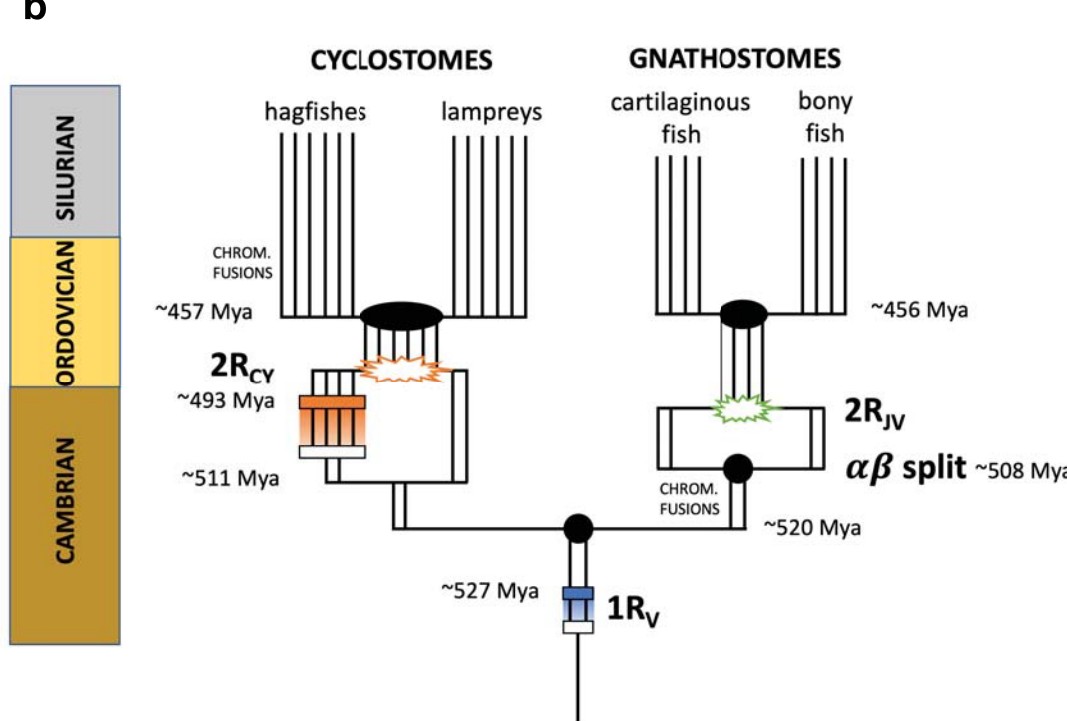

**Extended Data Fig. 5** | See next page for caption.

**Extended Data Fig. 5 | Timescale of vertebrate genome evolution.**
**a**. Distributions of timings for speciation and duplication events derived from paralogon phylogenies, showing details of the distributions indicated in Fig. 3c. **b**. Scenario for genome duplication and speciation events during early vertebrate evolution. Filled black circles or ovals mark speciation events; horizontal rectangles indicate presumptive auto-tetraploidies; starbursts indicate allo-polyploidies arising from hybridization of distinct progenitors (for example, alpha–beta in gnathostomes). Timings are based on **a**. Note that although speciation times (for example, the split between gnathostome progenitors alpha and beta, divergence of lamprey and hagfish lineages) can be estimated from gene or paralogon trees, hybridization times (for example, $2R_{JV}$, shown as green starburst) cannot be estimated from gene-tree analysis. Similarly, homoeologous recombination after auto-tetraploidization implies that the auto-tetraploidization event itself cannot be timed, but only the cessation of homoeologous recombination. Thus, the estimate of around 527 Ma for $1R_V$ (horizontal blue rectangle) represents the cessation of recombination after this presumptive auto-tetraploidy (open rectangle on vertebrate stem) with homologous recombination represented by blue shading. The absolute timing of $1R_V$ itself is unknown. (Auto-tetraploidy is suggested by the lack of differential gene loss between the two paralogous branches after 1R, as noted previously[19].) The rough estimate of a 10-million-year interval between the alpha–beta split and $2R_{JV}$ allo-tetraploidy is based on analogy with recent vertebrate allo-tetraploidies in frogs and goldfish. Cyclostome hexaploidization $2R_{CY}$ is shown as a two-step process culminating in the hybridization of diploid and tetraploid stem cyclostomes (orange starburst). This scenario follows the recent model of hexaploidy in sturgeon in which auto-tetraploids and diploid species coexist and hybridize[48]. In this scenario, the earliest divergences among cyclostome paralogues occurs around 511 Ma when the diploid and future tetraploid lineages split, which could be coincident with the early tetraploidization itself. Homoeologous recombination (shown as orange shading) is largely complete by around 493 Ma, defining a second peak in paralogue divergence (horizontal orange rectangle). Not shown is ongoing homoeologous recombination in CLGB which continues into the stem hagfish and lamprey lineages, as discussed further in the main text.

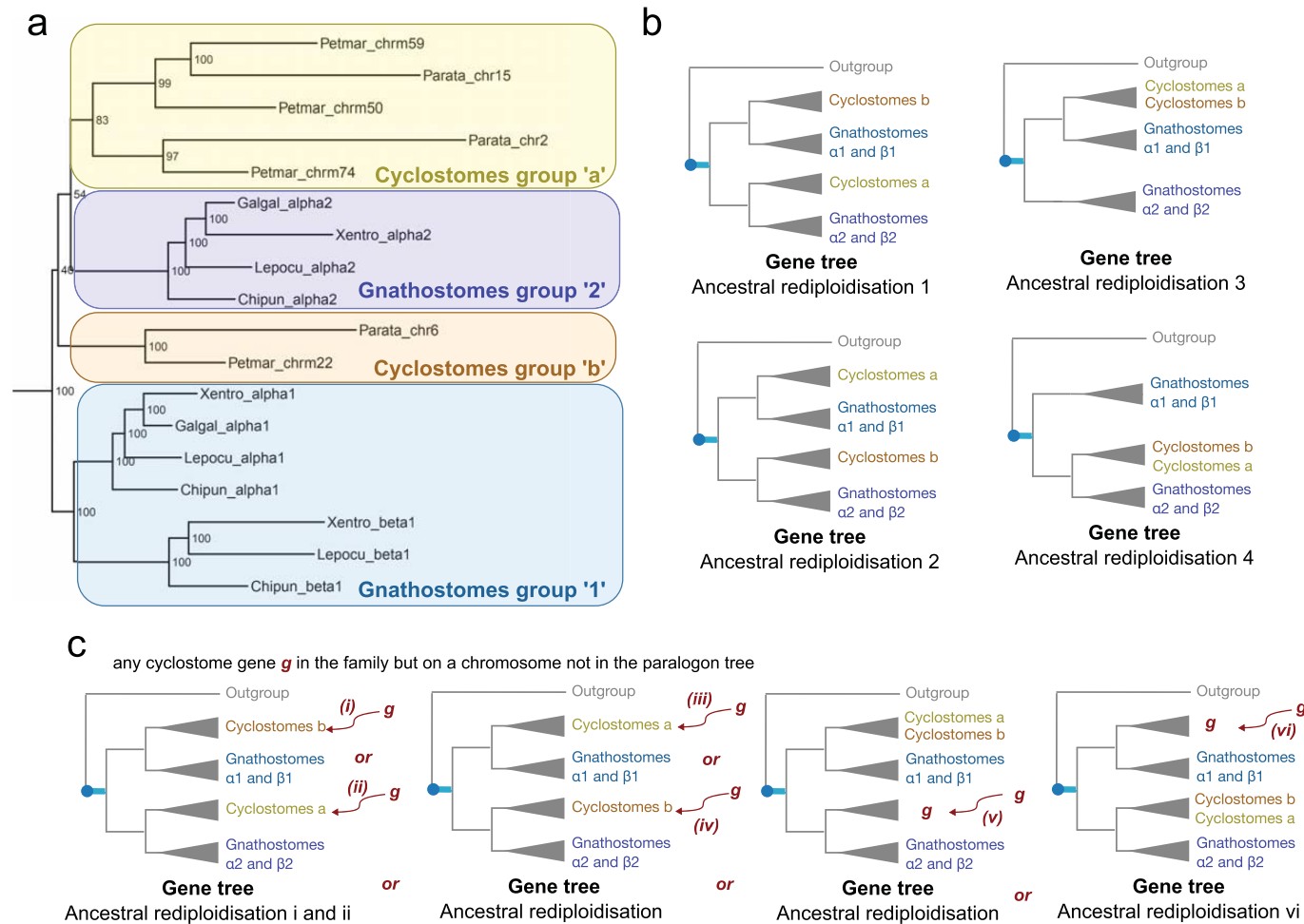

**Extended Data Fig. 6 | Method for construction of post-1R ancestral rediploidization constrained gene-tree topologies, using CLGM as an example. a**, Gnathostome 1R 1 and 1R 2 copies can be confidently identified and serve as a skeleton to build ancestral rediploidization tree topologies (blue-purple groups). Hagfish and lamprey chromosomes confidently grouped in a clade from the CLGM paralogon tree are defined as potential 1R-derived paralogons (yellow-orange groups) and kept together in the constrained ancestral rediploidization tree topology (see **b**). All sets of cyclostome chromosomes that were kept together for other CLGs are indicated in Supplementary Table 5. **b**, Possible groupings of hagfish and lamprey genes with gnathostome genes based on their chromosomal location, following 1R ancestral rediploidization. **c**. Genes located on hagfish and lamprey chromosomes that are not considered in the reconstructed paralogon tree (due to low representation because of small-scale rearrangements displacing them on different chromosomes) can each be placed on either side of the duplication in the absence of any prior information. In the presented scenario, this results in six different possible ancestral rediploidization (i to vi) constrained tree topologies. Only topologies with a maximum of three lamprey genes and three hagfish genes on each side of the 1R are permitted, to remove possibly confounding effects of complex multicopy gene families.

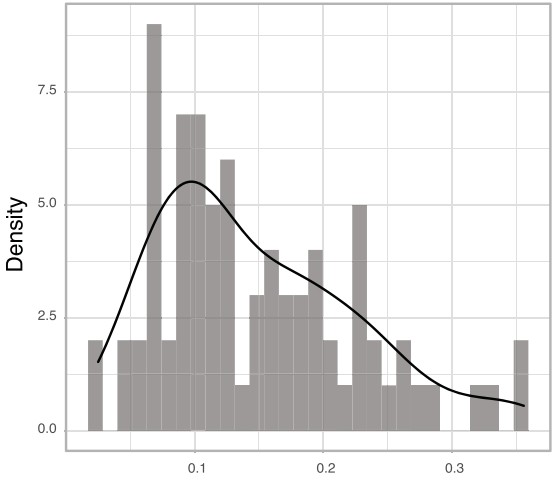

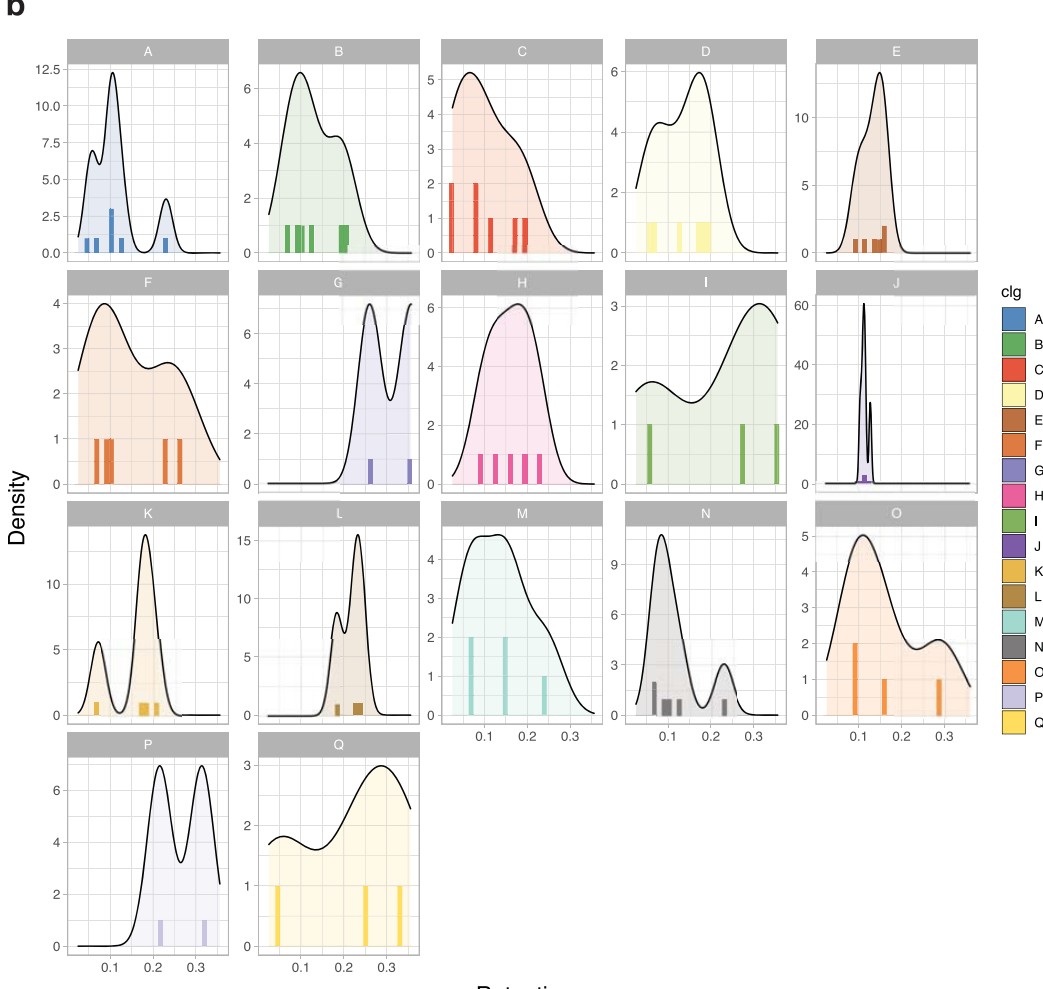

**Extended Data Fig. 7 | Orthologue retention rates after 2R$_{CY}$.** Retention is computed for each *P. marinus* chromosomal segment derived from a single CLG as the fraction of orthologues maintained on the segment in a comparison with the total number of orthologues for the same CLG in *Branchiostoma floridae*.

**a**, Distribution of retention rates plotted for all CLGs. **b**, Distribution of retention rates plotted for each CLG. These distributions are not distinctly bimodal, in contrast to the finding for 2R$_{JV}$[19]. Lamprey is used because it more closely preserves ancestral cyclostome state, as proposed previously[20].

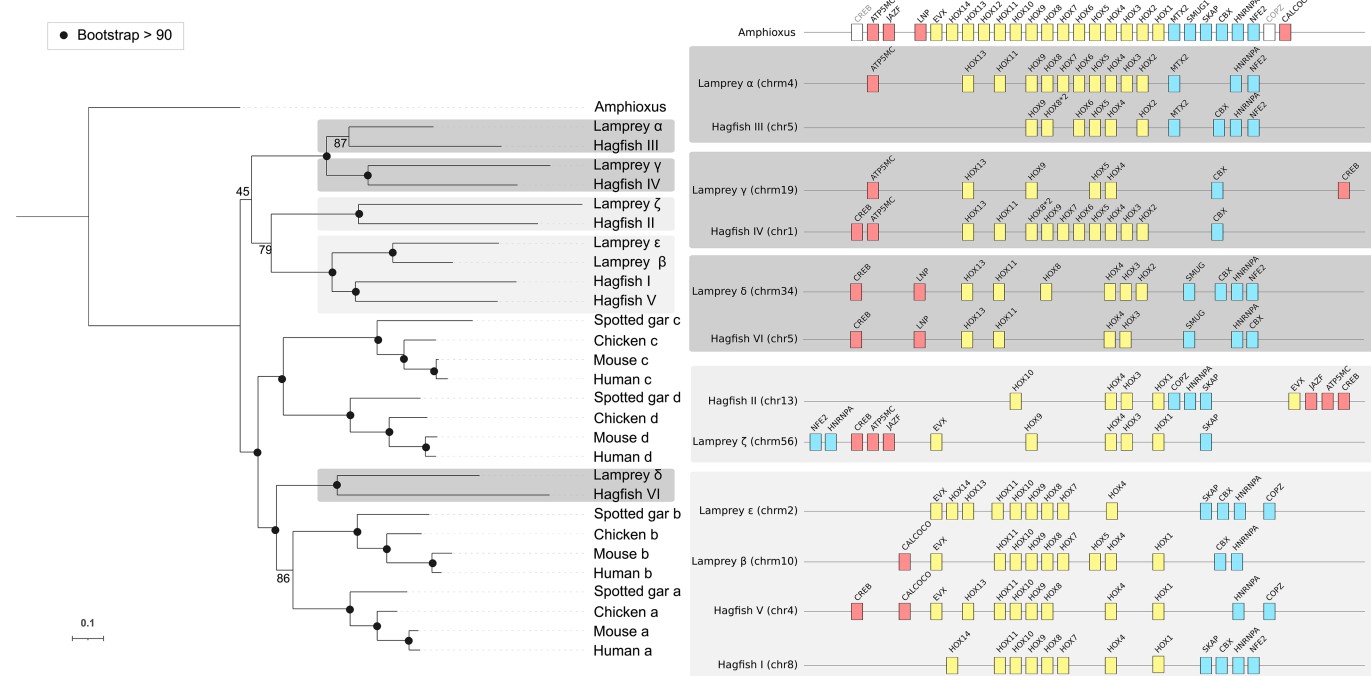

**Extended Data Fig. 8 | Phylogeny of the Hox clusters on the basis of a concatenation of Hox genes and bystanders.** Left, phylogeny of the Hox clusters, with node bootstrap support. One-to-one orthologies for gnathostome clusters are well-supported, similarly for cyclostome clusters with the exception of hagfish V-I and lamprey β-ε. Dark grey boxes highlight cyclostomes clusters that are expected to be orthologous to gnathostome clusters A/B based on chromosomal orthology (Fig. 3d), similarly light grey boxes are for expected orthologs to gnathostome clusters C/D. Right, schematic representation of cyclostome and gnathostome Hox clusters. Hox genes are shown as yellow boxes, 5' bystanders as red boxes and 3' bystanders as blue boxes. The order of genes reflects the actual arrangement of genes in each species.

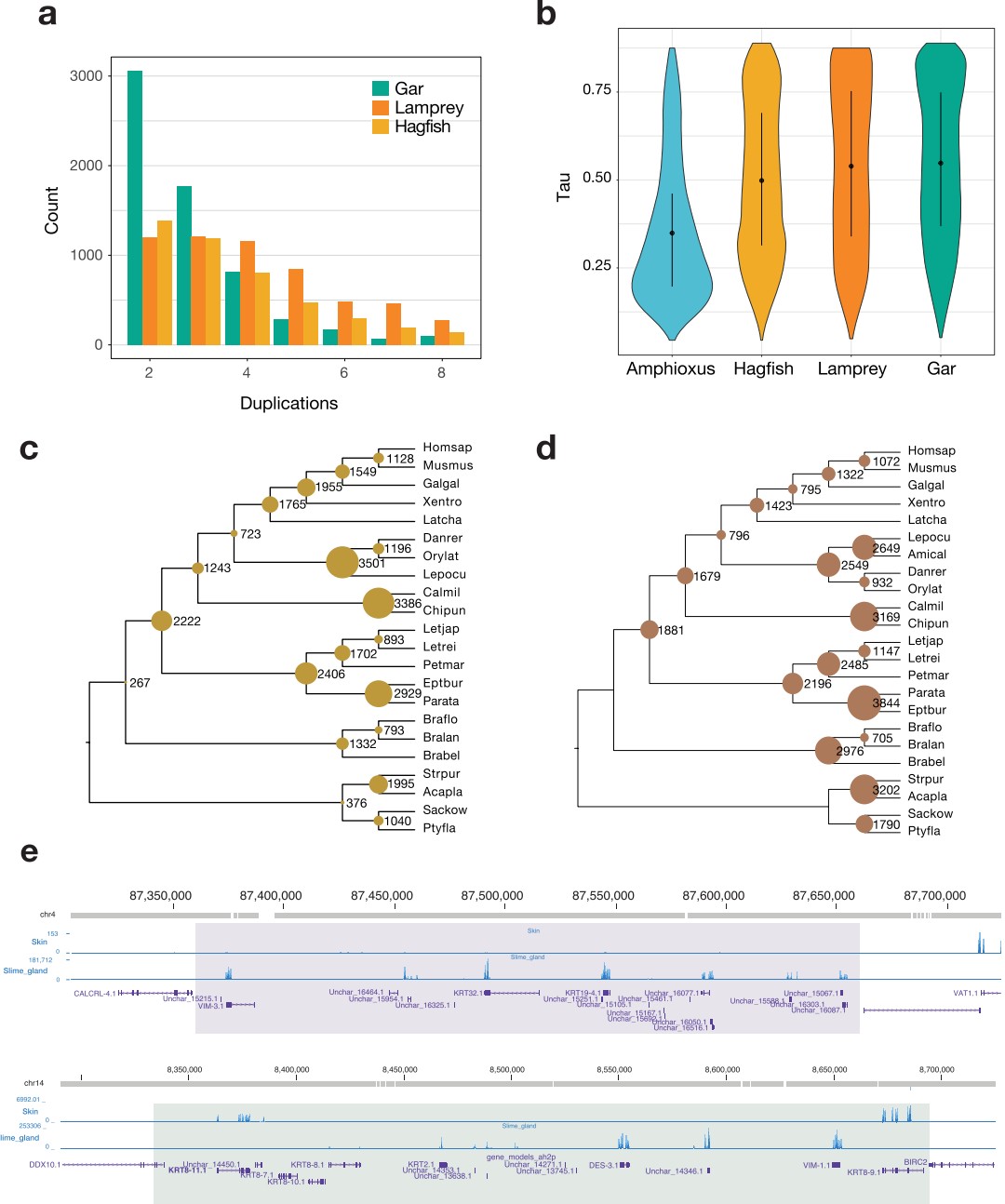

**Extended Data Fig. 9 | Evolution of duplicated genes and gene families in hagfish. a**, Counts of gene families containing the specified number of retained paralogues in gar, lamprey (*P. marinus*) and hagfish (*E. atami*). **b**, Comparison of the tissue-specificity of gene expression (tau index) for ohnologue gene families in lampreys, hagfish, gar and the (unduplicated) amphioxus outgroup (Methods). The distribution of the maximal tau value for each gene family is shown. **c**, Node-specific gene loss events inferred by GeneRax in a species–gene tree reconciliation framework (Methods). Species labels are specified in Supplementary Table 8. **d**, Loss of Panther families across deuterostomes species inferred as the most parsimonious events from gene-family composition. **e**, Genome structure of the two clusters of expanded keratin genes, with mRNA expression in slime gland and skin (blue track).

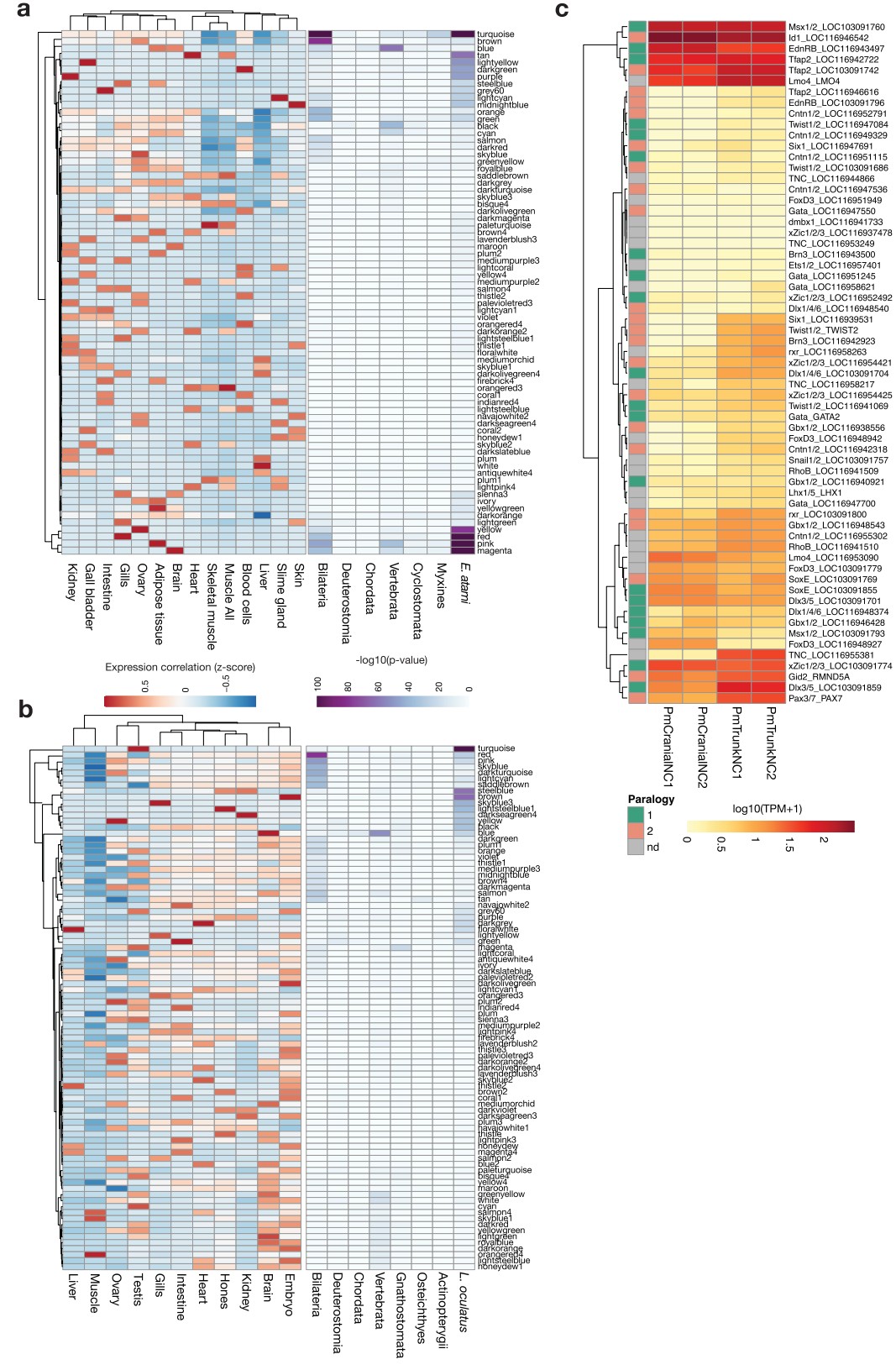

**Extended Data Fig. 10** | See next page for caption.

**Extended Data Fig. 10 | Gene expression and gene duplications in vertebrates. a,b,** Weighted gene co-expression network analysis (WGCNA) among organs for hagfish (**a**) and gar (**b**). Each row corresponds to a WGCNA cluster (with an arbitrary colour name) and its expression specificity is shown in selected tissues on the left (**a**, hagfish, **b**, gar). The enrichment of gene duplicated at successive phylogenetic nodes in each WGCNA cluster is indicated on the right as the p-value (-log10) of hypergeometric tests. A significant enrichment is observed in gene with strong neural expression (brain, blue cluster). **c,** Expression of selected paralogues involved in neural crest specification and migration in cranial and trunk neural crest tissues from lamprey *P. marinus*. RNA-seq data from a previous study[56] was quantified using the latest version of the lamprey genome and RefSeq annotation (kPetMar1). For each gene family, all paralogues derived from the vertebrate polyploidization event ($1R_V$ and $2R_{CY}$) are considered and classified (see Supplementary Tables 9 and 10). As denoted in inset, 1 (green cells) and 2 (pink cells) refer to the two original paralog branches derived from $1R_V$ (see main Fig. 4a). Grey groups could not be definitively assigned.

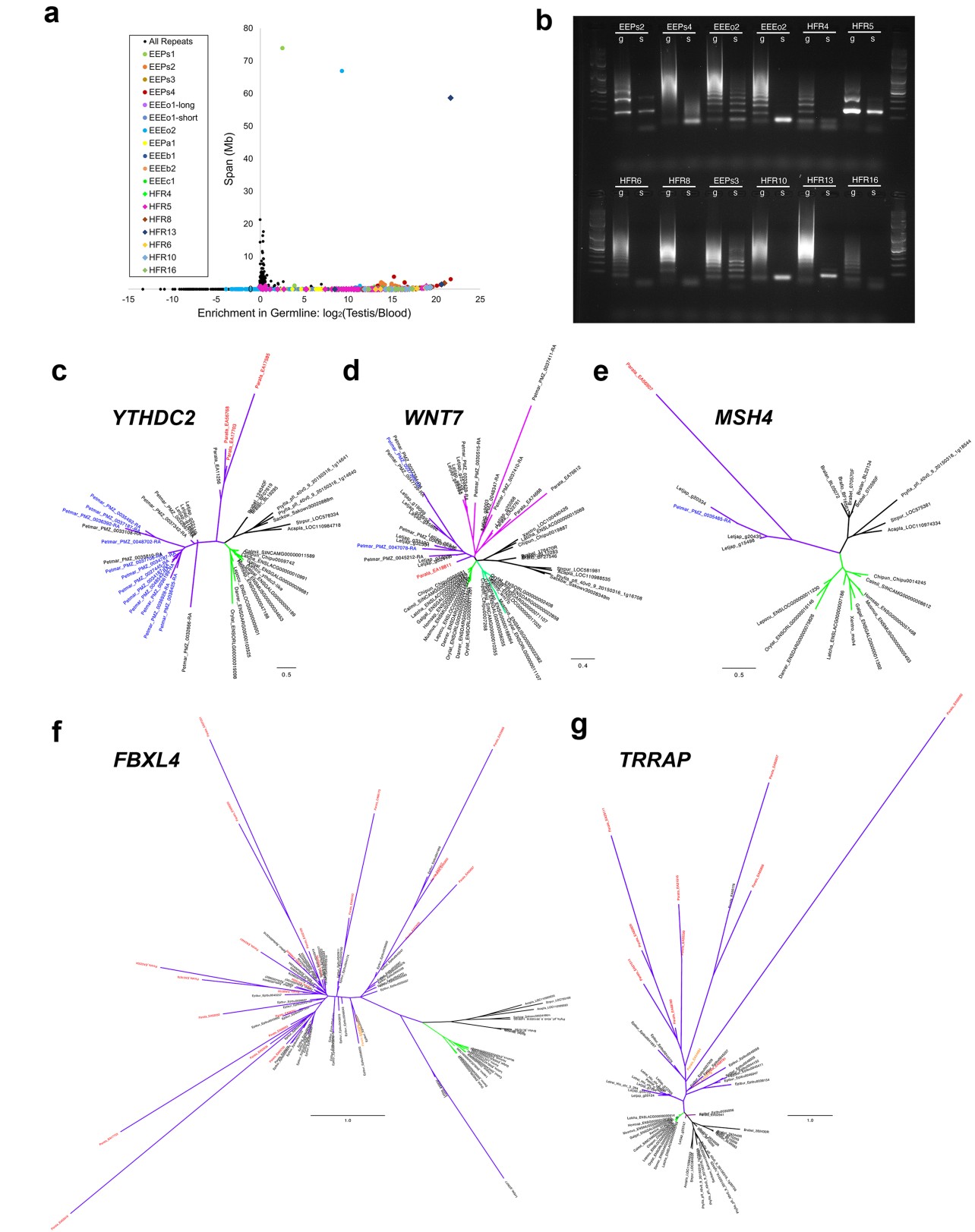

**Extended Data Fig. 11 | Eliminated genes and repeats identified in the hagfish genome. a**, Plot showing the degree of germline enrichment and estimated span of all predicted repetitive elements. Previously identified elements[30,33] are highlighted by coloured circles and new high-copy elements are highlighted by coloured diamonds. **b**, PCR validation illustrating germline enrichment and tandem repetition of predicted satellite elements. g: germline (testes) DNA used as template, s: somatic (blood) DNA used as template. **c–e**, Gene trees for homologues that are eliminated in both lamprey and hagfish. Gnathostome clades are highlighted in shades of green and cyclostome clades are highlighted in shades of purple. Individual germline-specific genes are highlighted in red (hagfish) or blue (lamprey). **c**, Tree for YTHCD2 homologues. **d**, Tree for WNT7 homologues. **e**, Tree for MSH4 homologues. **f,g**, Gene trees for homologues that are highly duplicated in hagfish. Gnathostome clades are highlighted in shades of green and cyclostome clades are highlighted in shades of purple. Individual germline-specific genes are highlighted in red. **f**, Tree for *FBXL4* homologues. **g**, Tree for *TRRAP* homologues.

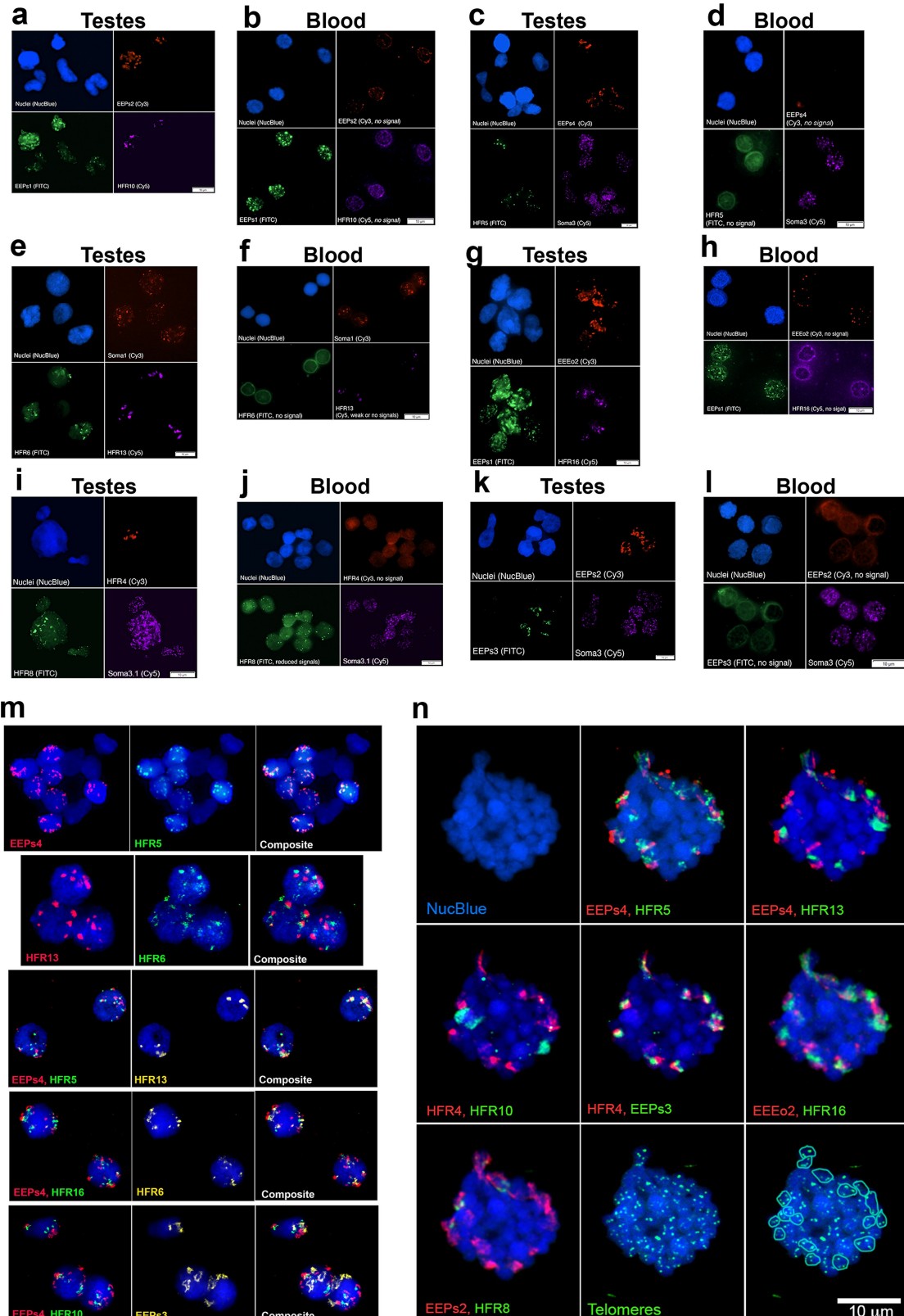

**Extended Data Fig. 12** | See next page for caption.

**Extended Data Fig. 12 | FISH of repeats to germline and somatic interphase nuclei.** Nuclei are labelled with the DNA stain NucBlue (blue) and for all panels labelled "no signal" fluorescence images are overexposed to both show background signal and aid in confirming the location of nuclei in those images. **a**,**b**, Germline enriched EEPs2 (red) and HFR10 (magenta), and the somatic repeat EEPs1 (green) are hybridized to nuclei isolated from **a**, germline: testes and **b**, soma: blood. **c**,**d**, Germline enriched repeats EEPs4 (red) and HFR5 (green), and the somatic repeat Soma3 (magenta) are hybridized to nuclei isolated from **c**, germline: testes and **d**, soma: blood. **e**,**f**, Germline enriched repeats HFR13 (magenta) and HFR6 (green), and the somatic repeat Soma1 (red) are hybridized to nuclei isolated from **e**, germline: testes and **f**, soma: blood. **g**,**h**, Germline enriched repeats EEEo2 (red) and HFR16 (magenta), and the somatic repeat EEPs1 (green) are hybridized to nuclei isolated from **g**, germline: testes and **h**, soma: blood. **i**,**j**, Germline enriched repeats HFR4 (red) and HFR8 (green), and the somatic repeat Soma3.1 (magenta) are hybridized to nuclei isolated from **i**, germline: testes and **j**, soma: blood. **k**,**l**, Germline enriched repeats EEPs2 (red) and EEPs3 (green), and the somatic repeat Soma3 (magenta) are hybridized to nuclei isolated from **k**, germline: testes and **l**, soma: blood. **m**,**n**, In situ Hybridization of probes for ten germline-enriched satellite sequences. **m**, Probes are hybridized to germline interphase nuclei. **n**, Probes are hybridized to germline interphase nuclei. The location of hybridization signals for telomere probes and approximate bounds of 18 germline-specific dyads, corresponding to nine distinct germline-specific chromosomes. For all images, pairs of repeats are shown to aid in visualizing the relative location of individual probes.

# Reporting Summary

## Statistics

For all statistical analyses, confirm that the following items are present in the figure legend, table legend, main text, or Methods section.

| n/a | Confirmed | |
|---|---|---|
| ☒ | ☐ | The exact sample size (*n*) for each experimental group/condition, given as a discrete number and unit of measurement |
| ☒ | ☐ | A statement on whether measurements were taken from distinct samples or whether the same sample was measured repeatedly |
| ☐ | ☒ | The statistical test(s) used AND whether they are one- or two-sided<br>*Only common tests should be described solely by name; describe more complex techniques in the Methods section.* |
| ☒ | ☐ | A description of all covariates tested |
| ☐ | ☒ | A description of any assumptions or corrections, such as tests of normality and adjustment for multiple comparisons |
| ☐ | ☒ | A full description of the statistical parameters including central tendency (e.g. means) or other basic estimates (e.g. regression coefficient) AND variation (e.g. standard deviation) or associated estimates of uncertainty (e.g. confidence intervals) |
| ☐ | ☒ | For null hypothesis testing, the test statistic (e.g. *F*, *t*, *r*) with confidence intervals, effect sizes, degrees of freedom and *P* value noted<br>*Give P values as exact values whenever suitable.* |
| ☐ | ☒ | For Bayesian analysis, information on the choice of priors and Markov chain Monte Carlo settings |
| ☒ | ☐ | For hierarchical and complex designs, identification of the appropriate level for tests and full reporting of outcomes |
| ☒ | ☐ | Estimates of effect sizes (e.g. Cohen's *d*, Pearson's *r*), indicating how they were calculated |

*Our web collection on statistics for biologists contains articles on many of the points above.*

## Software and code

Policy information about availability of computer code

| | |
|---|---|
| Data collection | No software used for data collection |
| Data analysis | Assembly: Meraculous (v2.2.2.5) , PBJelly (v15.8.24), Meryl (v1.1), Genomescope2<br>Annotation: STAR (v2.5.2b), Stringtie (v1.3.3b), Trinity (v2.11.0), quiver (v2.0.0), GMAP (v. 2018-03-25), Mikado (v1.2.1), Portcullis (v1.0.2), Trans-decoder (v5.5.0), RepeatModeler (v1.0.11), RepeatMasker (v4.0.7), PASA pipeline (v2.3.1).<br>Phylogenetics: OMA tool (v2.4.1), Hmmer (v3.1b2), Msaprobs (0.9.7), HmmCleaner (v1b), BMGE (v1.12),IQ-TREE (v2.1.1), Phylobayes (v4.1e).<br>Gene family analyses: MMSeqs2 (r12-113e3), Broccoli (v1.1), MAFFT (v7.305), generax (v1.2.2), WHALE (v2.1.0), RAxML-NG v. 1.1 , miniprot 0.5-r179 ,<br>Transcriptomics:  featureCount from subreads package (v1.6.3), WGCNA (v1.7.0)<br>DNA elimination: BWA-mem (v0.7.5a-r416), DifCover (Version 4), Jellyfish (v2.2.4), CD-HIT-EST (v4.6)<br><br>Custom code available at https://github.com/fmarletaz/hagfish |

For manuscripts utilizing custom algorithms or software that are central to the research but not yet described in published literature, software must be made available to editors and reviewers. We strongly encourage code deposition in a community repository (e.g. GitHub). See the Nature Portfolio guidelines for submitting code & software for further information.

## Data

Policy information about availability of data

All manuscripts must include a data availability statement. This statement should provide the following information, where applicable:

- Accession codes, unique identifiers, or web links for publicly available datasets
- A description of any restrictions on data availability
- For clinical datasets or third party data, please ensure that the statement adheres to our policy

Raw and processed sequences have been deposited in NCBI SRA (PRJNA953751 ) and Gene Expression Omnibus (GSE230176). The RNA-seq of Myxine glutinosa is available on the SRA (SRR25213276). The resequenced somatic tissues are also available on the SRA (Blood: SRR24133795 and Testes: SRR24130678).  RNA-seq datasets used for comparative analyses are publicly available for Japanese lamprey (PRJNA354821, PRJNA349779, PRJNA312435), gar (PRJNA255881) and amphioxus (PRJNA416977) as well as the sea lamprey (PRJNA497902).

# Field-specific reporting

Please select the one below that is the best fit for your research. If you are not sure, read the appropriate sections before making your selection.

☒ Life sciences ☐ Behavioural & social sciences ☐ Ecological, evolutionary & environmental sciences

For a reference copy of the document with all sections, see nature.com/documents/nr-reporting-summary-flat.pdf

# Life sciences study design

All studies must disclose on these points even when the disclosure is negative.

| | |
|---|---|
| Sample size | No statistical method was used for determining sample size. |
| Data exclusions | No data exclusions. |
| Replication | No such experiments. |
| Randomization | No randomization was used |
| Blinding | Blinding was not relevant to our study as comparison were performed by computer software not influenced by investigator. |

# Reporting for specific materials, systems and methods

We require information from authors about some types of materials, experimental systems and methods used in many studies. Here, indicate whether each material, system or method listed is relevant to your study. If you are not sure if a list item applies to your research, read the appropriate section before selecting a response.

### Materials & experimental systems

| n/a | Involved in the study |
|---|---|
| ☒ | Antibodies |
| ☒ | Eukaryotic cell lines |
| ☒ | Palaeontology and archaeology |
| ☐ | ☒ Animals and other organisms |
| ☒ | Human research participants |
| ☒ | Clinical data |
| ☒ | Dual use research of concern |

### Methods

| n/a | Involved in the study |
|---|---|
| ☒ | ChIP-seq |
| ☒ | Flow cytometry |
| ☒ | MRI-based neuroimaging |

## Animals and other organisms

Policy information about studies involving animals; ARRIVE guidelines recommended for reporting animal research

| | |
|---|---|
| Laboratory animals | *For laboratory animals, report species, strain, sex and age OR state that the study did not involve laboratory animals.* |
| Wild animals | Animals were samples in Suruga Bay, off Yaizu (300-330m depth) and maintained in seawater aquariums at 11-13°C. |
| Field-collected samples | *For laboratory work with field-collected samples, describe all relevant parameters such as housing, maintenance, temperature, photoperiod and end-of-experiment protocol OR state that the study did not involve samples collected from the field.* |

Ethics oversight | In agreement with procedures authorized by Guidelines for Proper Conduct of Animal Experiments by the Science Council of Japan (2006), animals were anaesthetized using Tricaine (MS222, Sigma) before sacrifice and dissection.

Note that full information on the approval of the study protocol must also be provided in the manuscript.

