## [Peer Review File · Nature]

Manuscript Title: The hagfish genome and the evolution of vertebrates

Reviewer Comments & Author Rebuttals

Reviewer Reports on the Initial Version:

Referees' comments:

Referee #1:

Remarks to the Author:

Overview

This manuscript reports a genome assembly and analysis for a hagfish species. Lampreys and hagfishes are the only two surviving lineages of jawless fish. Together they occupy a unique position in vertebrate phylogeny and hence a unique opportunity to gain insight into early vertebrate evolution. Lamprey genomes have been sequenced and analysed already. This manuscript offers the comparator genome from hagfishes, where study has lagged behind lamprey due to the difficulty in obtaining specimens and much larger genome size.

Assembling a high-quality genome facilitates comparative genomic studies. Some are outward looking, essentially focusing on how the early vertebrate genome evolved both prior to the separation of jawless and jawed vertebrate genomes, and within the jawless vertebrate lineage. Others are inward looking, focusing on the peculiar biology of hagfishes including their slime production and somatic genome reorganisation.

In general, I found the analyses robust. I have multiple comments/views about individual sections and which I will detail below. I have refrained from a more forensic review of the text at this stage as I think there is substantial work to do to make this suitable for publication and that might come later.

My overarching view of the manuscript is mixed. There are some genuinely insightful aspects that pin down important parts of vertebrate evolutionary history and should be of broad relevant and interest. There are also parts which are much less compelling, either because they address questions that have already been substantially advanced elsewhere, or because they are incomplete, speculative or do not address the interesting questions in those sub-areas.

In summary I think there is potential for this manuscript to make an exciting contribution to the literature that will be widely respected, and cited well and for a long time. However, I suggest to focus the paper onto the big questions here: chromosome evolution, genome duplication and the fates of duplicated genes. Possibly also somatic genome rearrangement, as I think this is a major outstanding question of genome function/evolution into which hagfishes are well placed to give insight, though as my comments below will show I struggled to understand what the authors were saying here.

General comments on individual sections

1. Chromosome evolution

This analysis is robust, important and interesting. Together with the section on genome duplications it seals off one of the big remaining questions of vertebrate genome evolution

2. Phylogenomics

This is technically robust but essentially confirmatory. Although the craniate hypothesis has some historical relevance, it has been clear for many years now that hagfishes and lampreys are monophyletic. Its an appropriate supplementary analysis for this paper but I felt its prominence here as an important finding and opening figure of the paper was overstated.

3. Genome duplications

This section adds two important findings on genome duplications: their timing relative to lineage splits and their true timing as deduced by clock analyses. I think these are both pivotal outcomes of this paper and with (10 above provide a compelling picture of early vertebrate genome evolution.

4. Hox clusters

The analysis in this section is well done and interesting to the many who work on Hox genes, though essentially descriptive. Biological insight as discussed is very speculative however.

5. Neural crest

As with the Hox clusters, I found the insight gained from this section to be a bit limited. It also seemed to ignore some of the published experimental work on lamprey neural crest genes, for example PMID: 17765683. I think there is insight to be gained from the approach, however, but would just need to be better developed and integrated with current functional knowledge (and possibly new expression data) to be persuasive.

6. Parologue fates

The analysis in this section looks robust. I found the outcomes interesting: particularly that there are differences in parologue fates in cyclostomes. This follows nicely on from the sections on the timing and consequences of chromosome evolution/genome duplication.

7. Hagfish gene novelties and losses

This section combines two analyses: a generic view of gene loss and some selected 'biology insight' analyses. The former fits well with the sections on parologue fates above. Insight from specific gene losses that may be linked to lost phenotypic traits like eyes is, however, limited. Many analogous cases have been discovered but really we learn little from such analyses without some functional follow-on.

8. Somatic genome rearrangement

In my opinion this is one of the big, fascinating questions about this clade of animals: how and why does such drastic somatic DNA loss occur? A complete genome is a fundamental basis for analysing this and I was hoping that having hagfish and lamprey as parallel examples would really shed insight into this peculiar biology. However I struggled to understand what the authors were concluding here. The study of repeats is important in a sense, but what does it tell us? Similarly, it is interesting to see that genes removed by somatic recombination seem to be mostly non-overlapping between lamprey and hagfish, but again it wasn't clear what this actually meant.

Other aspects

The supplementary figures are an integral part of this manuscript and essential to evaluate the analyses. However their utility is seriously undermined by a lack of description in the legends. For example, it is just about possible to work out what many graphs actually show by referring back to the methods, but even this can be challenging and that's speaking as someone with significant experience in this area. If this paper progresses and these transfer to extended data figures, its vital they be accompanied by proper explanatory legends.

Referee #2:

Remarks to the Author:

I have read the manuscript authored by Marletáz et al. with great interest, as the hagfish remains a poorly studied organism despite its significant phylogenetic position. The authors present a high-quality chromosome-scale assembly of the hagfish genome, which is utilized to explore important aspects of both vertebrate evolution and the unique biology of this species.

The first contribution of the paper is to provide further support for the monophyly of cyclostomes. The authors analyze a large set of orthologous genes using a collection of models, demonstrating that lampreys and hagfish share a common ancestor independent of gnathostomes. My impression was that the debate between the Cyclostomata vs Craniata hypotheses had been settled by a series of molecular studies performed in the 00's. Thus, while re-visiting the phylogeny of vertebrates with this new genome assembly is certainly warranted, I found this section of the study to be of limited impact.

Some parts of the study are compelling. For instance, the authors employ paralogon-based analysis of hagfish chromosomal organization, unveiling that its "simpler" karyotype is a derived state resulting from multiple fusion-with-mixing events. This finding is particularly relevant as it refutes two commonly held assumptions in the field: firstly, that the unusually complex karyotype observed in lampreys reflects a derived state, and secondly, that the large number of small chromosomes in lampreys is somehow linked to mechanisms of programmed DNA elimination. The results presented by the authors convincingly demonstrate that neither of these speculations holds true.

The study also tackles the sequence and timing of genome duplications in early vertebrate evolution. The authors employ paralogon-based phylogenetic analysis to propose an auto-tetraploidization event shared by vertebrates and establish the timing of independent duplications in gnathostomes and cyclostomes. The strategy for this analysis is well-designed, but the conclusions do not appear particularly novel, as they seem to confirm the ideas of cyclostome hexaploidy proposed by Nakatani and colleagues (Nat. Comm. 2021). The major contribution here lies in demonstrating that hexaploidy is shared by both hagfish and lamprey, which in itself is a significant finding.

Next, the authors delve into the evolution of different gene groups, such as Hox genes and neural crest genes, in light of their paralogon-based phylogenies. They propose that the origin of the neural

crest predates the first vertebrate genome duplication, as paralogs of genes from the SoxE, Tfap2, and Twist families retain neural crest-related functions. This finding provides further support for an early origin of the neural crest and aligns with the identification of parts of the neural crest gene regulatory network in cells at the tunicate neural plate border. Overall, I found this to be an interesting section of the manuscript that unfortunately concludes with speculative, open-ended statements.

The section on hagfish-specific gene losses and duplications has similar issues. It is important to annotate the extensive gene loss observed in the hagfish, particularly considering its simplified body plan. However, this section presents a collection of observations that do not clearly coalesce into an impactful idea.

The final section of the manuscript focuses on the intricate process of programmed DNA elimination, which is considered one of the most enigmatic developmental processes in vertebrate development. By conducting a comparison between the germline and soma (blood?) genomes, the authors meticulously document the genome regions that undergo loss during hagfish embryogenesis. In my assessment, this section of the paper appeared to be the weakest, primarily due to technical shortcomings that should be addressed prior to publication.

The results obtained from the FISH analysis presented in Figure S10 exhibit noise and do not align entirely with the observed loss of the tested regions in the soma. To enhance the clarity and robustness of their findings, the authors may wish to include multiplexed FISH data on both germline and soma in the main figure (Fig 5) and validate the loss of FISH signal of germline-specific genomic regions. The conclusion that lost genes are targets of developmental signaling pathways is not particularly convincing. Perhaps plotting the statistics for this analysis would strengthen this claim.

In terms of the paper's structure, I found the data presentation in certain sections to lack clarity for the general audience. It requires considerable effort to navigate between the main figures, supplemental data, and methods to grasp the rationale and main findings of some of the analyses. This is particularly true for Figure 3, which would benefit from the inclusion of some of the supplemental data. Also, Figure 5 is poorly integrated into the text, with the results depicted in the different panels lacking proper explanation within the manuscript text. Overall, the selection of the material that is displayed in supplemental versus main figures could be significantly improved.

Minor:

- Typo on the dates in Figure S6

Referee #3:

Remarks to the Author:

"The hagfish genome and the evolution of vertebrates" by Marlétaz et al

In this paper the authors describe the sequencing of the brown hagfish (*E. atami*) as well as several evolutionary analyses of the genome. They confirm the monophyly of cyclostomes (no surprise) as does the other hagfish paper, they present a reconstruction of macro chromosome evolution (hard to follow and interpret due to sparse details), confirm ancient polyploidies and How cluster evolution, investigate gene content evolution, and look at the patterns of programmed genome rearrangement.

This is the second of two hagfish genome papers (two different species) to recently appear in BioRxiv. This one presents the genome of the brown hagfish *E. atami*. It is interesting to compare the two papers, and unfortunately, this paper is inferior in presentation and in analysis and interpretation.

The genome will undoubtedly be valuable resource when it is publicly available as a second hagfish genome for comparison, but I do not think that the analysis presented is of sufficient interest or novelty for the broad readership of your journal.

Unfortunately, this manuscript is very sloppy and appears to have been submitted when it was not quite finished. It was a quite frustrating experience to try to review it, and this might show in the way these comments are presented. Many of the figures are non-intuitive, not explained, and some contain errors in the labelling or legends. There are places in the tables where there are apparent notes between the authors, not intended for publication. The supplementary files were not all submitted to the journal and the GitHub were were are supposed to retrieve them has broken links and typos. It was very challenging to review this manuscript because of how badly it is written and how disorganised it is. I can in fact only provide a partial opinion, because there are some parts that are simply opaque and I cannot comment on whether the analyses and interpretation are correct because the descriptions are inadequate. The authors provide sparse details of important aspects, and the manuscript has several internal contradictions.

In multiple places in this paper the authors have used the previously published *E. burgeri* hagfish genome. I think they need to be more clear about how much they relied on this for their analysis and appropriately cite the other hagfish genome paper.

On the ancestral genome reconstructions: The authors present a model with 18 inferred porto-vertebrate chromosomes but do not cite Nakatani et al (2021) as the first who showed this. The reconstruction here appears to differ from the previously published version by some of the same authors (Simakov et al 2020) which only had 17 inferred ancestral chromosomes. However the details of this new reconstruction are sparse and comparison with the Nakatani reconstruction seems to be absent. As it stands the chromosomal evolution analysis is not well described and it is unclear what can be learned from it. It is also unclear which reconstruction the authors are using because sometimes they refer to 18 ancestral chromosomes (fig2) and other times there are only 17 (fig 3, tables S6 & S7) - it is very hard to interpret what has been done here given these inconsistencies and lack of explanation. What is the value of this new reconstruction over the previously published ones? What is novel here?

Regarding the monophyly of cyclostomes, I have no doubt that this is the correct result, but the analysis is not necessarily as robust as the authors claim. They state that the cyclostome monophyly topology is robust to compositional heterogeneity. They use CATGTR and recoding to help with this, but if they want to support this claim of robustness they need to do posterior predictive simulations using the DIV and COMP tests in Phylobayes to actually back it up. Otherwise they should remove the claim that their methods are robust to this heterogeneity.

On the prolonged rediploidization (page 8, line 29 onwards), this seems to be heavy on the discussion but quite light on the actual analysis and data. The Supp tables are not very useful because the data provided are quite superficial. There are several good papers with useful methods and approaches to revealing prolonged rediploidization and the authors should consider adopting some of those methods.

Fig 4 is very confusing and not well described. I cannot figure out the details of panel a; in panel b what is meant by 'optional' retention, and how is that different from the cases (which are also presented) or retention and non-retention?; panel c is completely non-intuitive to me and not well explained; panel d seems to have incorporated analysis of the *Eptatretus burger* genome, so that should be properly cited (now that it is published); panel e, what is the significance of the coloured labels?; panel f, what is the significance of the stars in the keratin cluster? How do the coloured boxes/genes above correspond to the gene names below??

Regarding the gene loss analysis (page 12, line 38) the authors should have included closer outgroups (tunicates/urochordates) for comparison; the authors also need to consider that deuterostomes may not be monophyletic (Kapli et al 2021) and how this impacts their analysis.

Fig 5: panel a - what does the 2.3Gb above the graph refer to?; panel b, what is the justification for these p-value thresholds? The legend says that the colour scheme is shared in panels a, b, [d, e], but they are representing entirely different things -- this makes no sense.

Fig S7 - panel a is not intuitive and not explained. What is it representing?

In sum, while I am confident that with less rushing and more care and attention the authors can address the problems in this manuscript, and that the genome will be a useful resource, I nonetheless think that there is insufficient novelty to be of interest to the readership of this journal.

Other points:

Regarding the cyclostome genome duplication. The hexaploidy event inferred by Nakatani et al was inferred to have occurred in the proto-cyclostome, that is, shared with hagfish, and not only in lamprey as implied here (page 5 line 22-3)

For suppl file 2/3 (numbering is off and I'm not sure which it is meant to be) - it is very hard to find

the associated alignments underlying the paralogon trees. The github repository needs tidying and maybe fleshing out. It is important to provide these data

Table S5 includes analysis of the first hagfish genome - please cite their paper

Table S6 includes a note "needs to be finished?" Signed by the first author. Is this the finished version of the table??

Table S7 only has 17 ancestral chromosomes, not 18. Why? Also, this table seems to contain several notes between the authors not intended for publication. Is this the finished version??

Page 16 line 43 - only Nakatani previously proposed hexaploidy, not the other papers

Page 17 line 11 - please cite additional sources for delayed and prolonged rediploidization

Fig S7 legend has two parts (d) ... which one is correct?

Author Rebuttals to Initial Comments:**Referee #1 (Remarks to the Author):**

Overview

This manuscript reports a genome assembly and analysis for a hagfish species. Lampreys and hagfishes are the only two surviving lineages of jawless fish. Together they occupy a unique position in vertebrate phylogeny and hence a unique opportunity to gain insight into early vertebrate evolution. Lamprey genomes have been sequenced and analysed already. This manuscript offers the comparator genome from hagfishes, where study has lagged behind lamprey due to the difficulty in obtaining specimens and much larger genome size.

Assembling a high-quality genome facilitates comparative genomic studies. Some are outward looking, essentially focusing on how the early vertebrate genome evolved both prior to the separation of jawless and jawed vertebrate genomes, and within the jawless vertebrate lineage. Others are inward looking, focusing on the peculiar biology of hagfishes including their slime production and somatic genome reorganisation.

In general, I found the analyses robust. I have multiple comments/views about individual sections and which I will detail below. I have refrained from a more forensic review of the text at this stage as I think there is substantial work to do to make this suitable for publication and that might come later.

My overarching view of the manuscript is mixed. There are some genuinely insightful aspects that pin down important parts of vertebrate evolutionary history and should be of broad relevant and interest. There are also parts which are much less compelling, either because they address questions that have already been substantially advanced elsewhere, or because they are incomplete, speculative or do not address the interesting questions in those sub-areas.

In summary I think there is potential for this manuscript to make an exciting contribution to the literature that will be widely respected, and cited well and for a long time. However, I suggest to focus the paper onto the big questions here: chromosome evolution, genome duplication and the fates of duplicated genes. Possibly also somatic genome rearrangement, as I think this is a major outstanding question of genome function/evolution into which hagfishes are well placed to give insight, though as my comments below will show I struggled to understand what the authors were saying here.

Thanks. We have revised our manuscript to emphasise these big questions related to chromosome evolution, genome duplication and the fates of duplicated genes, and clarify our discussion of somatic genome rearrangement (i.e., germline-specific genes and elements), taking into account criticisms and suggestions below. Conversely we de-emphasized phylogenomics and reduced speculative comments. We hope that these revisions, including additional analyses to clarify several key insights, have improved the manuscript.

General comments on individual sections

1. Chromosome evolution

This analysis is robust, important and interesting. Together with the section on genome duplications it seals off one of the big remaining questions of vertebrate genome evolution

Thanks. This was a major focus of our effort. We agree that we have established a comprehensive model for vertebrate genome evolution, supported by both chromosome-scale and gene-scale phylogenetic analyses. In the revision we have added an analysis of lineage-specific rediploidisation in early vertebrate evolution, finding evidence for hagfish- and lamprey- lineage-specific rediploidization involving the Hox bearing chromosome.

2. Phylogenomics

This is technically robust but essentially confirmatory. Although the craniate hypothesis has some historical relevance, it has been clear for many years now that hagfishes and lampreys are monophyletic. Its an appropriate supplementary analysis for this paper but I felt its prominence here as an important finding and opening figure of the paper was overstated.

Thanks. As noted we probably overstressed the classical “craniate vs.cyclostome” debate in the original submission, and have now streamlined the introduction to de-emphasize this point. As suggested we moved most of the discussion of this analysis to Supplementary Note 1 and Supplementary Figure 2, but continue to show Figure 1b to orient the reader.

Nevertheless, since this classical question had not been revisited in nearly 15 years (Delsuc et al. 2008), we felt it was important to bring to bear new datasets and phylogenetic techniques to test this hypothesis. In the revised submission, we emphasise these methodological improvements including site-heterogeneous model via CAT-GTR, and an increased attention for compositional heterogeneity given the known high GC% of lamprey and to a lesser extent hagfish. At the request of Referee 3 we also included results of posterior predictive tests (DIV and COMP) to quantify these effects. Our analysis also takes advantage of increased taxon sampling including representatives of all major hagfish and lamprey groups, (vs. only a single species for hagfish and lamprey in the pioneering early studies), eightfold increase in the number of genes used, and the advantage of an orthologue sets developed from complete hagfish and lamprey genomes (which are much less prone to inadvertent use of paralogs than transcriptome based datasets). This does not take up much space in the main text (one paragraph and a main figure panel (Fig. 1b) that is needed anyway to orient the reader) but we think it is a worthwhile contribution, with details now found in Supplementary Note 1 and Supplementary Figure 2.

3. Genome duplications

This section adds two important findings on genome duplications: their timing relative to lineage splits and their true timing as deduced by clock analyses. I think these are both pivotal

outcomes of this paper and with (1) above provide a compelling picture of early vertebrate genome evolution.

Thanks! We are excited to have nailed down a comprehensive scenario for vertebrate genome duplications with both “relative” and “true” timings that have previously been elusive.

4. Hox clusters

The analysis in this section is well done and interesting to the many who work on Hox genes, though essentially descriptive. Biological insight as discussed is very speculative however.

We have revised this discussion and added additional analyses, integrating it with new analysis of rediploidisation. We present the first detailed scenario for the expansion of a single proto-vertebrate Hox cluster into the multiple Hox clusters of various vertebrate lineages through the various genome duplications. In particular we now highlight a case of lineage-specific rediploidization that affects two of the cyclostome Hox clusters, a novel aspect of Hox evolution. Regarding our comments on Evx gene evolution, functional studies have been performed by other groups demonstrating the differential roles of EVX paralogs in gnathostomes, as cited, but the improved understanding of genome evolution afforded by our analyses allows us to interpret those studies in a new evolutionary light.

5. Neural crest

As with the Hox clusters, I found the insight gained from this section to be a bit limited. It also seemed to ignore some of the published experimental work on lamprey neural crest genes, for example PMID: 17765683. I think there is insight to be gained from the approach, however, but would just need to be better developed and integrated with current functional knowledge (and possibly new expression data) to be persuasive.

Thanks. We agree that it is useful to consider lamprey work on neural crest and as suggested we have revised and expanded our analysis of the paralogues involved in the specification and migration of migratory neural crest cells (NCC) to include experimental work performed in lamprey. To do that, we established the paralogy status of gene previously studies in lamprey NCC (e.g., Hockman et al. 2019) by aligning them to the transcripts of the most recent lamprey genome (kPetmar1). We also reanalysed published RNA-seq data (Martik et al. 2019) to examine expression level of different lamprey paralogues for NCC-related genes. These analyses are shown in updated Figure 4a and the new Supplementary Figure 9 showing paralogue expression in lamprey NCCs.

We have also streamlined our presentation. The key question we address is whether migratory neural crest arose before or after the shared duplication on the vertebrate stem (1R_V). If neural crest arose after 1R_V, then we would expect migratory neural crest functions to be found in some paralogs arising from the duplication, with others retaining

ancestral (i.e., pre-migratory neural crest) function. Conversely, the involvement of both paralogues derived from this first duplication in NCC-related functions indicates that these functions predated the 1R_V event. Our revised analyses address this more precisely and shows that neural crest-associated gene functions predated 1R_V. We also identified a few cases where neural crest genes have diverged in their function after the divergence of cyclostomes and gnathostomes. While logically it was sufficient for our argument to document this in gnathostomes (since gnathostomes and cyclostomes share 1R_V), it is of course stronger to expand the discussion to include the extensive published studies of sea lamprey, and we have done so as suggested.

6. Parologue fates

The analysis in this section looks robust. I found the outcomes interesting: particularly that there are differences in parologue fates in cyclostomes. This follows nicely on from the sections on the timing and consequences of chromosome evolution/genome duplication.

Thanks!

7. Hagfish gene novelties and losses

This section combines two analyses: a generic view of gene loss and some selected 'biology insight' analyses. The former fits well with the sections on parologue fates above. Insight from specific gene losses that may be linked to lost phenotypic traits like eyes is, however, limited. Many analogous cases have been discovered but really we learn little from such analyses without some functional follow-on.

We think the propensity of loss of entire gene families is in interesting contrast with the multiple genome duplication events that took place in this lineage, as the presence of multiple paralogues makes gene family loss more difficult. We found that gene losses in hagfish could be related to characters that were evidently lost in hagfishes relative to lampreys and gnathostomes based on cyclostome monophyly. To do that, we curated the literature and particularly relied on existing functional work performed in gnathostome models (e.g. mouse or zebrafish) to comment on gene function. To avoid sounding overly speculative we have streamlined this section in the revision to make it more synthetic, and focused on fewer key findings.

8. Somatic genome rearrangement

In my opinion this is one of the big, fascinating questions about this clade of animals: how and why does such drastic somatic DNA loss occur? A complete genome is a fundamental basis for analysing this and I was hoping that having hagfish and lamprey as parallel examples would really shed insight into this peculiar biology. However I struggled to understand what the authors were concluding here. The study of repeats is important in a sense, but what does it tell us? Similarly, it is interesting to see that genes removed by somatic recombination seem to be mostly non-overlapping between lamprey and hagfish, but again it wasn't clear what this actually meant.

We have completely reorganised and rewritten this section to bring key “biological” (i.e., gene-related) insights to the forefront and reduce the description of computational and methodological aspects as requested by the Referees. We hope it will make it clearer how our results advance the understanding of programmed DNA elimination based on the first identification of germline-specific genes in any hagfish.

We also include a new section on amplification of germline-specific duplicates within germline-specific chromosomes. In total there are 1,654 genes deleted from somatic tissue progenitors, with functions related to cell cycle, cell motility and chromatin/DNA repair (none of these were known previously, since no germline-specific genes had ever been described in any hagfish). Importantly, these genes are typically fast-evolving paralogs of genes that are retained on somatic chromosomes, which provide the first insights into patterns that drive the gene content of germline-specific genes in hagfish. This is perhaps a not widely understood feature of programmed DNA elimination in lampreys and birds (e.g., finches).

Repeats are a key feature of somatic DNA elimination for three reasons, but we have streamlined this discussion. First, prior knowledge of eliminated sequence in hagfish was limited solely to repeats and analyses of germline-specific repeat content provide an important connection to previous studies. Second, repetitive sequences account for the majority of the content of the eliminated chromosomes, including one new sequence discovered in our analysis that accounts for nearly 4% of the genome. Finally, in lamprey (where functional studies are possible) repetitive elements likely play a functional role in DNA loss. Functional analysis in hagfish will require the development of embryological methods and descriptions of pre-gastrula development that are not yet possible given the rarity of hagfish embryos.

Other aspects

The supplementary figures are an integral part of this manuscript and essential to evaluate the analyses. However their utility is seriously undermined by a lack of description in the legends. For example, it is just about possible to work out what many graphs actually show by referring back to the methods, but even this can be challenging and that’s speaking as someone with significant experience in this area. If this paper progresses and these transfer to extended data figures, its vital they be accompanied by proper explanatory legends.

We apologise that these legends appeared scarce and seemed incomplete in the original submission. We have made sure to make them as clear and descriptive as possible in the resubmitted manuscript.

Referee #2 (Remarks to the Author):

I have read the manuscript authored by Marletáz et al. with great interest, as the hagfish remains a poorly studied organism despite its significant phylogenetic position. The authors present a high-quality chromosome-scale assembly of the hagfish genome, which is utilized to explore important aspects of both vertebrate evolution and the unique biology of this species.

The first contribution of the paper is to provide further support for the monophyly of cyclostomes. The authors analyze a large set of orthologous genes using a collection of models, demonstrating that lampreys and hagfish share a common ancestor independent of gnathostomes. My impression was that the debate between the Cyclostomata vs Craniata hypotheses had been settled by a series of molecular studies performed in the 00's. Thus, while re-visiting the phylogeny of vertebrates with this new genome assembly is certainly warranted, I found this section of the study to be of limited impact.

Thanks. We probably overstressed the classical "craniate vs.cyclostome" debate in the original submission, and have now streamlined the introduction to de-emphasize this point. As suggested we moved most of the discussion of this analysis to Supplementary Note 1 and Supplementary Figure 2, but continue to show Figure 1b to orient the reader.

Nevertheless, since this classical question has not been revisited in nearly 15 years (Delsuc et al. 2008), we felt it was important to bring to bear new datasets and phylogenetic techniques. We now emphasise these methodological improvements including site-heterogeneous model via CAT-GTR, and an increased attention for compositional heterogeneity given the known high GC% of lamprey and to a lesser extent hagfish. At the request of Referee 3 we included results from posterior predictive tests (DIV and COMP) to quantify these effects. Our analysis also takes advantage of increased taxon sampling including representatives of all major hagfish and lamprey groups, (vs. only a single species of hagfish and lamprey in the pioneering early studies), used eightfold more genes, and took advantage of orthologue sets developed from complete hagfish and lamprey genomes (which are much less prone to inadvertent use of paralogs than transcriptome based datasets). This does not take up much space in the main text (one paragraph and a main figure panel (Fig. 1b) that is needed anyway to orient the reader) but we think it is a worthwhile contribution, with details to be found in Supplementary Note 1 and Supplementary Figure 2.

Some parts of the study are compelling. For instance, the authors employ paralogon-based analysis of hagfish chromosomal organization, unveiling that its "simpler" karyotype is a derived state resulting from multiple fusion-with-mixing events. This finding is particularly relevant as it refutes two commonly held assumptions in the field: firstly, that the unusually complex karyotype observed in lampreys reflects a derived state, and secondly, that the large number of small chromosomes in lampreys is somehow linked to mechanisms of programmed DNA elimination. The results presented by the authors convincingly demonstrate that neither of these speculations holds true.

Thanks. In the revised manuscript, we have emphasised the dramatic contrast between the karyotypes of lamprey and hagfish, and how our analyses show that hagfish are the chromosomally derived group. We also have included more comparative discussion of germline-specific genes in the two cyclostomes to make clear general similarities in programmed DNA elimination between the deeply diverged hagfish and lamprey.

The study also tackles the sequence and timing of genome duplications in early vertebrate evolution. The authors employ paralogon-based phylogenetic analysis to propose an auto-tetraploidization event shared by vertebrates and establish the timing of independent duplications in gnathostomes and cyclostomes. The strategy for this analysis is well-designed, but the conclusions do not appear particularly novel, as they seem to confirm the ideas of cyclostome hexaploidy proposed by Nakatani and colleagues (Nat. Comm. 2021). The major contribution here lies in demonstrating that hexaploidy is shared by both hagfish and lamprey, which in itself is a significant finding.

Thanks. Regarding novelty, in our revised introduction we have tried to clarify what is, and what is not, currently firmly known about vertebrate duplications and their timing. We provide (1) the first molecular phylogenetic analysis (using synteny-based paralogons to enhance signal) that resolves all duplication and speciation nodes in early vertebrates, and (2) a uniform set of estimated dates for these events that can be correlated with the geological record.

While the $1R_V$ autotetraploidy proposal is not new (see, most recently, Simakov 2020, and also Nakatani 2021), the evidence for shared stem-vertebrate duplication is largely based on parsimony. In particular it has not previously been definitively shown that a single $1R$ predated the cyclostome-gnathostome split. The present manuscript is the first time the proposal has been tested with molecular phylogeny, which removes some ambiguity in the synteny-based model (our Simakov et al. 2020, and Nakatani et al. 2021) and provides absolute dates for the divergence of progenitors via molecular clock.

As you note, one of our major contributions is demonstrating that hexaploidy is shared by both hagfish and lamprey. Although Nakatani et al announce “cyclostome hexaploidy” in their title, a careful reading of their manuscript shows that they really only provide evidence for “lamprey hexaploidy.” Their argument for extending their (Northern) lamprey-based findings to the cyclostome ancestor is based solely on the *assumption* that the duplication history of hagfishes and lampreys are the same. This assumption is stated in the legend of their Fig 2: “It is presently considered that the hagfish and lamprey lineages share the same duplication history[27], but this argument should eventually be confirmed by sequencing the hagfish genome.” (Reference 27 is to Pascual-Anaya 2018, who only showed that hagfish have the same number of Hox clusters as lampreys, and did not even establish a 1:1 correspondence.) Given the dramatic karyotypic differences between hagfishes and lampreys, the “same duplication history” can hardly be taken for

granted. The definitive answer to this question requires a genome-scale hagfish-lamprey comparison as provided by our study.

We have also now added a new section on rediploidization after polyploidization which shows that for some chromosomes lineage-specific rediploidization occurred after the shared cyclostome polyploidy, which we find occurred over an extended period of time. These findings (both timing and rediploidization) go well beyond Nakatani et al, who did not perform molecular phylogenetic analysis.

Next, the authors delve into the evolution of different gene groups, such as Hox genes and neural crest genes, in light of their paralogon-based phylogenies. They propose that the origin of the neural crest predates the first vertebrate genome duplication, as paralogs of genes from the SoxE, Tfap2, and Twist families retain neural crest-related functions. This finding provides further support for an early origin of the neural crest and aligns with the identification of parts of the neural crest gene regulatory network in cells at the tunicate neural plate border. Overall, I found this to be an interesting section of the manuscript that unfortunately concludes with speculative, open-ended statements.

Thanks. We have rewritten the concluding statements to be more precise and revised this section to incorporate published gene expression and functional data from lamprey. The key point is that, having established $1R_V$ on the vertebrate stem, we show that paralogs on both branches of $1R_V$ have neural crest function in both cyclostomes and gnathostomes, which strongly supports the existence of migratory neural crest prior to $1R_V$. The inclusion of lamprey data further strengthens our conclusion.

The section on hagfish-specific gene losses and duplications has similar issues. It is important to annotate the extensive gene loss observed in the hagfish, particularly considering its simplified body plan. However, this section presents a collection of observations that do not clearly coalesce into an impactful idea.

As suggested, we streamlined this section around fewer important ideas and made it more concise. In particular we highlight the loss of genes involved in the biology of organs specifically lost in the hagfish lineages (eyes, limited ossification), which we think are quite striking examples given the historical discussion of the 'primitive' body plan of hagfish.

The final section of the manuscript focuses on the intricate process of programmed DNA elimination, which is considered one of the most enigmatic developmental processes in vertebrate development. By conducting a comparison between the germline and soma (blood?) genomes, the authors meticulously document the genome regions that undergo loss during hagfish embryogenesis. In my assessment, this section of the paper appeared to be the weakest, primarily due to technical shortcomings that should be addressed prior to publication.

The results obtained from the FISH analysis presented in Figure S10 exhibit noise and do not align entirely with the observed loss of the tested regions in the soma. To enhance the clarity and robustness of their findings, the authors may wish to include multiplexed FISH data on both germline and soma in the main figure (Fig 5) and validate the loss of FISH signal of germline-specific genomic regions. The conclusion that lost genes are targets of developmental signaling pathways is not particularly convincing. Perhaps plotting the statistics for this analysis would strengthen this claim.

We have rewritten this section to bring key “biological” (i.e., gene-related) insights to the forefront as requested by Referees. We also include a new section on amplification of germline-specific duplicates within germline-specific chromosomes. In total there are 1,654 genes deleted from somatic tissue progenitors, with functions related to cell cycle, cell motility and chromatin/DNA repair (none of these were known previously). Importantly, these genes are typically fast-evolving paralogs of genes that are retained on somatic chromosomes, which provide the first insights into patterns that drive the gene content of germline-specific genes in hagfish. This is perhaps a not widely understood feature of programmed DNA elimination in lampreys and some birds (e.g., finches).

Regarding what you referred to as “noise” in the original multiplexed FISH data in Figure S10 (now Figure S13), we have revised this figure and its associated legend to make it easier to understand. Please note that for each probe, panels marked “no signal” show somatic nuclei are overexposed to both show background signal and aid in confirming the location of nuclei in those images. (These images would otherwise be black.) We have clarified this point in the figure legend. Our key finding is that FISH hybridization signals from different repeat probes are present only in germline cells and do not overlap in interphase nuclei. We conclude that these repetitive sequences are present in non-overlapping arrays in the chromatin fiber in germline and that these same regions are absent from soma.

If we are interpreting the second suggestion properly, the reviewer is asking for us to incorporate the multiplexed FISH data and validation into main Figure 5. While it would be technically possible to include our multiplex FISH for the 11 validated germline-specific repeats (using data provided in Supplementary Figures 13 and 14) and three somatic repeats in a main body figure, we feel that such a figure would be quite unwieldy. We think that our illustration of the location of validated repeats by FISH on a germline metaphase spread (which we note is quite difficult to obtain) provides a more succinct summary of our identification and validation of the germline-specific repeats.

Regarding the request for p-value reporting in relation to ontology enrichment analyses, these were given in the original Figure 5 (now panel e) and detailed test statistics (and p-values) are given for two different methods of enrichment analyses in Supplementary Tables S14 and 15). However, we understand where this might have been confusions, and have modified the figure legend to aid in the interpretation of presented p-values,

which now refers more directly to Supplementary Table S14. Part of the analyses related to inferring higher-order regulatory relationships (previous Table S15) was removed in the interest of clarity and space. We hope these revisions clarify the question raised by the referee.

In terms of the paper's structure, I found the data presentation in certain sections to lack clarity for the general audience. It requires considerable effort to navigate between the main figures, supplemental data, and methods to grasp the rationale and main findings of some of the analyses. This is particularly true for Figure 3, which would benefit from the inclusion of some of the supplemental data. Also, Figure 5 is poorly integrated into the text, with the results depicted in the different panels lacking proper explanation within the manuscript text. Overall, the selection of the material that is displayed in supplemental versus main figures could be significantly improved.

Thanks for this comment. In the revision we have extensively revised our manuscript to hopefully make it more understandable to a general audience and to improve the data presentation, relationship between text and figures, and navigation. As suggested, this includes a reframing of the original Figure 3 (duplication scenarios, dating, Hox, now in revised Figures 2 and 3) and a better integration of Figure 5 (DNA elimination) with the main text. The revised manuscript has been read by several colleagues who provided additional helpful feedback, and we hope that it avoids the problems noted with the original.

Minor:

- Typo on the dates in Figure S6

Thanks, we have corrected this.

Referee #3 (Remarks to the Author):

"The hagfish genome and the evolution of vertebrates" by Marlétaz et al

In this paper the authors describe the sequencing of the brown hagfish (*E. atami*) as well as several evolutionary analyses of the genome. They confirm the monophyly of cyclostomes (no surprise) as does the other hagfish paper, they present a reconstruction of macro chromosome evolution (hard to follow and interpret due to sparse details), confirm ancient polyploidies and How cluster evolution, investigate gene content evolution, and look at the patterns of programmed genome rearrangement.

This is the second of two hagfish genome papers (two different species) to recently appear in BioRxiv. This one presents the genome of the brown hagfish *E. atami*. It is interesting to compare the two papers, and unfortunately, this paper is inferior in presentation and in analysis and interpretation.

The genome will undoubtedly be valuable resource when it is publicly available as a second hagfish genome for comparison, but I do not think that the analysis presented is of sufficient interest or novelty for the broad readership of your journal.

We respectfully disagree with this evaluation of our manuscript and work, as discussed further below.

Unfortunately, this manuscript is very sloppy and appears to have been submitted when it was not quite finished. It was a quite frustrating experience to try to review it, and this might show in the way these comments are presented. Many of the figures are non-intuitive, not explained, and some contain errors in the labelling or legends. There are places in the tables where there are apparent notes between the authors, not intended for publication. The supplementary files were not all submitted to the journal and the GitHub were were are supposed to retrieve them has broken links and typos. It was very challenging to review this manuscript because of how badly it is written and how disorganised it is. I can in fact only provide a partial opinion, because there are some parts that are simply opaque and I cannot comment on whether the analyses and interpretation are correct because the descriptions are inadequate. The authors provide sparse details of important aspects, and the manuscript has several internal contradictions.

We apologise for the problems with the original submission, which have been rectified in the revision.

In multiple places in this paper the authors have used the previously published *E. burgeri* hagfish genome. I think they need to be more clear about how much they relied on this for their analysis and appropriately cite the other hagfish genome paper.

To be very clear, the analyses we describe are based on a chromosome-scale *E. atami* genome sequence and annotation that we generated specifically for this long-running

project, which was independent of any other hagfish genome sequencing effort. The suggestion that we inappropriately used other datasets is simply incorrect.

In our gene family analyses, we used the gene models of *E. burgeri* to (i) assess gene loss in the hagfish lineage and (ii) evaluate the timing of expansion of germline-specific gene families identified from our *E. atami* analysis. In this context we used the protein-coding genes from *E. burgeri* reported in

- Nishimura O, Yamaguchi K, Hara Y, Tatsumi K, Smith JJ, Kadota M, and Kuraku S. “Inference of a genome-wide protein-coding gene set of the inshore hagfish *Eptatretus burgeri*” *F1000Research* 2022, 11:1270 (<https://doi.org/10.12688/f1000research.124719.1>).

One of us (JJ Smith) was a co-author of this peer-reviewed report, whose stated purpose was to make this data publicly and freely available. In our original submission, we did cite this paper as a preprint (Yamaguchi et al. 2020). In the revised manuscript, it is now cited in its peer reviewed form as Nishimura et al. 2022. We have included this citation in both places where this data was used.

In light of your comment about bioRxiv at the top of this report, however, please note that Nishimura et al. *describe a completely independent sequencing project for E. burgeri* relative to the subsequent biorxiv paper by Yu et al. 2023. We did not use any *E. burgeri* data from Yu et al. 2023 (or any derivative of Yu et al. 2023 available at Ensembl).

On the ancestral genome reconstructions: The authors present a model with 18 inferred proto-vertebrate chromosomes but do not cite Nakatani et al (2021) as the first who showed this. The reconstruction here appears to differ from the previously published version by some of the same authors (Simakov et al 2020) which only had 17 inferred ancestral chromosomes. However the details of this new reconstruction are sparse and comparison with the Nakatani reconstruction seems to be absent. As it stands the chromosomal evolution analysis is not well described and it is unclear what can be learned from it. It is also unclear which reconstruction the authors are using because sometimes they refer to 18 ancestral chromosomes (fig2) and other times there are only 17 (fig 3, tables S6 & S7) - it is very hard to interpret what has been done here given these inconsistencies and lack of explanation. What is the value of this new reconstruction over the previously published ones? What is novel here?

Thanks. We see the confusion, which is hopefully remedied by the revised text and new Supplementary Note 2, and summarised below. To clarify, we did not produce a “new reconstruction,” but instead use the 18 ancestral linkage groups previously defined by Simakov et al. 2022 based on a comprehensive analysis of bilaterian chromosome evolution, including multiple outgroups. These units were designated as A1, A2, and B-Q, which is the terminology that we use here. The use of multiple outgroups makes Simakov et al. 2022 a definitive and stable reconstruction of the vertebrate stem linkage groups. This was noted (perhaps too cryptically) original Figure 2 whose inset showed these 18 labels and their associated colours as they are projected onto the hagfish and

sea lamprey genomes. In the revised manuscript these panels (with inset) are now found in Figure 1c,d, with further description in the legend and main text.

As now more explicitly noted, the 18 linkage groups of Simakov et al. 2022 are in 1:1 correspondence with the 18 Pvc's of Nakatani et al. 2021. In particular CLGs B-Q (which are the same as those described by Simakov et al. 2020) have already been shown to be equivalent to Pvc's 1-16 of Nakatani et al. 2021 (see Supplementary Note 3.1.2 and Supplementary Table 9 of Nakatani et al. 2021). CLGA of Simakov et al. 2020, however, was split into A1 and A2 by Simakov 2022 based on the finding that these two parts are fused and mixed in amphioxus but are consistently (and therefore ancestrally) separate in multiple other bilaterians. Our A1 corresponds to Pvc17 and A2 corresponds to Pvc18, which is evident by comparisons to scallop chromosomes used by Nakatani et al. and Simakov et al. The correspondence between A1/A2 and Pvc17/18 is now made explicit in our revised text.

For our paralogon-based molecular phylogenetic analyses of vertebrate genome duplications we require that paralogons be present in both cyclostomes and gnathostomes as well as invertebrate outgroups (which is why we use Simakov et al. 2022). As now more clearly stated, A2 is small and has too few consistently linked genes in gnathostomes for robust phylogenetic analysis, and so is excluded from these analyses. This explains why we report only 17 paralogon trees (A1 and B-Q). The exclusion of A2, and reason for it, is now explicitly noted in the main text. The omission of A2 from our paralogon-based molecular phylogeny is consistent with several anomalous features of the corresponding Pvc18 pointed out by Nakatani et al. 2021. Specifically, Pvc18 is (1) poorly represented in gnathostomes (2) could not be segmented by their method due to extensive fragmentation in lampreys, and (3) was excluded from analysis of lamprey hexaploidy. These anomalous features are now noted with citation to Nakatani et al. 2021.

We hope that this explanation and the revised text (with additional discussion Supplementary Note 2) clarifies the 17/18 confusion noted in the original submission. Thank you for encouraging us to clarify this.

(Finally, in light of your initial comment about the Yu et al. 2023 preprint, please note that the sixteen "ancestral chromosomes" AC1-16 defined by Yu et al. are the same as the 17 Simakov et al. 2020 CLGs, with CLG I and CLG Q (incorrectly) combined. This correspondence can be read off (tediously) by comparing their Extended Data Figure 6 to Simakov et al. 2020's Extended Data Figure 8 (which shows *P. marinus* scaffolds vs. CLGs) or, alternatively, our present Figure 1. It follows that the the "ancestral chromosomes" of Yu et al. are (1) not a new result, (2) incorrectly fuse A1 and A2 contra Nakatani 2021 and Simakov 2022, and (3) incorrectly fuses I and Q (Pvc9 and 8) contra both Simakov et al. 2020 and Nakatani et al. 2021.)

Regarding the monophyly of cyclostomes, I have no doubt that this is the correct result, but the analysis is not necessarily as robust as the authors claim. They state that the cyclostome monophyly topology is robust to compositional heterogeneity. They use CATGTR and recoding to help with this, but if they want to support this claim of robustness they need to do posterior predictive simulations using the DIV and COMP tests in Phylobayes to actually back it up. Otherwise they should remove the claim that their methods are robust to this heterogeneity.

Thanks for this suggestion. We wanted to discuss and rule out the possibility that the divergent base composition of lamprey and to a lesser extent hagfish could induce an artifactual topology. As suggested we have added results from the posterior predictive tests on amino acid diversity (DIV) and compositional heterogeneity (COMP) to the revised manuscript. These results indicate that the fit of the dataset to the model greatly improves using CAT-GTR and particularly with a Dayhoff6 recoding, thereby supporting our claims that the monophyly of cyclostome was likely not due to such bias. Details are provided in Supplementary Table 4, Supplementary Figure 2d and the new Supplementary Note 1. We have also streamlined the discussion of vertebrate phylogeny in the introduction to place less emphasis on the craniate-vs-cyclostome debate.

On the prolonged rediploidization (page 8, line 29 onwards), this seems to be heavy on the discussion but quite light on the actual analysis and data. The Supp tables are not very useful because the data provided are quite superficial. There are several good papers with useful methods and approaches to revealing prolonged rediploidization and the authors should consider adopting some of those methods.

Thanks. As suggested we have now added new analyses addressing the question of lineage-specific rediploidization, which we hope you will find interesting. In particular, we (1) find no evidence for lineage-specific rediploidization after $1R_V$ but (2) find some chromosomal segments that show such a signal after the cyclostome-specific polyploidization $2R_{CY}$.

There are several places in the text, however, where we cannot provide such a quantitative analysis of delayed rediploidization because it appears to be complete before the next available speciation node. In these cases we make the qualitative but important point that a duplication event and divergence of resulting paralogs are not the same. This occurs in two contexts:

- Reiploidization after autotetraploidy: we find that paralogs arising from the $1R_V$ duplication on the vertebrate stem diverged an estimated ~527 Ma. This is of course not an estimate of the date of the tetraploidization event itself, but rather of the cessation of homoeologous recombination, which follows autotetraploidization by some unknown interval. In noting this distinction we cited Furlong and Holland 2002 (who made this important conceptual point relatively early) and the more recent salmon papers that detected regionally delayed rediploidization from more recent

duplications. In the revision we have also added a citation to Redmond et al. 2023's elegant work on the sturgeon-paddlefish case.

- Progenitor split vs allotetraploidy: In the discussion of the gnathostome-specific $2R_{JV}$, we note that the divergence of the alpha and beta progenitors ~508 Mya does not correspond to the allo-tetraploidization itself, which occurred some unknown time later when alpha and beta progenitors hybridized. To give some context we use *Xenopus* and goldfish/carp as recent vertebrate allotetraploid analogs of $2R_{JV}$ for which estimates of the delay between progenitor divergence and allo-hybridization are available.

(In light of your initial comment about the Yu et al. 2023 bioRxiv preprint, please note that they do not mention these these subtle but important points about the timing of duplication events relative to molecular clock dates at all, and refer to molecular split times as duplication times.)

Fig 4 is very confusing and not well described. I cannot figure out the details of panel a; in panel b what is meant by 'optional' retention, and how is that different from the cases (which are also presented) or retention and non-retention?; panel c is completely non-intuitive to me and not well explained; panel d seems to have incorporated analysis of the *Eptatretus burger* genome, so that should be properly cited (now that it is published); panel e, what is the significance of the coloured labels?; panel f, what is the significance of the stars in the keratin cluster? How do the coloured boxes/genes above correspond to the gene names below??

We are sorry that this figure and its legend were so confusing; we tried to pack a lot of information in it and we should have been more careful that everything remained perfectly clear. We have revised the figure and particularly the legend to improve clarity:

- We now give more detail in the legend to help interpret panel b : “Each column corresponds to a species and a specific pattern of post-duplication retention: ‘retained’ indicates families where both paralogues have been preserved, while ‘optional’ indicate that enrichment is considered without regard for whether paralogues were retained for the subsequent cyclostome ($2R_{CY}$) or gnathostome ($2R_{JV}$) duplication.” Our goal was to examine the functional enrichment for different lists of genes showing alternative retention patterns, which we hope is now clear.
- We also rephrase the explanation of panel c as “Distribution of the difference of positive expression domains between select vertebrate species and the amphioxus outgroup for ohnologue gene families. A shift to the left in the distribution (as seen for the gar) indicates an extensive subfunctionalization through the restriction of gene expression domains.”
- For panel f, we specified: “Gene are displayed in the same order in the heatmap as they are located in the two clusters. Stars indicate the two genes that are expressed preferentially in the skin” .

We hope these changes make the interpretation of the different panels easier.

Regarding the gene loss analysis (page 12, line 38) the authors should have included closer outgroups (tunicates/urochordates) for comparison; the authors also need to consider that deuterostomes may not be monophyletic (Kapli et al 2021) and how this impacts their analysis.

Thanks. The main purpose of this analysis was to measure gene family loss in vertebrates. We originally omitted tunicates since they are known outliers that are evolving very fast and underwent extensive gene loss, Nevertheless, as suggested we have now included two tunicates (*Ciona intestinalis* and *Styela clava*) in our gene family reconstruction (see revised Figure 4d). As expected, this inclusion indicates extensive gene loss in the tunicate lineage but does not affect the accounting of gene losses in vertebrates relative to our original analysis without tunicates, demonstrating the robustness of our findings.

Regarding the possibility that deuterostomes may not be monophyletic, we are focused on gene losses within chordates or vertebrates, whose monophyly has not been questioned. Hemichordates and echinoderms clearly represent outgroups to chordates, and both have a slower rate of evolution than protostomes (as noted in Kapli et al). Therefore, even allowing for the possibility of deuterostome paraphyly we think that the choice of a different outgroup would not improve our analyses in its detection of gene gain and losses patterns.

Fig 5: panel a - what does the 2.3Gb above the graph refer to?; panel b, what is the justification for these p-value thresholds? The legend says that the colour scheme is shared in panels a, b, [d, e], but they are representing entirely different things -- this makes no sense.

Thanks for pointing out these issues with Figure 5, which have been addressed in the revised manuscript.

- Panel a shows the distribution of relative copy number of different sequences. There is a large peak at copy number 1 (log copy number = 0) that would otherwise swamp the detail shown. We have therefore cut off the y-axis, and included the "2.3 Gb" to indicate that the peak around unit copy number shared by both germline and soma accounts for this much total sequence. Additional modifications to this section suggested by other referees should make this clearer.
- In panel b (and Supplementary Tables S13 and 14), we report standard p-values for ontology enrichment tests, and do not use p-value thresholds.
- Regarding the colour scheme, we apologise for this error, and have modified the figure legend to refer to the correct panels. The colour scheme is meant to aid the reader in cross-referencing analyses of germline-enriched repeats and applies only to those panels. We had reorganised the figure just before the original submission, but had not adjusted the figure legend accordingly. Thanks for pointing this out.

Fig S7 - panel a is not intuitive and not explained. What is it representing?

Thanks. Supplementary figure 7a (now 10a) shows “Count of gene families presenting given number of retained paralogues in gar, lamprey (*P. marinus*) and hagfish (*E. atami*).” We have expanded and clarified the legend to make this clearer.

In sum, while I am confident that with less rushing and more care and attention the authors can address the problems in this manuscript, and that the genome will be a useful resource, I nonetheless think that there is insufficient novelty to be of interest to the readership of this journal.

We apologise for any incompleteness in the original submitted materials, and have addressed these in the revised submission.

We strongly disagree, however, with the assessment of the novelty of our findings. We hope that the revision makes this clearer.

With the hagfish genome in hand we provide the first comprehensive history of vertebrate genome evolution that includes all major extant lineages. We use our paralogon-based trees and recently developed molecular-phylogenetic approaches (e.g., WHALE, lineage-specific rediploidization methods), to resolve remaining ambiguities in phylogenetic signal that are the product of duplication, rediploidization and speciation over the deep history of vertebrates. We provide what we believe to be the first robust estimates for the timing of these events, integrating genomic data with the deep fossil record.

While there has been speculation about delayed rediploidization in early vertebrate evolution since Furlong and Holland 2002, this has never been systematically analysed. We now find that (1) rediploidization after $1R_V$ was complete before the gnathostome-cyclostome split, but (2) rediploidization after $2R_{CY}$ continued past the hagfish-lamprey split. occurrence of this phenomenon in cyclostomes after $2R_{CY}$. These have major impacts on the interpretation of comparative data from cyclostomes. As expected, we found no evidence for lineage-specific rediploidization after the gnathostome allotetraploidy $2R_{JV}$.

We use the reconstructed history of vertebrate genomes to map the origin and diversification of migratory neural crest relative to $1R_V$, implying a vertebrate stem origin for neural crest. Similarly, body plan simplification in hagfish is implied by cyclostome monophyly, but the molecular underpinnings of this secondary simplification have not previously been discussed; we show that hagfish has experienced extensive gene family loss, including genes related to character loss. Conversely, we find a gene family expansion associated with the iconic hagfish slime.

Finally, we report the first germline specific genes from any hagfish along with insights into the germline-specific function of these genes. Importantly, these genes are typically fast-evolving paralogs of genes that are retained on somatic chromosomes, which

provide the first insights into patterns that drive the gene content of germline-specific genes in hagfish.

Other points:

Regarding the cyclostome genome duplication. The hexaploidy event inferred by Nakatani et al was inferred to have occurred in the proto-cyclostome, that is, shared with hagfish, and not only in lamprey as implied here (page 5 line 22-3)

We have carefully revised our introduction to ensure that proper credit is given to Nakatani et al. 2021 and other authors for prior findings and hypotheses, as was our original intent. Nakatani et al. 2021 performed their chromosomal analysis on the Japanese and sea lamprey genomes and infer a shared paleo-hexaploidy in the history of these (Northern hemisphere) lampreys. Their claim that this hexaploidy is shared with hagfishes, however, depends crucially on hagfishes and lampreys sharing the same duplication history, as stated in the legend of their Figure 1: “It is presently considered that the hagfish and lamprey lineages share the same duplication history²⁷, but this argument should eventually be confirmed by sequencing the hagfish genome.” Reference 27 is the hagfish Hox cluster study of Pascual-Anaya et al. (2018) which only shows that hagfish and lamprey have the same number of Hox clusters. Pascual-Anaya et al. 2018 does not even show a 1:1 correspondence between these six cyclostome Hox clusters as would be expected, and leaves open the possibility of independent expansions. Otherwise we could find no additional evidence provided by Nakatani et al. 2021 for their titular claim of cyclostome hexaploidy. We therefore consider “lamprey hexaploidy” to be their “finding” but “cyclostome hexaploidy” to be a “hypothesis” that we explicitly credit to Nakatani et al. 2021. This logical dependency is now hopefully clarified in our revised introduction.

For suppl file 2/3 (numbering is off and I'm not sure which it is meant to be) - it is very hard to find the associated alignments underlying the paralogon trees. The github repository needs tidying and maybe fleshing out. It is important to provide these data

Thanks for pointing this out. We apologise for the mistake in numbering the supplementary files. The alignments underlying the paralogon trees were previously provided (accessible as https://github.com/fmarletaz/hagfish/tree/7fd484a6eec07f6025e3c19e13b87c7e32f96b86/Paralogons/test6/clgAliRx2_loose) but we agree that the repository could be better organised. We have therefore extensively streamlined the directory structure, removed some redundant files as suggested, and include an informative readme file in each main subfolder describing one of the key analyses of the paper. Alignments for 'relaxed' and 'strict' paralogon reconstructions are now included in 'relaxed/alis_relax' and 'strict/alis_strict'.

Table S5 includes analysis of the first hagfish genome - please cite their paper

Thanks. This table lists the genomes used in gene family reconstruction. As noted above for *E. burgeri* we used data from Nishimura et al. 2022. We now call this out explicitly in the main text wherever this data is used to avoid any confusion with Yu et al. 2023 preprint. As noted above the citation is

- Nishimura O, Yamaguchi K, Hara Y, Tatsumi K, Smith JJ, Kadota M, Kuraku S. "Inference of a genome-wide protein-coding gene set of the inshore hagfish *Eptatretus burgeri*" *F1000Research* 2022, 11:1270 (<https://doi.org/10.12688/f1000research.124719.1>).

We emphasise again that these data are completely independent of the Yu et al. 2023 bioarxiv manuscript.

Table S6 includes a note "needs to be finished?" Signed by the first author. Is this the finished version of the table??

We apologise for uploading this table with the comments unresolved, indicating items to be double checked. It has been updated.

Table S7 only has 17 ancestral chromosomes, not 18. Why? Also, this table seems to contain several notes between the authors not intended for publication. Is this the finished version??

This is now Table S5. As described above, we have clarified the reason for restricting our attention to these 17 units in the main text and in the new Supplementary Note 2. We have also stated in the note at the bottom of Table S5 the reason that A2 was not done, with pointer to the Supplementary Note 2. As noted above linkage group A2 of Simakov 2022 (corresponding to Pvc18 in Nakatani 2021) was too small (i.e., had too few linked genes in various species) for robust phylogenetic analysis, and was therefore not used in our paralogon phylogenies. We also note in the new Supplementary Note 2 that this is consistent with Nakatani 2021's description of Pvc18 as being anomalous in its conservation in gnathostomes and its fragmented distribution in lampreys. Please note that Pvc18 is similarly excluded from some analyses in Nakatani et al. 2021, including analysis of hexaploidy (which uses "K=17", see their Methods).

Page 16 line 43 - only Nakatani previously proposed hexaploidy, not the other papers

Thanks. We have rewritten this sentence to separate the Nakatani et al. 2021 citation from the others to ensure that credit for the cyclostome hexaploidy hypothesis is clearly and specifically attributed to Nakatani et al. 2021. Smith et al. 2013, 2015, 2018, and Simakov et al. 2020 found additional large-scale/ genome-wide duplication(s) on the lamprey lineage but did not specifically propose hexaploidy, although, as noted by Nakatani et al. 2021, Smith et al. 2013 did mention the possibility of hexaploidy in passing.

Page 17 line 11 - please cite additional sources for delayed and prolonged rediploidization

We originally cited Furlong and Holland 2002 for the important conceptual point that duplication and rediploidization (i.e., divergence of paralogs) are distinct, and cited several pioneering salmon papers for direct demonstration of this effect. We have now added considerable further analysis and discussion, and now also cite Parey et al. 2022 (on rediploidization in teleost fish), Robertson et al. 2017 and Gundappa et al 2022 (salmonids), and the elegant recent work by Redmond et al. 2023 on lineage-specific rediploidization in sturgeon and paddlefish. This seems to be a comprehensive set of references, but we are open to other suggestions.

Fig S7 legend has two parts (d) ... which one is correct?

Thanks, we have corrected this.

Reviewer Reports on the First Revision:

Referees' comments:

Referee #1 (Remarks to the Author):

In my original review of this manuscript I identified some significant strengths (the nailing down (my words) of chromosome evolution, timing of genome duplications and fates of duplicated genes), and some aspects that were not so strong for a variety of reasons (phylogenomics, Hox clusters, Neural crest, hagfish genetic novelties and somatic recombination).

The authors have attempted to address these weaknesses, and on the whole have done a good job. I'll detail a bit more on these below, in the order of what I think is most interesting/important, to the least interesting/important. As before it's important to note that I don't have any reservations about the quality of the data/ analyses themselves: these are well done and robust, and now that the figures and extended data figures have full explanatory legends they are also much easier to follow.

1. Somatic recombination: this section reads so much better and is genuinely interesting. As well as properly examining overlap between lost gene sets in lamprey and hagfish, the authors start down the route of identifying how this has evolved through analysis of gene expansions.
2. Neural crest. This is also much more precise. Hypothesis is explicit and the outcome clear. And interesting and important. The extension of these arguments into trunk vs cranial neural crest is still quite speculative however, as the number of relevant genes is quite low. I understand why the authors might want to include this, but it is still something of a footnote to the study from Martik et al 2019.
3. Hox clusters. By embedding this analysis more explicitly in their paralog analysis, this now adds well to the overall paper. The addition on *Evx* genes is speculative, however
4. Phylogenomics. The authors have cut this section back following comments from myself and other reviewers. However, I take their logic that they anyway need to show a tree early on in the paper and if they do that, it might as well be one that springs from their analysis. This is persuasive and for phylogeny aficionados a supplementary note is now there to provide the detail missing in the main manuscript.
5. Hagfish genetic novelties. I am still unpersuaded by this section. While I understand the logic, and agree with the results, it is still a set of speculative correlations between genetic differences and major, ancient phenotypic changes.

Overall I think this is a much improved version. I suggest the authors and editors many wish to further focus the manuscript, but fundamentally it is an exciting piece of work that will have lasting impact on the broader field of genome evolution.

Referee #2 (Remarks to the Author):

This is a revised version of the manuscript by Marletaz and colleagues, which presents an in-depth exploration of the hagfish genome. I found the results of the revision to be mixed. The modifications included in the manuscript strengthened data quality and presentation, but its accessibility and impact remain limited. Sections of the study reaffirm conclusions made by previous reports (to their credit, this is acknowledged by the authors through expanded citations).

Overall, I found the paper offers a collection of confirmatory insights, but with greater rigor than previous studies.

The additions to the study highlight its technical merit, such as the robustness of its methodology used in the whole genome duplication section, the quality of the genome assembly, as well as the analyses concerning synteny and ancestral chromosomes. Data presentation is rigorous - I am persuaded that the high-quality genome allows the authors to resolve the timing of vertebrate genome duplications with higher confidence than published reports.

However, two significant issues remain. First, the manuscript remains difficult to follow. This is mostly due to the breadth of the analysis – my impression is that the authors are trying to do too much in a short article by taking the studies to multiple fronts like genome duplication, chromosome evolution, gene loss, etc. Despite clarifications in the results section and figure legends, the reach of the manuscript is still limited - and it is hard for me to imagine it will be a satisfying read for non-specialists.

Second (perhaps for the same reasons), the conclusions in sections of the study remain broad and mostly speculative. For instance, the examination of the genome reduction in somatic cells is interesting but does not progress beyond cataloging of the genes that were lost. The detailed account of this process does not lead to a new insight or overarching conclusion.

This is also clear in the discussion about the loss of genes related to the reduced morphology of some hagfish characteristics (eyes, ossification). The Gene Ontology analysis serves as a disappointing conclusion of that section, which is mostly a detailed description of the process. This limitation is perhaps best illustrated by the last sentence of the manuscript "...and reaffirm the importance of gene loss as an evolutionary force". The study documents the gene losses in the hagfish, but does not demonstrate this as an evolutionary force, only that it happened. And as it has been extensively debated, losing genes may be a consequence and not the cause of a reduction in morphology.

In summary, the study provides important pieces of the puzzle of the evolution of the vertebrate genome - but the broad scope of topics covered presents challenges for clarity and focus.

Referee #3 (Remarks to the Author):

The new version of the manuscript is much improved over the very sloppy original submission. I wish the authors had put this work in the first time. The paper is easier to read and follow now.

The clarifications of figure 4 were useful, but I still find some elements quite confusing. In fig 4b, the 'optional' class is still a little obscure to me. Does the first column (Hagfish retained after 1R but 'optional' after 2Rcy) overlap with the 6th column (Hagfish retained after 1R but lost after 2Rcy)? Logically it seems to me that they do overlap, or the description of 'optional' is inadequate. But if they do overlap, can the authors interpret why they are not grouping in the heatmap? In fact, whereas the two gnathostomes always seem to group by retention pattern, the hagfish and lamprey only have one instance (retention post 1R and loss after 2Rcy) where the cyclostomes are grouped for the same retention pattern. How do the authors interpret, for example, the apparently stark difference in the retained post1R and 'optional' post2Rcy for hagfish and lamprey? I see only one sentence on page 13 (line 12) on differential loss in lamprey and hagfish, but I am still left puzzling over what fig 4b shows. Seeing as the closing sentiment of the entire paper is on the evolutionary importance of gene loss, it would seem of central importance to clarify this.

I cannot find any explanation why only some retention patterns are considered. Whereas all retention patterns are displayed for gnathostomes, the pattern of retention post 1R AND post 2Rcy is not shown. Was it examined and excluded from the figure? I am struggling to see how this figure should be properly interpreted.

I also don't understand the comment in the legend that 'hagfish shows a lesser degree of enrichment of gene ontology terms classically associated with 2R paralogues, such as embryonic development and nervous system activity'. The final column of fig 4b shows the hagfish post-2Rcy retention, and these are high for those categories ... or am I totally misreading this figure??

On page 16-17 the analysis of the programmed DNA elimination is potentially the most novel part of this paper, but I am a bit confused by the new text on page 17. By my reading there are several statements made without providing adequate evidence to support them. In the middle paragraph on page 17 (lines 19-20) the authors write that the gremlin specific chromosomes are a 'testing ground for selection' and that it is a 'selective environment', but this seems to be purely speculative. If there are indeed more gene duplication events in these parts of the genome (I can't find a quantitative analysis of this in the paper — did I miss it?) then the authors need to consider the possibilities of a mutation bias and/or a lack of constraint. It would seem obvious to me that parts of the genome that are eliminated after development can tolerate different kinds of disturbances than those that must remain properly regulated throughout the lifespan of the animal. If instead this section is instead a case study of some gene families of interest, then the text should be rewritten accordingly.

Minor points:

Also cite Lien et al (2016) for delayed rediploidisation (page 2-3)

Lien, S. et al. The Atlantic salmon genome provides insights into rediploidization. *Nature* 533, 200–205 (2016).

Final sentence (page 18), cite Maynard Olsen (1999) for the idea of gene loss as an evolutionary force

Olson, M. V. When less is more: gene loss as an engine of evolutionary change. *American journal of human genetics* 64, 18–23 (1999).

Author Rebuttals to First Revision:

Referees' comments:

Referee #1 (Remarks to the Author):

In my original review of this manuscript I identified some significant strengths (the nailing down (my words) of chromosome evolution, timing of genome duplications and fates of duplicated genes), and some aspects that were not so strong for a variety of reasons (phylogenomics, Hox clusters, Neural crest, hagfish genetic novelties and somatic recombination).

The authors have attempted to address these weaknesses, and on the whole have done a good job. I'll detail a bit more on these below, in the order of what I think is most interesting/important, to the least interesting/important. As before its important to note that I don't have any reservations about the quality of the data/ analyses themselves: these are well done and robust, and now that the figures and extended data figures have full explanatory legends they are also much easier to follow.

Thanks.

1. Somatic recombination: this section reads so much better and is genuinely interesting. As well as properly examining overlap between lost gene sets in lamprey and hagfish, the authors start down the route of identifying how this has evolved through analysis of gene expansions.

Thanks!

2. Neural crest. This is also much more precise. Hypothesis is explicit and the outcome clear. And interesting and important. The extension of these arguments into trunk vs cranial neural crest is still quite speculative however, as the number of relevant genes is quite low. I understand why the authors might want to include this, but it is still something of a footnote to the study from Martik et al 2019.

Thanks. We have attempted to streamline the trunk/cranial discussion. Ours is the first analysis of neural crest evolution in the context of a complete picture of vertebrate genome evolution,

3. Hox clusters. By embedding this analysis more explicitly in their paralog analysis, this now adds well to the overall paper. The addition on Evx genes is speculative, however

Thanks. We have moved the discussion of the evolution of Evx duplicates and their possible relation to limb evolution to the rewritten discussion section at the end of the paper, where we can discuss it in a more integrative context.

4. Phylogenomics. The authors have cut this section back following comments from myself and other reviewers. However, I take their logic that they anyway need to show a tree early on in the paper and if they do that, it might as well be one that springs from their analysis. This is persuasive and for phylogeny aficionados a supplementary note is now there to provide the detail missing in the main manuscript.

Thanks.

5. Hagfish genetic novelties. I am still unpersuaded by this section. While I understand the logic, and agree with the results, it is still a set of speculative correlations between genetic differences and major, ancient phenotypic changes.

We have rewritten and shortened this section. Given the hagfish genome, it would be perverse not to say something about the implications for hagfishes! We document the extensive gene losses in the hagfish lineage relative to lampreys, and speculate that this is associated with the character losses implied by the cyclostome hypothesis. We also find expansions in gene families associated with slime, one of the defining characters of hagfishes. These genes are expressed in slime gland and clearly associated with the trait. In our opinion these are essential points to make in our manuscript and we hope that the shortened text makes them succinctly.

Overall I think this is a much improved version. I suggest the authors and editors many wish to further focus the manuscript, but fundamentally it is an exciting piece of work that will have lasting impact on the broader field of genome evolution.

Thanks for your helpful comments throughout the review process!

Referee #2 (Remarks to the Author):

This is a revised version of the manuscript by Marletaz and colleagues, which presents an in-depth exploration of the hagfish genome. I found the results of the revision to be mixed. The modifications included in the manuscript strengthened data quality and presentation, but its accessibility and impact remain limited. Sections of the study reaffirm conclusions made by previous reports (to their credit, this is acknowledged by the authors through expanded citations).

Overall, I found the paper offers a collection of confirmatory insights, but with greater rigor than previous studies.

Thanks. Over the years many hypotheses about the nature and timing of vertebrate genome duplications have been proposed, many with very limited evidence. For example, the proposal by Nakatani et al. that the lamprey hexaploidy they identified occurred in a common cyclostome ancestor relied entirely on the matching number of hox clusters in hagfishes and lampreys. To our knowledge, there have been no global molecular phylogenetic analysis that supports a comprehensive scenario for early genome vertebrate genome duplications. With the hagfish genome, and our paralogon-based approach, we have reached this goal, which inevitably confirms previously made hypotheses as cited.

The additions to the study highlight its technical merit, such as the robustness of its methodology used in the whole genome duplication section, the quality of the genome assembly, as well as the analyses concerning synteny and ancestral chromosomes. Data presentation is rigorous - I am persuaded that the high-quality genome allows the authors to resolve the timing of vertebrate genome duplications with higher confidence than published reports.

Thanks.

However, two significant issues remain. First, the manuscript remains difficult to follow. This is mostly due to the breadth of the analysis – my impression is that the authors are trying to do too much in a short article by taking the studies to multiple fronts like genome duplication, chromosome evolution, gene loss, etc. Despite clarifications in the results section and figure legends, the reach of the manuscript is still limited - and it is hard for me to imagine it will be a satisfying read for non-specialists.

Thanks for this comment. We have taken it to heart and extensively streamlined the manuscript to (1) make it more accessible to non-specialists, and (2) highlight our main results, focused on vertebrate genome evolution in the light of the hagfish genome. The “multiple fronts” referred

to are simply different aspects of this perspective, and by simplifying the text throughout (and adding a new Supplementary Note 3 on germline repeats to allow that section to be focused on gene evolution) we hope that the paper will be a more satisfying read for non-specialists.

Second (perhaps for the same reasons), the conclusions in sections of the study remain broad and mostly speculative. For instance, the examination of the genome reduction in somatic cells is interesting but does not progress beyond cataloging of the genes that were lost. The detailed account of this process does not lead to a new insight or overarching conclusion.

We have rewritten several of the section-level conclusions and moved what we feel are the key broader implications to a revised and more integrative discussion at the end of the paper. Regarding genes lost in somatic cells, we show that these are associated with germline functions, and that these genes are (1) derived from somatic genes, and (2) experiencing ongoing duplication on the germline chromosomes. Combined with emerging findings in lampreys and songbirds these point to general trends in germline-specific chromosome evolution.

This is also clear in the discussion about the loss of genes related to the reduced morphology of some hagfish characteristics (eyes, ossification). The Gene Ontology analysis serves as a disappointing conclusion of that section, which is mostly a detailed description of the process. This limitation is perhaps best illustrated by the last sentence of the manuscript "...and reaffirm the importance of gene loss as an evolutionary force". The study documents the gene losses in the hagfish, but does not demonstrate this as an evolutionary force, only that it happened. And as it has been extensively debated, losing genes may be a consequence and not the cause of a reduction in morphology.

Regarding the discussion of hagfish-specific features, we have streamlined this section, but we feel that it is important to point out the more extensive gene loss in hagfish and how some of these losses are correlated with morphological simplification.

Regarding the last sentence and gene loss as an evolutionary "force," we have removed this speculative discussion and completely rewritten the concluding section of the paper due to both space constraints and to better emphasize the implications of our findings.

In summary, the study provides important pieces of the puzzle of the evolution of the vertebrate genome - but the broad scope of topics covered presents challenges for clarity and focus.

Thanks for these helpful comments. In the revision we have attempted to streamline the manuscript to make it more accessible to non-specialists while still presenting what we feel are a broad range of exciting results that emerge from our hagfish genome analysis.

Referee #3 (Remarks to the Author):

The new version of the manuscript is much improved over the very sloppy original submission. I wish the authors had put this work in the first time. The paper is easier to read and follow now.

Thanks for your helpful comments.

The clarifications of figure 4 were useful, but I still find some elements quite confusing. In fig 4b, the 'optional' class is still a little obscure to me. Does the first column (Hagfish retained after 1R but 'optional' after 2Rcy) overlap with the 6th column (Hagfish retained after 1R but lost after 2Rcy)? Logically it seems to me that they do overlap, or the description of 'optional' is inadequate. But if they do overlap, can the authors interpret why they are not grouping in the heatmap? In fact, whereas the two gnathostomes always seem to group by retention pattern, the hagfish and lamprey only have one instance (retention post 1R and loss after 2Rcy) where the cyclostomes are grouped for the same retention pattern. How do the authors interpret, for example, the apparently stark difference in the retained post1R and 'optional' post2Rcy for hagfish and lamprey? I see only one sentence on page 13 (line 12) on differential loss in lamprey and hagfish, but I am still left puzzling over what fig 4b shows. Seeing as the closing sentiment of the entire paper is on the evolutionary importance of gene loss, it would seem of central importance to clarify this.

I cannot find any explanation why only some retention patterns are considered. Whereas all retention patterns are displayed for gnathostomes, the pattern of retention post 1R AND post 2Rcy is not shown. Was it examined and excluded from the figure? I am struggling to see how this figure should be properly interpreted.

I also don't understand the comment in the legend that 'hagfish shows a lesser degree of enrichment of gene ontology terms classically associated with 2R paralogues, such as embryonic development and nervous system activity'. The final column of fig 4b shows the hagfish post-2Rcy retention, and these are high for those categories ... or am I totally misreading this figure??

Thank you for catching this; it is correct that (i) some of these retention patterns are overlapping and (ii) we had inadvertently omitted one pattern for hagfish and lamprey. We reran the analysis and revised the figure to make the retention pattern nomenclature explicit and test all patterns in gnathostomes and cyclostomes. In this more comprehensive GO enrichment heatmap, similar cyclostome retention patterns group together (and similarly for gnathostomes). In general, similar retention patterns in gnathostomes and cyclostomes display the same functional enrichments, with subtle differences. Paralogs that were retained after 1R_v-only show a weaker enrichment in terms associated with developmental functions in cyclostomes than in gnathostomes (e.g. "anterior/posterior pattern specification", "regulation of neurogenesis"). This

trend is in the opposite direction for paralogs retained after all duplications (i.e. higher retention of genes linked with developmental functions in cyclostomes versus gnathostomes), suggesting a higher retention of development-related paralogs following $2R_{CY}$ as opposed to $2R_{IV}$. We have updated the figure legend and main text accordingly.

Regarding the overarching importance of gene loss, we have rewritten the concluding remarks to emphasize other findings and de-emphasized gene loss.

On page 16-17 the analysis of the programmed DNA elimination is potentially the most novel part of this paper, but I am a bit confused by the new text on page 17. By my reading there are several statements made without providing adequate evidence to support them. In the middle paragraph on page 17 (lines 19-20) the authors write that the gremlin specific chromosomes are a 'testing ground for selection' and that it is a 'selective environment', but this seems to be purely speculative. If there are indeed more gene duplication events in these parts of the genome (I can't find a quantitative analysis of this in the paper — did I miss it?) then the authors need to consider the possibilities of a mutation bias and/or a lack of constraint. It would seem obvious to me that parts of the genome that are eliminated after development can tolerate different kinds of disturbances than those that must remain properly regulated throughout the lifespan of the animal. If instead this section is instead a case study of some gene families of interest, then the text should be rewritten accordingly.

We have revised this discussion to emphasize that hagfish provides a third, deeply divergent vertebrate lineage that allows us to test ideas that have been proposed based on emerging data from germline-specific chromosomes in lamprey and songbirds. As you point out, it is possible that the mutational environment is also important with respect to the evolution of germline-specific genes, as was also previously proposed, and we have incorporated this into the revised text.

Minor points:

Also cite Lien et al (2016) for delayed rediploidisation (page 2-3)

Lien, S. et al. The Atlantic salmon genome provides insights into rediploidization. *Nature* 533, 200–205 (2016).

Thanks, we have added this citation.

Final sentence (page 18), cite Maynard Olsen (1999) for the idea of gene loss as an evolutionary force

Olson, M. V. When less is more: gene loss as an engine of evolutionary change. *American journal of human genetics* 64, 18–23 (1999).

Thanks, we have rewritten the conclusions based on your comments and those of reviewer 2 and deemphasized gene loss.

Reviewer Reports on the Second Revision:

Referees' comments:

Referee #3 (Remarks to the Author):

The authors have revised and clarified their manuscript and I have no further comments.

I just note that in fig2c and d the numbers beside the nodes should be explicitly explained in the legend.